# Policy Gradient Methods Converge Globally in Imperfect-Information Extensive-Form Games

**Fivos Kalogiannis**
UCSD CSE
La Jolla, CA 92093
fkalogiannis@ucsd.edu

**Gabriele Farina**
MIT EECS
Cambridge, MA 02139
gfarina@mit.edu

## Abstract

Multi-agent reinforcement learning (MARL) has long been seen as inseparable from Markov games (Littman, 1994). Yet, the most remarkable achievements of practical MARL have arguably been in extensive-form games (EFGs)—spanning games like Poker, Stratego, and Hanabi. At the same time, little is known about provable equilibrium convergence for MARL algorithms applied to EFGs as they stumble upon the inherent nonconvexity of the optimization landscape and the failure of the value-iteration subroutine in EFGs. To this goal, we utilize contemporary advances in nonconvex optimization theory to prove that regularized alternating policy gradient with (i) *direct policy parametrization*, (ii) *softmax policy parametrization*, and (iii) *softmax policy parametrization with natural policy gradient updates* converge to an approximate Nash equilibrium (NE) in the *last-iterate* in imperfect-information perfect-recall zero-sum EFGs. Namely, we observe that since the individual utilities are concave with respect to the sequence-form strategy, they satisfy gradient dominance with respect to the behavioral strategy—or, *policy*, in reinforcement learning terms. We exploit this structure to further prove that the regularized utility satisfies the much stronger proximal Polyak-Łojasiewicz condition. In turn, we show that the different flavors of alternating policy gradient methods converge to an $\epsilon$-approximate NE with a number of iterations and trajectory samples that are polynomial in $1/\epsilon$ and the natural parameters of the game. Our work is a preliminary—yet principled—attempt in bridging the conceptual gap between the theory of Markov and imperfect-information EFGs while it aspires to stimulate a deeper dialogue between them.

## 1 Introduction

Reinforcement learning (RL) dominates contemporary applied and theoretical research. The flagship of RL, *policy optimization methods*, appears to lend reasoning capabilities to language models (Shao et al., 2024), defeats human Go world champions (Silver et al., 2016), and navigates real-world roads safely (Lu et al., 2023; Cusumano-Towner et al., 2025). As is evident from even more examples (Vinyals et al., 2019; Schrittwieser et al., 2020), machine gameplay has transformed by incorporating RL techniques into its algorithmic arsenal. Although theoretical literature (Littman, 1994) posits that the canonical model of MARL are Markov games (MGs), MARL has handled imperfect-information extensive-form games (EFGs) with commendable success (Brown and Sandholm, 2019b; Bard et al., 2020; Perolat et al., 2022).

At first, the theory and practice of imperfect-information EFGs can seem saturated. Exhaustive research in the properties of EFGs has exposed its convex structure using *sequence-form* strategies (Romanovskii, 1962; Koller et al., 1996; Von Stengel, 1996) and yielded the different counterfactual-regret minimization algorithms (CFR) (Zinkevich et al., 2007; Tammelin, 2014; Brown and Sandholm,

2019a). These algorithms can solve games using tabular policies with unmatched computational efficiency. Notwithstanding, these techniques seem to hit a wall when faced with large-scale games whose size makes the use of tabular policies infeasible and calls for a *neural network parametrized policy* (or, more generally, policy function approximation). The picture is even more grave when CFR needs to be combined with model-free counterfactual value estimation. Its call for importance sampling yields a feedback of prohibitively high variance. Further, CFR's average-iterate convergence makes the task of extracting a single policy network highly nontrivial. Since practitioners have extensively studied policy optimization for imperfect-information games (Lanctot et al., 2017; Srinivasan et al., 2018; Lockhart et al., 2019; Hennes et al., 2020; Rudolph et al., 2025) without offering guarantees of polynomial time convergence, we are naturally lead to the question:

> *Do policy gradient methods provably converge to an equilibrium in*      (♥)
> *imperfect-information EFGs using a polynomial number of iterations and samples?*

To answer, we need to face the two obstacles that imperfect-information games raise against optimization, the failure of value iteration—*which we sidestep by solely using policy gradient updates*—and a highly nonconvex policy optimization landscape—*which we prove to be benign*.

**Failure of value iteration**     In MARL for MGs, the overwhelming majority (Shapley, 1953; Wei et al., 2021; Zhao et al., 2022; Alacaoglu et al., 2022b; Zhang et al., 2019) of existing algorithmic solutions for equilibrium learning or computation makes use of a *value iteration* subroutine or a *value critic*—which is in essence a backwards induction of the estimated value of the game. Instead, solving imperfect-information games requires leveraging the opponent's uncertainty about the underlying state. In other words, one needs to trade off exploiting private information and the benefit of keeping it secret. This precludes solving subtree-by-subtree conditioned on private information and leads to the emergence of behaviors such as bluffing at optimality.

**Gradient Domination in Nonconvex Problems.**     Contemporary machine learning is arguably propelled by large-scale optimization of systems of astounding size to perform increasingly elaborate tasks. The corresponding objective functions are by no means convex in terms of parameters, which precludes theoretical guarantees of even reaching a local optimum in a reasonable number of iterations (Murty and Kabadi, 1985). Yet, practice indicates a different reality and theory is gradually catching up. It has painstakingly been demonstrated that the nonconvexity of various ML optimization problems is seriously benign—significantly often, *stationarity implies global optimality*. Cases in point, gradient domination is exhibited for *the loss functions of overparametrized neural networks* (Liu et al., 2022a; Scaman et al., 2022), *the linear quadratic regulator* (Fazel et al., 2018), *value functions of Markov decision processes (MDPs)* (Agarwal et al., 2021; Bhandari and Russo, 2024), *matrix completion* (Ge et al., 2016), *dictionary learning* (Sun et al., 2015), and more. For a thorough discussion of gradient domination and other regularity conditions we refer the reader to (Karimi et al., 2016; Li and Pong, 2018; Drusvyatskiy and Paquette, 2019; Drusvyatskiy and Lewis, 2018; Liao et al., 2024; Rebjock and Boumal, 2024; Oikonomidis et al., 2025) and references therein. With the latter in mind, one could make the case that when game theory researchers seek equilibrium computation in general nonconvex games (Cai et al., 2024a; Angelopoulos et al., 2025) they set the bar too high. Still, the study of benign nonconvexity seems of great importance and rather underexplored (Yang et al., 2020; Mulvaney-Kemp et al., 2023; Vlatakis-Gkaragkounis et al., 2021; Sakos et al., 2023).

## 1.1 Contributions

We answer (♥) in the affirmative by developing three policy gradient methods (Theorems 3.1 to 3.3). All three algorithmic approaches lead to last-iterate convergence to a regularized NE of the EFG. We contribute,

- a novel decentralized exploration scheme that yields sufficient visitation of all information sets;
- a proof that the nonconvex utilities of the (un-)regularized game satisfy gradient domination;
- guarantee of last-iterate convergence of three different alternating policy gradient (PG) methods: (1) PG with *direct parametrization* and $\ell_2$-*norm regularization* (2) PG *softmax parametrization* and *entropy regularization* (3) *natural policy gradient* (NPG) with *softmax parametrization* and *entropy regularization*.

On a sidenote, we offer a sharper dependence of the PŁ modulus to the hidden convexity modulus than the one suggested by (Karimi et al., 2016, Appendix G) for constrained optimization.

## 1.2 Overview of Techniques

The theoretical guarantees for our three algorithmic solutions are pinpointed by a simple unifying conceptual principle. That is, *the nonconvex optimization problem* of computing an equilibrium by directly optimizing the behavioral strategies (or, policies) *is a constrained two-sided* PŁ *optimization problem* where alternating gradient descent ascent is known to converge. Namely, we show that the optimization landscape viewed in terms of *policies* is nonconvex in a rather benign way; the utility is *hidden concave*. In particular, after appropriate regularization, each utility function satisfies a strong gradient domination property, *i.e.,* the proximal Polyak-Łojasiewicz condition.

**Hidden concavity.** Going into more detail, utilities in EFGs are concave in terms of *sequence-form* strategies. We select an appropriate *regularizer* that enhances concavity to strong concavity. Moreover, enforcing a positive lower bound on the probability of reaching every information set yields a uniform Lipschitz constant for the bijection that maps sequence-form strategies to behavioral policies. Taken together, these two observations imply a strong gradient-domination condition for each player's policy.

**PŁ condition.** For the sake of offering an intuitive exposition, we forego the nuances of constrained optimization to explain how the PŁ condition is proven to hold. We say that an optimization problem $\min_x f(x)$ exhibits *hidden strong convexity* when there exists an invertible mapping $u = c(x)$ and a function $H(u)$ that is $\mu$-strongly convex in $u$ and $f(x) = H(c(x))$. Strong convexity implies that $f(x) - f^\star \equiv H(u) - H^\star \leq \frac{1}{2\mu} \|\nabla_u H(u)\|^2$. Now, a bounded Lipshcitz modulus $L_{c^{-1}} > 0$ of the inverse transform, $c^{-1}(u) = x$, leads to the PŁ inequality $f(x) - f^\star \leq \frac{L_{c^{-1}}^2}{2\mu} \|\nabla f(x)\|^2$ by merely applying the chain rule of differentiation. Similar arguments work for the proximal-PŁ condition.

**Convergence.** Then, *alternating gradient descent ascent* on $\min_{x \in \mathcal{X}} \max_{y \in \mathcal{Y}} f(x, y)$,

$$x_{t+1} \leftarrow \operatorname{Proj}_{\mathcal{X}} [x_t - \eta_x \nabla_x f(x_t, y_t)]; \quad y_{t+1} \leftarrow \operatorname{Proj}_{\mathcal{Y}} [y_t + \eta_y \nabla_y f(x_{t+1}, y_t)],$$

is proven to converge to a saddle-point point using a typical Lyapunov function argument. We tune the stepsizes $\eta_x, \eta_y$ in such a way that one player learns faster than the other. Since the function is PŁ, this means that after each update the optimizer is significantly approximated. Intuitively, after enough iterations, the update scheme can be viewed as optimizing for $\Phi(x) := \max_{y \in \mathcal{Y}} f(x, y)$ as $x_{t+1} \approx \operatorname{Proj}_{\mathcal{X}} (x_t - \eta_x \nabla_x \Phi(x_t))$. Crucially, our convergence analysis sets aside the usual regret minimization arguments that are used to either prove *average-iterate* or *best-iterate* convergence (*e.g.,* Anagnostides et al. (2022); Liu et al. (2024)).

## 1.3 Comparison to Related Work

We point out two particular results (Sokota et al., 2022; Liu et al., 2024) directly related to our endeavor of policy gradient/optimization methods for imperfect-information EFGs. Although the magnetic mirror descent method proposed in (Sokota et al., 2022) does not come with guarantees in EFGs, it exhibits impressive empirical performance. (Liu et al., 2024) lays the foundation of our approach as it introduces the *bidilated regularizer* although it does not offer a convergence guarantee that is polynomial in the parameters of the game and $1/\epsilon$.

Our work follows arguments utilized in the context of policy gradient methods for Markov decision processes (MDPs) and MGs. Namely, we use techniques from (Kalogiannis et al., 2025) that analyzed alternating gradient descent in the constrained parameter case and arguments from (Mei et al., 2020; Cen et al., 2022a) as the entropic bidilated regularizer is almost identical to discounted entropy. Further, we use arguments from (Zhang et al., 2021) to show that the mapping from sequence-form strategies to policies is Lipschitz continuous.

| | Altern./Simult. Updates | Provable Convergence | Regularization | Feedback |
|---|---|---|---|---|
| (Liu et al., 2024) | simultaneous | yes, best-iterate[*] | bidilated | CFR, $Q, \overline{Q}$ |
| (Sokota et al., 2022) | simultaneous | no | policy entropy | $Q$ |
| Ours | alternating | yes, last-iterate, polynomial time | bidilated | $\nabla_\theta V, Q$ |

Table 1: Comparison of policy gradient/optimization methods.

CFR, $Q, \overline{Q}, \nabla_\theta V$ stand for counterfactual value, action-value, traject. action-value, and policy gradient.
[*] Guarantees are pseudo-polynomial in the game-size.

## 2 Preliminaries

In this section we introduce the key ingredients required for our analysis. For IIEFGs, we highlight how the utility is expressed as a concave function of the sequence-form strategies. We also review the—Euclidean or entropic—bidilated regularizer whose strong convexity underpins our gradient-domination arguments. With regards to RL theory, we recall the definition of the value and and action-value functions and show that trajectory samples, or *roll-outs*, give unbiased Monte-Carlo estimates of both the utility and the bidilated regularizer via the (REINFORCE) estimator (Williams, 1992; Sutton et al., 1999). Finally, we review the optimization notions of hidden concavity and gradient dominance, used to prove convergence in of our algorithmic solutions.

### 2.1 Imperfect-Information Extensive-Form Games

We briefly go over the definition of an IIEFG and move on to the sequence-form strategies and the corresponding regularizers.

**Definition 1** (IIEFG). *A two player zero-sum extensive-form game, $\Gamma$, is defined by the tuple $(\mathcal{T}, \mathcal{H}, \mathcal{S}, \mathcal{A}, \mathcal{B}, r)$. A special* chance player, $c$, *models uncontrollable randomness while,*

- $\mathcal{T}$ *is a rooted game tree of height $D(\mathcal{T})$,*

- $\mathcal{H} := \mathcal{H}_1 \cup \mathcal{H}_2 \cup \mathcal{H}_c$ *is the set of $\mathcal{T}$'s nodes, referred to as* histories. *Each history, $h$, belongs to exactly one of the sets $\mathcal{H}_1, \mathcal{H}_2, \mathcal{H}_c$ depending on the player responsible for taking action at $h$.*

- $\mathcal{S} := \mathcal{S}_1 \cup \mathcal{S}_2$ *is a finite set of information sets (*infosets*). The infosets partition histories, $\mathcal{H}_i$, of the acting player $i$ into sets of nodes that are indistinguishable. We will note $S := \max\{|\mathcal{S}_1|, |\mathcal{S}_2|\}$.*

- $\mathcal{A} := \{\mathcal{A}_s\}_{s \in \mathcal{S}_1}, \mathcal{B} := \{\mathcal{B}_s\}_{s \in \mathcal{S}_2}$ *are the action sets of player 1 and 2, respectively. Each infoset $s \in \mathcal{S}$ has a corresponding set of actions $\mathcal{A}_s$, and respectively $\mathcal{B}_s$. Further, we will denote $A_s := |\mathcal{A}_s|, A := \max_s A_s$ and $B_s := |\mathcal{B}_s|, B_{\max} := \max_s B_s$.*

- $r : \mathcal{H} \to [0, 1]$ *is a payoff function mapping leaves of $\mathcal{T}$ to a payoff for player 1; player 2 gets the opposite payoff.*

A *perfect recall* assumption is made, ensuring that players remember their past observations and actions. This implies that nodes in the same infoset have the same past observation sequence. We will use $\sigma_1(s), \sigma_2(s)$ to denote the last parent infoset-action pair $(s', a'), s \in \mathcal{S}_1$ and $(s', b'), s \in \mathcal{S}_2$ encountered when descending from the game tree's root to history $h$. $\sigma_1(\cdot), \sigma_2(\cdot)$ are either unique for non-root nodes or the null set for the root. We will overload notation $\sigma_1(h)$ to mean $\sigma_1(s)$ for the infoset $s$ where $h$ belongs (resp. for $\sigma_2(h)$).

**Sequence-Form Strategies** A player's behavioral strategy is a probability distribution over actions at each of their infosets. With $\Sigma_1$ we denote player 1's subsequences of play starting at the root. In *sequence-form*, the strategy of player 1, $\mu_1^{\pi_1} \in \mathbb{R}^{|\Sigma_1|}$, with $|\Sigma_1| := 1 + \sum_s A_s$ is defined as:

$$\mu_1^{\pi_1}(s, a) := \mu_1^{\pi_1}(\sigma_1(s))\pi_1(a|s), \forall s \in \mathcal{S}_1, \forall a \in \mathcal{A}_s; \quad \mu_1^{\pi_1}(\varnothing) = 1.$$

The sequence-form strategy and $\Sigma_2$ of player 2 is defined in a symmetric fashion. Introduced in (Romanovskii, 1962; Von Stengel, 1996; Koller et al., 1996), sequence-form strategies are generalizations of simplices and express the sequential structure of an IIEFG. The set of sequence-form strategies,

$\mathcal{M}_1, \mathcal{M}_2$ are convex polytopes as they are is defined only by linear equalities and non-negativity constraints. The chance player's contribution to the probability of reaching history $h$ is given by $\mu_c(h)$ and it is assumed to be strictly positive for reachable nodes. For player 1, the expected utility is given by the bilinear form:

$$V^{\pi_1, \pi_2} := (\mu_1^{\pi_1})^\top \mathbf{R} \mu_2^{\pi_2},$$

where $\mathbf{R}$ is the matrix representation of payoff function $r$. Forward, we will refer to behavioral strategies as policies which will be denoted as $\pi_1, \pi_2$. The solution concept we are after is an $\epsilon$-approximate Nash equilibrium.

**Definition 2** ($\epsilon$-NE). *A policy profile $\pi_1^\star, \pi_2^\star$ is an $\epsilon$-approximate Nash equilibrium of an IIEFG $\Gamma$, if, for any policies $\pi_1$ and $\pi_2$ it holds true that,*

$$V^{\pi_1^\star, \pi_2} - \epsilon \leq V^{\pi_1^\star, \pi_2^\star} \leq V^{\pi_1, \pi_2^\star} + \epsilon.$$

**The bidilated regularizer.** Introduced in (Liu et al., 2024), the unweighted *bidilated regularizer* is defined using a strongly-convex regularizer $\psi(\cdot)$ multiplied by the total probability of reaching the corresponding infoset. Since it depends on both players' policies we write $\mathcal{R}(\pi_1, \pi_2), \mathcal{R}(\pi_2, \pi_2)$, s.t.:

$$\mathcal{R}_1(\pi_1, \pi_2) := \mathbb{E}_{h \sim \pi}\left[\sum_h \psi(\pi_1(\cdot|h))\right] \quad \text{and} \quad \mathcal{R}_2(\pi_1, \pi_2) := \mathbb{E}_{h \sim \pi}\left[\sum_h \psi(\pi_2(\cdot|h))\right].$$

## 2.2 RL Fundamentals

Moving on, we define the value, action-value, and advantage functions in the context of IIEFGs. Inspired by the occupancy measure of MGs, we define the history occupancy measure $d^\pi$ for a given policy profile $\pi := (\pi_1, \pi_2)$ which simply is the reach probability of each history and comes in handy as a shorthand notation in the description of the algorithms and their analysis. Moreover, we recall the definitions of direct and softmax policy parametrization. Last but not least, we demonstrate how the (REINFORCE) gradient estimator computes policy gradients for IIEFGs for both the unregularized and regularized utility.

**Value, action-value, and advantage functions.** Without loss of generality, we assume that players get a payoff only on a terminal history $\overline{h}$. This way we can define the *value function* of an infoset $s$, as the expected utility if the game were to start at a history $h_0$ belonging to $s$,

$$V^\pi(s) := \mathbb{E}_{\overline{h} \sim \pi}\left[r(\overline{h})|h_0 \in s\right].$$

In a similar vein, we define the action-value function, or $Q$, as the expected utility if the game started at at a history $h_0$ belonging in $s$ and after the player had taken action $a_0$, (or, resp. $b_0$),

$$Q_1^\pi(s, a) := \mathbb{E}_{\overline{h} \sim \pi}\left[-r(\overline{h})|h_0 \in s, a_0 = a\right] \quad \text{and} \quad Q_2^\pi(s, b) := \mathbb{E}_{\overline{h} \sim \pi}\left[r(\overline{h})|h_0 \in s, b_0 = b\right].$$

Finally, the advantage function is defined for each player as the difference between an action-value and the infoset's value $A_1^\pi(s, a) := -V^\pi(s) - Q_1^\pi(s, a)$ and $A_2^\pi(s, b) := V^\pi(s) - Q_1^\pi(s, b)$. Similar to the state occupancy measure of an MG, we can define the history occupancy measure $d^\pi : \mathcal{H} \to [0, 1]$ which is defined as, $d^\pi(h) := \mathbb{E}_{h' \sim \pi}[\mathbb{1}\{h' = h\}]$. Overloading notation, for an infoset $s \in \mathcal{S}$ $d^\pi(s) := \sum_{h \in s} d^\pi(h)$.

**Policies.** Policies are precisely parametrized behavioral strategies. We will consider two parametrizations of policies, (i) *direct parametrization*, and (ii) *softmax parametrization*. For directly parametrized policies, we denote the parameters as $x, y$ which are $x \in \bigtimes_{s \in \mathcal{S}_1} \Delta(\mathcal{A}_s), y \in \bigtimes_{s \in \mathcal{S}_2} \Delta(\mathcal{B}_s)$. The parameters of softmax policies will be denoted $\chi, \theta$ with $\chi \in \mathbb{R}^A, A = \sum_s A_s$ and $\theta \in \mathbb{R}^B, B = \sum_s B_s$.

**Gradient estimation with** REINFORCE**.** The ability to estimate a gradient of the value function using trajectory samples, or *roll-outs*, has endowed the theory and practice of RL with the rich toolbox of gradient-based optimization. In fact, the (REINFORCE) gradient estimator (Williams, 1992; Sutton et al., 1999) is also an unbiased estimator of the policy gradient in the IIEFG setting, and thus provides a sound foundation for our analysis.

**Definition 3** (REINFORCE). *Let $\xi$ denote a trajectory of infoset and actions sampled by implementing policies $\pi_1, \pi_2$, $\xi := (s_{(1)}, a_{(k)}, \dots)$. We define REINFORCE, $\left(\widehat{\nabla}_x, \widehat{\nabla}_y\right)$, to be the stochastic gradient estimators:*

$$\widehat{\nabla}_x = r_\xi \sum_{k=1}^{K_\xi} \nabla_x \log \pi_x\big(a_{(k)}|s_{(k)}\big) \quad and \quad \widehat{\nabla}_y = r_\xi \sum_{k=1}^{K_\xi} \nabla_x \log \pi_y\big(b_{(k)}|s_{(k)}\big). \quad \text{(REINFORCE)}$$

The addition of regularization, leads to the definition a regularized value function, $V_\tau$,

$$V_\tau^\pi(s) := \mathbb{E}_{\xi \sim \pi}\left[\sum_k r(h_{(k)}) + \tau \sum_h \left[\psi(\pi_1(\cdot|h_{(k)})) + \psi(\pi_2(\cdot|h_{(k)}))\right] \big| h_0 \in s\right].$$

The regularized $Q$-value and advantage functions, $Q_\tau^\pi, A_\tau^\pi$, are defined accordingly (see Appendix B.2). Furthermore, (REINFORCE) can be minimally modified to estimate the policy gradient of the regularized value function without importance sampling (discussed in detail in Appendix F.1).

**Assumption 1.** *For an $\varepsilon > 0$, both players' policies, for every infoset and action, satisfy*

$$\pi_1(a|s) \geq \varepsilon, \ \forall s \in \mathcal{S}_1, \forall a \in \mathcal{A}_s \quad \pi_2(b|s) \geq \varepsilon, \ \forall s \in \mathcal{S}_2, \forall b \in \mathcal{B}_s. \quad (\varepsilon\text{-trunc.})$$

Guaranteeing that ($\varepsilon$-trunc.) holds is straightforward for directly parametrized policies. The players need to pick policies $x, y$, from the cartesian product of appropriately truncated simplices, to be denoted $\mathcal{X}^\varepsilon, \mathcal{Y}^\varepsilon$ respectively. As for softmax parametrized policies, ($\varepsilon$-trunc.) is achieved when both players' parameters are restricted to the polytopes $X_R, \Theta_R$. To demonstrate, $X_R$ is defined in the following manner, $X_R := \left\{\chi \in \mathbb{R}^A, A = \sum_s A_s : \chi_s^\top \mathbf{1} = 0, \ \forall s \in \mathcal{S}_1, |\chi_{s,i} - \chi_{s,j}| \leq 2R, \ \forall i, j \in [A_s]\right\}$, and the definition of $\Theta_R$ follows suit. We highlight that the images of $X_R, \Psi_R$ under the softmax map are convex sets (Lemma D.5) and we will denote the resulting truncated policy sets as $\Pi_1^R, \Pi_2^R$.

## 2.3 Hidden Concavity and Gradient Domination

In this subsection, we define the two key backbone concepts of hidden concavity and gradient domination. Gradient domination of a weak or strong form has been extensively investigated in the theory of RL and MARL (Bhandari and Russo, 2024; Agarwal et al., 2021; Mei et al., 2020; Zhang et al., 2019; Daskalakis et al., 2020). Simply put, the nonconvex value function satisfies a gradient-domination property and any stationary point is globally optimal. Thus, any guarantee of convergence to a stationary point is elevated to a guarantee of convergence to global optimality.

**Definition 4** (Hidden convexity). *A nonconvex function $f : \mathcal{X} \to \mathbb{R}$ defined over the set $\mathcal{X}$ is said to be hidden (strongly) convex if there exists (i) a bijective mapping $c : \mathcal{X} \to \mathcal{U}$ for some convex set $\mathcal{U}$; (ii) a function $H : \mathcal{U} \to \mathbb{R}$ that is strongly convex with modulus $\alpha_H \geq 0$; such that $f(x) = H(c(x)), \forall x \in \mathcal{X}$.*

When the Lipschitz continuity modulus of the inverse transform, $c^{-1}$, is uniformly bounded it implies the gradient domination condition as shown in (Fatkhullin et al., 2023, Prop. 2) coupled with (Karimi et al., 2016, App. G).

**Definition 5** (pPŁ condition (Karimi et al., 2016)). *Assume $F : \mathbb{R}^d \to \mathbb{R}$ defined as $F(x) := f(x) + g(x)$. Let $f : \mathbb{R}^d \to \mathbb{R}$ be an $\ell$-smooth function and $g : \mathbb{R}^d \to \mathbb{R}$ be convex. Define*

$$\mathcal{D}_g(x, \ell) := -2\ell \min_z \left\{\langle \nabla f(x), z - x \rangle + \frac{\ell}{2} B(z\|x) + g(z) - g(x)\right\}.$$

*for a choice of Bregman divergence $B(\cdot\|\cdot)$. We say that $F$ satisfies the pPŁ condition with modulus $\alpha > 0$ if, for every $x$,*

$$\frac{1}{2}\mathcal{D}_g(x, \ell) \geq \alpha\left[F(x) - F^\star\right],$$

*where $F^\star = \min_x F(x)$. When $g$ is the indicator function of a set $\mathcal{X}$ we write $\mathcal{D}_\mathcal{X}(x, \ell)$.*

## 3  Main Results

With the latter in hand, we are ready to state our main contributions, (i) the independent *exploration strategy*, (ii) the *gradient domination condition* for utilities of EFGs (iii) and the *global convergence of three variants of policy gradient methods* to an approximate Nash equilibrium.

## 3.1 Efficient Exploration Scheme

We propose a novel approach to exploration. Each player is expected to reach every subsequence with probability at least $\frac{\gamma}{|\mathcal{H}|}$. The rule is simple:

**Assumption 2** (Efficient Exploration). *Both players follow the following exploration strategy:*

- *At the start of each game, the player flips a biased coin that shows "heads" with probability $\gamma$.*
- *If the coin shows "heads", the player selects a sequence uniformly at random and then executes it.*
- *After this sequence, or if the coin shows "tails", the player resumes play according to their policy.*

**Remark 1.** *It is noteworthy that using this exploration strategy, one can exercise direct control over the modulus of gradient domination. Whereas, policy gradient literature (Agarwal et al., 2021; Daskalakis et al., 2020; Mei et al., 2020; Zeng et al., 2022) needs to make an assumption on the boundedness of the distribution mismatch coefficient.*

## 3.2 Gradient Domination Property of the Utilities

In this subsection, we establish that the utility of an imperfect-information EFG under different policy parametrizations is pPŁ with regards to the policy. This observation is central in proving convergence of policy gradient methods to a Nash equilibrium. First, we state the weak gradient domination property for the unregularized utilities of the game.

**Lemma 3.1** (Utility Weak Gradient Domination). *Let $\Gamma$ be an imperfect-information EFG, following Assumption 2, then it holds true that*

$$V^{\pi_1,\pi_2} - \min_{\pi_1'} V^{\pi_1',\pi_2} \leq \frac{1}{2\alpha} \max_{\pi_1'} \langle \nabla_{\pi_1} V^{\pi_1,\pi_2}, \pi_1 - \pi_1' \rangle;$$

$$\max_{\pi_2'} V^{\pi_1,\pi_2'} - V^{\pi_1,\pi_2} \leq \frac{1}{2\alpha} \max_{\pi_2'} \langle \nabla_{\pi_2} V^{\pi_1,\pi_2}, \pi_2' - \pi_2 \rangle,$$

*for an $\alpha > 0$ with $\alpha^{-1} = \mathsf{poly}\left(\frac{1}{\gamma}, |\mathcal{H}|, S, A, B\right)$.*

Now, by picking an appropriate regularization term to each player's utility we can enhance the weak gradient domination property to the much stronger pPŁ condition which ultimately guarantees last-iterate convergence to an equilibrium of the regularized game.

**Lemma 3.2** (Utility pPŁ; restated from Lemmata E.1 to E.3). *Let an imperfect-information EFG, $\Gamma$, perturbed by a pair of weighted bidilated regularizers $(\mathcal{R}_1, \mathcal{R}_2)$ with a coefficient $\tau > 0$. Also, assume that each player follows Assumption 1 and Assumption 2. Then, each player's utility satisfies the pPŁ condition with a modulus $\alpha^{-1} = \frac{1}{\tau} \times \mathsf{poly}\left(\frac{1}{\varepsilon}, \frac{1}{\gamma}, \frac{1}{\min_h \mu_c(h)}, |\mathcal{H}|, S, A, B, 2^{D(\mathcal{T})}\right).$*

A key observation in both conditions is that the modulus is a polynomial of the exploration parameter $1/\gamma$. This stresses the importance of efficient exploration and our corresponding contribution of the scheme in Assumption 2. Also,

## 3.3 Convergence of Alternating Regularized Policy Gradient

Having established the required background and notation, we are ready to present our main results. In Theorem 3.1 we show the convergence of simple alternating regularized policy gradient to an approximate NE in the last iterate. Moving to Theorem 3.2, we prove a similar result for softmax-parametrized policies. Finally, we analyze *alternating regularized natural policy gradient* through a mirror-descent lens, demonstrate its relationship to multiplicative weight updates of the policies, and prove its convergence to an approximate NE in the last iterate (Theorem 3.3).

Throughout, $\eta_x, \eta_y$ denote the stepsizes and $\hat{\nabla}^\tau$ denotes the (REINFORCE) gradient estimate of the utility w.r.t. to a player's parameters accounting only for their own regularization term.

### 3.3.1 Direct Policy Parametrization

The first result we present is the a simple policy gradient scheme with alternating updates and a Euclidean regularizer. The parameter updates of alternating regularized policy gradient takes the

following form,

$$x_{t+1} = \underset{\mathcal{X}^\varepsilon}{\text{Proj}} \left[ x_t - \eta_x \hat{\nabla}_x^\tau(x_t, y_t) \right]$$

$$y_{t+1} = \underset{\mathcal{Y}^\varepsilon}{\text{Proj}} \left[ y_t + \eta_y \hat{\nabla}_y^\tau(x_{t+1}, y_t) \right].$$
(Alt-RegPG)

where $\text{Proj}_{\mathcal{X}^\varepsilon}, \text{Proj}_{\mathcal{Y}^\varepsilon}$ denote the Euclidean projection of the parameters to the truncated simplices dictated by ($\varepsilon$-trunc.). We state our first convergence theorem which settles question (♥) and defer its formal statement to the Appendix H.1.

**Theorem 3.1** (Informal; restated from Thm. H.1). *With direct policy parametrization and the Euclidean bidilated regularizer, alternating policy-gradient algorithm attains a last-iterate $\epsilon$-Nash equilibrium in*

$$T = \text{poly}\left( \tfrac{1}{\epsilon}, \tfrac{1}{\varepsilon}, \tfrac{1}{\gamma}, |\mathcal{H}|, |\mathcal{S}_1|, |\mathcal{S}_2|, A, B, 2^{D(\mathcal{T})} \right) \text{ iterations,}$$

*using batches of* $\text{poly}\left( \tfrac{1}{\epsilon}, \tfrac{1}{\varepsilon}, \tfrac{1}{\gamma}, |\mathcal{H}|, |\mathcal{S}_1|, |\mathcal{S}_2|, A, B, 2^{D(\mathcal{T})} \right)$ *trajectory samples at each step.*

**Remark 2.** *We note that the exponential dependence on $D(\mathcal{T})$ is still polynomial in the game size as the height has itself logarithmic dependence in size of the game.*

### 3.3.2 Softmax Policy Parametrization

We move on to convergence under softmax parametrization and entropic regularization. This choice of parametrization is an important step towards getting provable guarantees for policy gradient methods in imperfect-information EFGs using function approximation (*e.g.* neural networks). The projection to $X_R, \Theta_R$ guarantees that ($\varepsilon$-trunc.) is satisfied,

$$\chi_{t+1} = \underset{X_R}{\text{Proj}} \left[ \chi_t - \eta_x \hat{\nabla}_\chi^\tau(\chi_t, \theta_t) \right];$$

$$\theta_{t+1} = \underset{\Theta_R}{\text{Proj}} \left[ \theta_t + \eta_y \hat{\nabla}_\theta^\tau(\chi_{t+1}, \theta_t) \right].$$
(Alt-EntRegPG)

**Theorem 3.2** (Informal; restated from Thm. H.2). *Alternating policy-gradient algorithm with softmax policy parametrization and the entropic bidilated regularizer, converges in expectation in the last-iterate to an $\epsilon$-Nash equilibrium after a number of iterations $T$, that is*

$$T = \text{poly}\left( \tfrac{1}{\epsilon}, \tfrac{1}{\varepsilon}, \tfrac{1}{\gamma}, |\mathcal{H}|, |\mathcal{S}_1|, |\mathcal{S}_2|, A, B, 2^{D(\mathcal{T})} \right) \text{ iterations,}$$

*using batches of* $\text{poly}\left( \tfrac{1}{\epsilon}, \tfrac{1}{\varepsilon}, \tfrac{1}{\gamma}, |\mathcal{H}|, |\mathcal{S}_1|, |\mathcal{S}_2|, A, B, 2^{D(\mathcal{T})} \right)$ *trajectory samples at each step.*

### 3.3.3 Natural Policy Gradient

Finally, we consider the natural policy gradient algorithm (Kakade, 2001) which is an adaptation of natural gradient (Amari, 1998). This algorithm is of particular interest due to its intimate connection to the TRPO, PPO (Schulman et al., 2015, 2017) policy optimization algorithms. Natural policy gradient uses a *Fisher information matrix* induced by the policy as a preconditioner for policy gradient updates:

$$\mathbf{F}_\chi(\chi, \theta) := \sum_s d^{\chi,\theta}(s) \sum_a \pi_\chi(a|s) \nabla \log \pi_\chi(a|s) \left[ \nabla \log \pi_\chi(a|s) \right]^\top$$

We cast *natural policy gradient* steps as *mirror descent steps* with a Mahalanobis norm induced by the Fisher information matrix (for a more nuanced discussion on this connection see (Raskutti and Mukherjee, 2015)).The update scheme can be equivalently written as:

$$\chi_{t+1} = \underset{\chi \in X_R}{\arg\min} \left\| \chi_t - \eta_x \mathbf{F}_\chi^\dagger(\chi_t, \theta_t) \nabla_\chi V(\chi_t, \theta_t) - \chi \right\|_{\mathbf{F}_\chi(\chi_t, \theta_t)}^2$$

$$\theta_{t+1} = \underset{\theta \in \theta_R}{\arg\min} \left\| \theta_t + \eta_y \mathbf{F}_\theta^\dagger(\chi_{t+1}, \theta_t) \nabla_\theta V(\chi_{t+1}, \theta_t) - \theta \right\|_{\mathbf{F}_\theta(\chi_{t+1}, \theta_t)}^2$$
(Alt-RegNPG)

More importantly, we note that in policy space, the update scheme of natural policy gradient takes a very simple form which, as expected, reads, for player 1 ($\odot$ is element-wise multiplication):

$$\overline{\pi}_{1,t+1}(\cdot|s) \propto \pi_{1,t}(\cdot|s)^{1-\eta_x\tau} \odot \exp\left(\eta_x Q_\tau^{\pi_t}(s,\cdot)\right);$$

$$\pi_{1,t+1}(\cdot|s) \approx \underset{\pi \in \Pi_1^R}{\arg\min} \, \mathrm{KL}\left(\pi(\cdot|s)\big\|\overline{\pi}_{1,t+1}(\cdot|s)\right).$$

To see why the second approximate equality holds, we note that the Mahalanobis distance over the parameters induced by the Fisher information matrix of the softmax policy, is a second-order approximation of policy KL divergence. The derivation and an extensive discussion are deferred to Appendices H.3 and I.

**Theorem 3.3** (Informal; restated from Thm. H.3). *For an appropriate tuning of $\eta_x, \eta_y > 0$, the last-iterate of alternating regularized natural policy gradient (`Alt-RegNPG`) converges in expectation to an $\epsilon$-approximate Nash equilibrium in a number of iterations $T$ that is:*

$$T = \mathsf{poly}\left(\tfrac{1}{\epsilon}, \tfrac{1}{\varepsilon}, \tfrac{1}{\gamma}, |\mathcal{H}|, |\mathcal{S}_1|, |\mathcal{S}_2|, A, B, 2^{D(\mathcal{T})}\right).$$

## 4 Empirical Validation

To corroborate our theoretical results, we tested `Alt-RegNPG` on four different imperfect information EFGs (Kuhn Poker, Leduc Poker, $2 \times 2$ Abrupt Dark Hex and Liar's Dice). Inspired by MMD (Sokota et al., 2022), we implement two variants of `Alt-RegNPG` where the (i) the regularization strength diminishes across time along the stepsizes and (ii) the regularizer is the discounted KL divergence from a moving reference policy. We observe that the exploitability (*i.e.* $\max_{\pi_1'} V^{\pi_1',\pi_2} - \min_{\pi_2'} V^{\pi_1,\pi_2'}$) diminishes across time for our method, and it compares well with CFR and MMD.

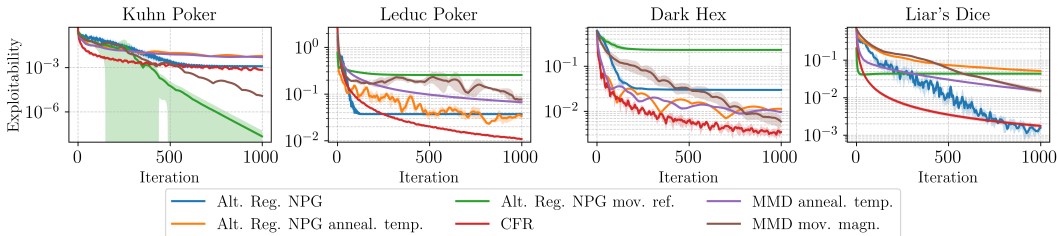

Figure 1: Three variants of `Alt-RegNPG` compared against CFR and MMD.

## 5 Discussion

We conclude our main text with a further comparison between MGs and imperfect information IIEFGs to further promote the connection between the two areas. Finally, we state our conclusions and suggestions for future work.

### 5.1 Further comparison of Markov and Imperfect Information Extensive-Form Games

Imperfect-information IIEFGs and MGs both model multi-stage strategic interaction. They differ sharply in what each player can observe while they maintain marked similarities in the way strategies are represented (*behavioral strategies* and *policies*), the *hidden concave* representation of utilities (concavity w.r.t. *sequence-form strategies* and *occupancy measures*), and regularization choices for optimization. The table and discussion below summarize this comparison along the axes of observability, strategy space, utility convex reformulation, regularization and optimization landscape. Clearly, an *infoset* (information set) in an imperfect-information EFG is to a behavioral strategy what a state is to a policy in an MG. However, imperfect information (or *partial observability*) leads to a discrepancy between the expected return of an infoset in an EFG and the expected return state

| | Game State | Observable State | Control Variables | Utility Concave In |
|---|---|---|---|---|
| **IIEFG** | History $h \in \mathcal{T}$ | Infoset $s \in \mathcal{S}$ | Behavioral Strategy $\pi(\cdot|s)$ | Sequence-form Strategy $\mu^\pi$ |
| | *each a node of game tree graph $\mathcal{T}$* | *each a disjoint set of multiple histories $h$* | *distribution over actions at infoset $s$* | *independent of opponents' strategies* |
| **MG** | State $s$ | | Markovian Policy $\pi(\cdot|s)$ | State-action Occupancy measure $\lambda^\pi$ |
| | *fully observable by all players potentially recurring in the finite or infinite horizon of the game* | | *distribution over actions at state $s$* | *depends on opponents' policies* |

Table 2: Imperfect-information extensive-form games (IIEFG) vs. Markov games (MG).

in an MG as highlighted in (Nayyar et al., 2013; Sokota et al., 2023). Interestingly, the concave reparametrization of EFG utilities exhibits a structure more favorable than the corresponding one in MGs. In particular, the utility is concave in sequence-form strategies of IIEFGs and the latter depend solely on a player's own behavioral strategy. This comes in stark contrast to the state-action occupancy measure of MGs which are conditioned on opponents' strategies.

Finally, similarities of the regularization techniques in IIEFGs and MGs are cornerstone to our work. The EFG entropic *bidilated regularizer* (Liu et al., 2024), $\mathcal{R}$, and the very commonly used MDP discounted entropy (Williams and Peng, 1991; Haarnoja et al., 2018; Mei et al., 2020; Cen et al., 2022a,b), $\mathcal{E}$, are virtually identical. We note that, in IIEFGs a regularizer is mostly used in context of directly optimizing in the sequence-form space. They induce a distance generating function of mirror descent instantiations. Some more recent works have used it to make the game strongly-monotone and guarantee convergence of gradient descent methods (Liu et al., 2022b). Liu et al. (2024), in the context of policy optimization, define the bidilated regularizer whose policy gradients can be estimated without importance sampling. Illustratively, the two regualaizers read side-by-side ($\gamma$ is a discount factor of MDPs):

$$\mathcal{R}(\pi) := \mathbb{E}_{\xi \sim \pi}\left[\sum_{s_{(k)} \in \xi} \psi(\pi(\cdot|s_{(k)}))\right] \qquad \Big| \qquad \mathcal{E}(\pi) := \mathbb{E}_{\xi \sim \pi}\left[\sum_k^H \gamma^{k-1}\psi(\pi(\cdot|s_{(k)}))\right].$$

## 5.2 Conclusion

We studied three different policy gradient methods for imperfect-information perfect-recall zero-sum IIEFGs under a unifying optimization principle. We managed to provide the first global last-iterate convergence guarantees of policy gradient methods to an $\epsilon$-approximate Nash equilibrium. Furthermore, our analysis requires a number of iterations and samples that is polynomial in $1/\epsilon$ and the parameters of the game. To do so, we demonstrated that utilities as functions of behavioral strategies (policies) exhibit gradient domination properties even though they are nonconvex; and provided a practical decentralized exploration scheme that implicitly controls the moduli of gradient domination. We departed from the usual route of regret analysis in IIEFGs and opted for more conventional convergence analysis arguments using a Lyapunov function. We hope to motivate further exchange between theoretical MARL research and the theory of IIEFGs as we strongly believe in the potential this communication fosters.

**Future directions.** Our main objective was proving polynomial time convergence of policy gradient in IIEFGs, our analysis is at places loose. We firmly believe that the convergence rates and constant dependencies can be improved, *e.g.*, by using the machinery of treeplex norms (Fan et al., 2024), relatively-smooth optimization (Lu et al., 2018; Fatkhullin and He, 2024), and other policy optimization arguments (Zhan et al., 2023; Cen et al., 2022b). To be particular, we would like to see guarantees that do not call for mini-batching and possibly use variance reduction techniques. Moreover, fundamental questions about the limit points of policy gradient methods in IIEFGs (similar to those of (Giannou et al., 2022) for MGs) are open. More broadly, do forms of benign nonconvexity (like hidden convexity) refine the results of (Cai et al., 2024b; Angelopoulos et al., 2025)?

## Acknowledgments

This work was supported in part by the NSF AI Institute for Learning-Enabled Optimization at Scale (TILOS, CCF-2112665), NSF Award CCF-244306, and the Office of Naval Research (ONR grants N000142412631 and N00014-25-1-2296). GF is supported in part by an AI2050 Early Career Fellowship.

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

# Appendix

# A  Further Related Work

In this section we attempt discussing related work. Arguably, since our work lies in the intersection of several already broad themes, we encourage the reader to follow references in the cited works.

**Relevant MARL for MG works**   In MDP and MG literature, policy optimization seems to come in two flavors—an *online learning* (Hazan et al., 2016; Lattimore and Szepesvári, 2020) approach and a *stochastic optimization* one. In the current work, we opt for the second approach.

The approach of (Zeng et al., 2022) which considers zero-sum Markov games is particularly similar to ours. Yet, we highlight that they make a rather strong assumption; they assume that the probability of playing each action in the support of the regularized Nash equilibrium is lower-bounded by a constant independent of the regularization coefficient $\tau$. In turn, we contribute the two-sided pPŁ condition for IIEFGs and, importantly, circumvent such an assumption by exercising direct control over the minimum probability of playing any action by projecting the parameters of the softmax parameters onto a convex polytope.

**Theory of Policy Gradient Methods**   The policy gradient method was introduced for Markov decision processes in (Williams, 1992; Sutton et al., 1999). Ever since provable guarantees have been yielded by a number of works for different variations of the algorithm:

- (Agarwal et al., 2021) prove the convergence of directly parametrized policy gradient. They use the convergence result of gradient descent for smooth nonconvex function along a gradient domination lemma to demonstrate a $O(1/\epsilon^2)$ convergence rate to optimality. Later, (Zhang et al., 2020, 2021) use the *hidden concave* structure of the problem to improve the convergence rate to $O(1/\epsilon)$.

- (Mei et al., 2020) provide the first non-asymptotic convergence rate result for the policy gradient method using discounted entropy regularization (the analogue of bidilated entropy regularization). The proof of convergence uses a novel nonuniform PŁ condition.

- (Cen et al., 2022a) analyze natural policy gradient (NPG) with discounted entropy regularization. Natural policy gradient can be seen as a form of *preconditioned* gradient descent. Natural policy gradient effectively boils down to policy multiplicative weight updates using the $Q$-functions as feedback. The analysis of convergence uses a linear dynamical system.

**Regularized Markov Decision Processes**   Regularization in RL seems to have a very broad development. It was theoretically analyzed by (Haarnoja et al., 2018; Nachum et al., 2017; Geist et al., 2019). Regularization helps with both the optimization landscape (Mei et al., 2020) as well as learning policies from offline data (Neu et al., 2017).

**RL & Regularization in IIEFGs**   Applying RL in IIEFGs, in the sense of using policy gradients and action-value functions is not a new endeavor. It has been extensively studied from both theoretical and practical viewpoints (Munos et al., 2020; Sokota et al., 2022; Rudolph et al., 2025). Yet, a provable convergence guarantee for policy gradient methods like ours was missing. Furthermore, using regularization has also been investigated in (Perolat et al., 2021; Liu et al., 2022b, 2024) to get favorable convergence guarantees to equilibria, to guarantee uniqueness of equilibria and continuity of best-response maps Sokota et al. (2023).

**Markov Games**   MGs have been extensively studied through the lens of policy gradient and policy optimization methods. For the zero-sum setting there have been numerous algorithmic approaches using multiple techniques (Brafman and Tennenholtz, 2002; Perolat et al., 2015; Alacaoglu et al., 2022a; Wei et al., 2021; Zhang et al., 2022).

## B   Further Preliminaries on IIEFGs

### B.1   The Behavioral and Sequence-Form Strategies

In this subsection, we investigate the continuity of the sequence-form map and that of its inverse.

**Lemma B.1.** *Under Assumption 2, the transforms $c_1^{-1} : \mathcal{M}_1 \to \mathcal{X}_\gamma, c_2^{-1} : \mathcal{M}_2 \to \mathcal{Y}_\gamma$ are Lipschitz continuous. I.e., for any $\mu_1, \mu_1'$, it holds true that,*

$$\left\| c_1^{-1}(\mu_1) - c_1^{-1}(\mu_1') \right\| \le \frac{2|\mathcal{H}|\sqrt{A}}{\gamma} \left\| \mu_1 - \mu_1' \right\|$$

*and for any $\mu_2, \mu_2'$,*

$$\left\| c_2^{-1}(\mu_2) - c_2^{-1}(\mu_2') \right\| \le \frac{2|\mathcal{H}|\sqrt{B}}{\gamma} \left\| \mu_2 - \mu_2' \right\|.$$

*Proof.* We will first observe the difference in $c_1^{-1}$ in the $(s,a)$-th entry of the the vector-valued mapping:

$$
\begin{aligned}
\frac{\mu_1(s,a)}{\mu_1(s)} - \frac{\mu_1'(s,a)}{\mu_1'(s)} &= \left( \frac{\mu_1(s,a)}{\mu_1(s)} - \frac{\mu_1'(s,a)}{\mu_1(s)} \right) + \left( \frac{\mu_1'(s,a)}{\mu_1(s)} - \frac{\mu_1'(s,a)}{\mu_1'(s)} \right) \\
&= \left( \frac{\mu_1(s,a)}{\mu_1(s)} - \frac{\mu_1'(s,a)}{\mu_1(s)} \right) + \left( \frac{1}{\mu_1(s)} - \frac{1}{\mu_1'(s)} \right) \mu_1'(s,a) \\
&= \left( \frac{\mu_1(s,a)}{\mu_1(s)} - \frac{\mu_1'(s,a)}{\mu_1(s)} \right) + \frac{\mu_1'(s) - \mu_1(s)}{\mu_1(s)\mu_1'(s)} \mu_1'(s,a)
\end{aligned}
$$

As a reminder, for all $s \in \mathcal{S}_1$ it holds that $\mu_1(s) \ge \frac{\gamma}{|\mathcal{H}|}$ by Assumption 2. Proceeding towards the desired inequality,

$$
\begin{aligned}
&\left\| c_1^{-1}(\mu_1) - c_1^{-1}(\mu_1') \right\|^2 \\
&= \sum_{s \in \mathcal{S}_1} \sum_{a \in \mathcal{A}_s} \left[ \left( \frac{\mu_1(s,a)}{\mu_1(s)} - \frac{\mu_1'(s,a)}{\mu_1(s)} \right) + \frac{\mu_1'(s) - \mu_1(s)}{\mu_1(s)\mu_1'(s)} \mu_1'(s,a) \right]^2 \\
&\le 2 \sum_{s \in \mathcal{S}_1} \sum_{a \in \mathcal{A}_s} \left( \frac{\mu_1(s,a)}{\mu_1(s)} - \frac{\mu_1'(s,a)}{\mu_1(s)} \right)^2 + 2 \sum_{s \in \mathcal{S}_1} \sum_{a \in \mathcal{A}_s} \left( \frac{\mu_1'(s) - \mu_1(s)}{\mu_1(s)\mu_1'(s)} \right)^2 {\mu_1'}^2(s,a) \\
&\le 2 \sum_{s \in \mathcal{S}_1} \sum_{a \in \mathcal{A}_s} \left( \frac{\mu_1(s,a)}{\mu_1(s)} - \frac{\mu_1'(s,a)}{\mu_1(s)} \right)^2 + 2 \sum_{s \in \mathcal{S}_1} \sum_{a \in \mathcal{A}_s} \left( \frac{\mu_1'(s) - \mu_1(s)}{\mu_1(s)\mu_1'(s)} \right)^2 {\mu_1'}^2(s) \\
&\le \frac{2|\mathcal{H}|^2}{\gamma^2} \sum_{s \in \mathcal{S}_1} \sum_{a \in \mathcal{A}_s} (\mu_1(s,a) - \mu_1'(s,a))^2 + \frac{2|\mathcal{H}|^2}{\gamma^2} \sum_{s \in \mathcal{S}_1} \sum_{a \in \mathcal{A}_s} (\mu_1'(s) - \mu_1(s))^2 \\
&\le \frac{2|\mathcal{H}|^2}{\gamma^2} \left\| \mu_1 - \mu_1' \right\|^2 + \frac{2A|\mathcal{H}|^2}{\gamma^2} \sum_{s \in \mathcal{S}_1} (\mu_1'(s) - \mu_1(s))^2 \\
&= \frac{2|\mathcal{H}|^2}{\gamma^2} \left\| \mu_1 - \mu_1' \right\|^2 + \frac{2A|\mathcal{H}|^2}{\gamma^2} \sum_{s \in \mathcal{S}_1} \left( \sum_{a \in \mathcal{A}_s} \mu_1'(s,a) - \mu_1(s,a) \right)^2. \quad (1)
\end{aligned}
$$

We need to upper bound the second term by some quantity proportional to $\|\mu_1 - \mu_1'\|$. We first note that by the triangular inequality,

$$
\begin{aligned}
\left| \sum_{a \in \mathcal{A}_s} \mu_1'(a|s) - \mu_1(a|s) \right| &\le \sum_{a \in \mathcal{A}_s} |\mu_1'(a|s) - \mu_1(a|s)| \\
&\le \sqrt{A} \left\| \mu_1'(\cdot|s) - \mu_1(\cdot|s) \right\|.
\end{aligned}
$$

where the last inequality is due to the fact that $\|x\|_1 \leq \sqrt{d}\|x\|, \forall x \in \mathbb{R}^d$. As such, we can note that,

$$\sum_{s \in \mathcal{S}_1} \left( \sum_{a \in \mathcal{A}_s} \mu_1'(a|s) - \mu_1(a|s) \right)^2 \leq \sum_{s \in \mathcal{S}_1} \left( \sqrt{A}\|\mu_1'(\cdot|s) - \mu_1(\cdot|s)\| \right)^2$$

$$= A \sum_{s \in \mathcal{S}_1} \sum_{a \in \mathcal{A}_s} (\mu_1'(s,a) - \mu_1(s,a))^2$$

$$= A \|\mu_1' - \mu_1\|^2.$$

Plugging this inequality into (1) yields the desired bound. □

**Lemma B.2.** *The sequence-form strategy $\mu_1 = c_1(\pi_1)$ is a $(\sqrt{|\Sigma_1|D(\mathcal{T})})$-Lipschitz and $(\sqrt{|\Sigma_1|D(\mathcal{T})})$-smooth function of the behavioral strategy $\pi_1$. That is,*

$$\|c_1(\pi_1) - c_1(\pi_1')\|_2 \leq \sqrt{|\Sigma_1|D(\mathcal{T})} \|\pi_1 - \pi_1'\|_2,$$

$$\|\mathbf{J}_{c_1}(\pi_1) - \mathbf{J}_{c_1}(\pi_1')\|_{\mathrm{op}} \leq \sqrt{|\Sigma_1|D(\mathcal{T})} \|\pi_1 - \pi_1'\|_2,$$

*for any $\pi_1, \pi_1'$, where $\mathbf{J}_{c_1}(\cdot)$ denotes the Jacobian of the sequence-form map.*

*Proof.* For the continuity of $\mu_1$ we observe that each entry of the Jacobian, $\mathbf{J}_{c_1}(\pi_1)$, is in $[0,1]$ as a product of variables in $[0,1]$. Further, the number of non-zero elements of each row of $\mathbf{J}_{c_1}$ is bounded by the height of the tree, $D(\mathcal{T})$. We can then write,

$$\max_{\pi_1} \|\mathbf{J}_{c_1}(\pi_1)\|_{\mathrm{op}}^2 \leq \max_{\pi_1} \|\mathbf{J}_{c_1}(\pi_1)\|_{\mathrm{F}}^2 \leq |\Sigma_1|D(\mathcal{T}).$$

Now, for the continuity of the Jacobian, $\mathbf{J}_{c_1}(\cdot)$, we make some observations on the Hessian tensor. In particular, for the matrix corresponding to a single entry of $\mu_1$, with index $i$, it is the case that all entries are in $[0,1]$ and are at most $D(\mathcal{T})^2$ in number. Then, we consider $\|\nabla^2 c(\pi_1)\|_{\mathrm{op}} :=$ $\sup_{\|u\|_2 = \|v\|_2 = 1} \sqrt{\sum_i \left( \sum_j \sum_k [\nabla^2 c(\pi_1)]_{ijk} u_j v_k \right)^2}$ where $j, k$ index entries of $\pi_1$. In this case, by bounding each $\left( \sum_j \sum_k [\nabla^2 c(\pi_1)]_{ijk} u_j v_k \right)$ by an upper bound on its Frobenius norm, we conclude that,

$$\|\nabla^2 c(\pi_1)\|_{\mathrm{op}}^2 \leq |\Sigma_1|D(\mathcal{T})^2.$$

□

**Lemma B.3.** *The sequence-form strategy $\mu_1 = c_1(\pi_\chi)$ is a $(\sqrt{|\Sigma_1|D(\mathcal{T})})$-Lipschitz and $(\sqrt{|\Sigma_1|D(\mathcal{T})})$-smooth function of the parameters of softmax policy $\pi_\chi$, $\chi$. That is,*

$$\|c_1(\pi_\chi) - c_1(\pi_{\chi'})\|_2 \leq \tfrac{1}{2} \sqrt{|\Sigma_1|D(\mathcal{T})} \|\chi - \chi'\|_2,$$

$$\|\mathbf{J}_{c_1}(\pi_\chi) - \mathbf{J}_{c_1}(\pi_{\chi'})\|_{\mathrm{op}} \leq 16\sqrt{|\Sigma_1|D(\mathcal{T})} \|\chi - \chi'\|_2,$$

*for any $\chi, \chi'$.*

*Proof.* We know that the softmax map is $\frac{1}{2}$-Lipschitz continuous and it has a 8-Lipschitz Jacobian Lemma D.2. Treating $c_1(\pi_\chi)$ as a composition of the sequence-form map and the softmax map, we can conclude that,

$$\|c_1(\pi_\chi) - c_1(\pi_{\chi'})\|_2 \leq \frac{\sqrt{\Sigma_1 D(\mathcal{T})}}{2} \|\chi - \chi'\|,$$

and

$$\|\mathbf{J}_{c_1}(\pi_\chi) - \mathbf{J}_{c_1}(\pi_{\chi'})\|_{\mathrm{op}} \leq \left( \sqrt{|\Sigma_1|}D(\mathcal{T}) \left(\tfrac{1}{2}\right)^2 + \left( \sqrt{|\Sigma_1|D(\mathcal{T})} \right) 8 \right) \|\chi - \chi'\|_2,$$

$$\leq 16\sqrt{|\Sigma_1|D(\mathcal{T})} \|\chi - \chi'\|_2.$$

□

## B.2 Value, Action-Value, and Advantage Functions

**On notation.** In this subsection, we will use the following shorthand notations,

- $\sigma_1(h), \sigma_2(h)$ returns the last history before $h$ where player 1 (player 2, resp.) took an action,
- $h \in s$ signifies that history $h$ belongs in the infoset $s$,
- $h' \succeq_{\mathcal{T}} h, h' \succeq_{\mathcal{T}} (h,a)$ signifies that $h'$ is a successor/child node of $h, (h,a)$;
- $h \in \xi, (h,a) \in \xi$ signifies that $h, h, a$ belongs in the game trajectory $\xi$ from the root to a terminal node.

**Occupancy measure** For a policy pair $\pi := (\pi_1, \pi_2)$, we define $d^\pi : \mathcal{S} \to [0,1]$ to be a finite measure over all the infosets—summing over all infosets $s \in \mathcal{S}$ yields the depth of the game tree $D(\mathcal{T})$—where for any infoset $s \in \mathcal{S}$,

$$d^\pi(s) := \sum_{h \in s} \mu_c(h) \mu_1^{\pi_1}(\sigma_1(h)) \mu_2^{\pi_2}(\sigma_2(h)).$$

The value function of each infoset is defined as,

$$
\begin{aligned}
V_1^\pi(s) :=& \mathbb{E}_{\xi \sim \pi} \left[ \sum_{h' \in \xi} r_1(h') \mathbb{1}\{h' \succeq_{\mathcal{T}} s\} \,\Big|\, \exists h \in s : h \in \xi \right] \\
=& \frac{1}{\sum_{h \in s} \mu_c(h) \mu_1^{\pi_1}(\sigma_1(h)) \mu_2^{\pi_2}(\sigma_2(h))} \sum_{h' : \exists h \in s, h' \succeq_{\mathcal{T}} h} \mu_c(h') \mu_1^{\pi_1}(\sigma_1(h')) \mu_2^{\pi_2}(\sigma_2(h')) r_1(h').
\end{aligned}
$$

Also, the action-value function reads:

$$
\begin{aligned}
Q_1^\pi(s,a) :=& \mathbb{E}_{\xi \sim \pi} \left[ \sum_{h' \in \xi, h' \succeq_{\mathcal{T}}(h,a)} r(h') \,\Big|\, \exists h \in s : (h,a) \in \xi \right] \\
=& \frac{1}{\sum_\xi \mathbb{P}^\pi(\xi) \mathbb{1}\{\exists h \in s : (h,a) \in \xi\}} \sum_\xi \mathbb{P}^\pi(\xi) \mathbb{1}\{\exists h \in s : (h,a) \in \xi\} \left[ \sum_{\substack{h' \in \xi, \\ h' \succeq_{\mathcal{T}}(h,a)}} r(h') \right].
\end{aligned}
$$

We define the advantage function to be:

$$A_1^\pi(s,a) := Q_1^\pi(s,a) - V_1^\pi(s).$$

Finally, let a policy pair $\pi_1, \pi_2$ and $\pi := (\pi_1, \pi_2)$. Let $\pi_1$ be parametrized by some vector $\theta$. We compute the policy gradient for $\theta$,

$$
\begin{aligned}
\frac{\partial V_1^\pi}{\partial \theta_{s,a}} &= \frac{\partial}{\partial \theta_{s,a}} \sum_\xi r_1(\xi) \, \mathbb{P}^\pi(\xi) \\
&= \sum_\xi r_1(\xi) \, \mathbb{P}^\pi(\xi) \frac{\partial \log \mathbb{P}^\pi(\xi)}{\partial \theta_{s,a}} \\
&= \sum_\xi \sum_{a'} r_1(\xi) \, \mathbb{P}^\pi(\xi) \frac{\partial \log \pi_1(a'|s)}{\partial \theta_{s,a}} \mathbb{1}\{\exists h \in s : (h,a') \in \xi\} \\
&= \sum_\xi \sum_{a'} \left( r_1(\xi) \, \mathbb{P}^\pi(\xi) \frac{\mathbb{1}\{\exists h \in s : (h,a') \in \xi\}}{\pi_1(a'|s)} \right) \pi_1(a'|s) \frac{\partial \log \pi_1(a'|s)}{\partial \theta_{s,a}} \\
&= \sum_\xi \sum_{a'} \left( \left[ \sum_{\substack{h' \in \xi, \\ h' \succeq_{\mathcal{T}}(h,a)}} r(h') + \sum_{\substack{h' \in \xi, \\ h' \prec_{\mathcal{T}}(h,a)}} r(h') \right] \mathbb{P}^\pi(\xi) \frac{\mathbb{1}\{\exists h \in s : (h,a') \in \xi\}}{\pi_1(a'|s)} \right) \pi_1(a'|s) \frac{\partial \log \pi_1(a'|s)}{\partial \theta_{s,a}}
\end{aligned}
$$

$$= \sum_{\xi} \sum_{a'} \left( \left[ \sum_{\substack{h' \in \xi, \\ h' \prec_{\mathcal{T}}(h,a)}} r(h') \right] \mathbb{P}^{\pi}(\xi) \frac{\mathbb{1}\{\exists h \in s : (h, a') \in \xi\}}{\pi_1(a'|s)} \right) \pi_1(a'|s) \frac{\partial \log \pi_1(a'|s)}{\partial \theta_{s,a}}$$

$$+ d^{\pi}(s) \sum_{a'} \pi_1(a'|s) \frac{\partial \log \pi_1(a'|s)}{\partial \theta_{s,a}} Q^{\pi}(s, a')$$

$$= d^{\pi}(s) \sum_{a'} \pi_1(a'|s) \frac{\partial \log \pi_1(a'|s)}{\partial \theta_{s,a}} Q^{\pi}(s, a'). \tag{2}$$

Where we have used the following fact,

$$\sum_{\xi} \sum_{a'} \left( \left[ \sum_{\substack{h' \in \xi, \\ h' \prec_{\mathcal{T}}(h,a)}} r(h') \right] \frac{\mathbb{1}\{\exists h \in s : (h, a') \in \xi\}}{\pi_1(a'|s)} \right) \pi_1(a'|s) \frac{\partial \log \pi_1(a'|s)}{\partial \theta_{s,a}}$$

$$= \sum_{a'} \underbrace{\sum_{\xi} \left( \left[ \sum_{\substack{h' \in \xi, \\ h' \prec_{\mathcal{T}}(h,a)}} r(h') \right] \frac{\mathbb{1}\{\exists h \in s : (h, a') \in \xi\}}{\pi_1(a'|s)} \right)}_{=:C(s)} \pi_1(a'|s) \frac{\partial \log \pi_1(a'|s)}{\partial \theta_{s,a}}$$

$$= \sum_{a'} C(s) \pi_1(a'|s) \frac{\partial \log \pi_1(a'|s)}{\partial \theta_{s,a}}$$

$$= C(s) \sum_{a'} \pi_1(a'|s) \frac{\partial \log \pi_1(a'|s)}{\partial \theta_{s,a}}$$

$$= C(s) \frac{\partial}{\partial \theta_{s,a}} \sum_{a'} \pi_1(a'|s)$$

$$= C(s) \frac{\partial}{\partial \theta_{s,a}} 1 = 0.$$

Further, for direct policy parametrization, we get,

$$\frac{\partial V_1^{\pi}}{\partial \pi_1(s, a)} = d^{\pi}(s) Q^{\pi}(s, a).$$

For the softmax policy parametrization, (2) yields,

$$\frac{\partial V_1^{\pi}}{\partial \theta_{s,a}} = d^{\pi}(s) \sum_{a'} \pi_1(a'|s) \frac{\partial \log \pi_1(a'|s)}{\partial \theta_{s,a}} Q^{\pi}(s, a')$$

$$= d^{\pi}(s) \sum_{a'} \pi_1(a'|s) \left[ \mathbb{1}\{a' = a\} - \pi_1(a'|s) \right] Q^{\pi}(s, a')$$

$$= d^{\pi}(s) \pi_1(a|s) \left[ Q^{\pi}(s, a) - V^{\pi}(s) \right]$$

$$= d^{\pi}(s) \pi_1(a|s) A^{\pi}(s, a).$$

### B.3 Continuity of the Utility

We briefly consider the Lipschitz continuity of the utility w.r.t. direct and softmax policy parametrizations.

**Lemma B.4.** *The utility of an IIEFG function as a function of direct-parametrized policies is* $(\max_{i \in \{1,2\}} \sqrt{|\Sigma_i|} D(\mathcal{T}))$*-smooth.*

*Proof.* Let $u := \mathbf{R}\mu_2^{\pi_2}$. It is a vector in $\mathbb{R}^{|\Sigma_1|}$ with entries in $[-1, 1]$. As such,

$$V^{\pi_1, \pi_2} = \langle \mu_1^{\pi_1}, u \rangle,$$

from which we write,

$$\left\| \nabla_{\pi_1} V^{\pi_1, \pi_2} - \nabla_{\pi_1} V^{\pi_1', \pi_2} \right\| = \left\| \nabla_{\pi_1} \langle \mu_1^{\pi_1}, u \rangle - \nabla_{\pi_1} \langle \mu_1^{\pi_1'} u \rangle \right\|$$
$$\leq \| u \| \sqrt{|\Sigma_1|} D(\mathcal{T}) \| \pi_1 - \pi_1' \|$$
$$\leq |\Sigma_1| D(\mathcal{T}) \| \pi_1 - \pi_1' \| .$$

Where, we used Lemma B.2 in the first inequality. $\square$

**Lemma B.5.** *The utility function as a function of softmax-parametrized policies is* $16(\max_{i \in \{1,2\}} \sqrt{|\Sigma_i|} D(\mathcal{T}))$*-smooth.*

*Proof.* We treat the utility function as a composition of the utility as a function of the policy and the softmax map (*i.e.*, Lemma B.4 along with Lemma D.2). $\square$

## B.4 Properties of the Bidilated Regularizer

Introduced in (Liu et al., 2024), the bidilated regularizer offers an alternative to the commonly used dilated regularizer (Hoda et al., 2010). It can be seamlessly used along $Q$ feedback by dropping the need of importance sampling which would be necessary for the *dilated regularizer* when the gradient is estimated through trajectory roll-outs. The purpose of this refined regularizer was introducing a distance generating function in the sequence-form space that would not necessitate importance sampling.

### B.4.1 Strong Convexity Modulus

**Lemma B.6.** *For a choice of strongly convex function* $\psi$*, and a weighting scheme* $\{w_{1,s}\}_{s \in \mathcal{S}_1}$*,* $\{w_{2,s}\}_{s \in \mathcal{S}_2}$ *and let* $\alpha_{\mathrm{dil}} > 0$ *be the modulus of the weighted dilated regularizer. Then, the corresponding bidiliated regularizer is strongly convex,*

$$\alpha_{\mathrm{bi}} := \frac{\gamma}{|\mathcal{H}|} \min_h \mu_c(h).$$

*Proof.* These calculations were used in the proof of (Liu et al., 2024, Lemma D.1); we repeat them for completeness. For an appropriate choice of weights $\{w_{1,s}\}_{s \in \mathcal{S}_1}$, $\{w_{2,s}\}_{s \in \mathcal{S}_2}$, the *weighted* bidilated regularizer is defined as,

$$\mathcal{R}_1^{\psi}(\mu_1^{\pi_1}, \mu_2^{\pi_2}) := \sum_s \mu_1^{\pi_1}(\sigma_1(s)) \left( \sum_{h \in s} \mu_c(h) \mu_2^{\pi_2}(\sigma_2(h)) \right) w_{1,s} \psi(\pi_1(\cdot|s))$$

$$\mathcal{R}_2^{\psi}(\mu_1^{\pi_1}, \mu_2^{\pi_2}) := \sum_s \mu_2^{\pi_2}(\sigma_2(s)) \left( \sum_{h \in s} \mu_c(h) \mu_2^{\pi_2}(\sigma_1(h)) \right) w_{2,s} \psi(\pi_2(\cdot|s)).$$

We can slightly refine (Liu et al., 2024, Lemma C.1) in order to compute an explicit lower bound on the convexity modulus of different weighted bidilated regularizer depending on the choice of $\psi$. From the fact that $\mathcal{R}_1(\mu_1^{\pi_1}, \mu_2^{\pi_2})$ is linear in $\mu_2^{\pi_2}$ and the definition of the Bregman divergence, we conclude that,

$$\left\langle \nabla(\mathcal{R}_1 + \mathcal{R}_2)(\mu_1^{\pi_1}, \mu_2^{\pi_2}) - \nabla(\mathcal{R}_1 + \mathcal{R}_2)(\mu_1^{\pi_1'}, \mu_2^{\pi_2'}), (\mu_1^{\pi_1}, \mu_2^{\pi_2}) - (\mu_1^{\pi_1'}, \mu_2^{\pi_2'}) \right\rangle$$
$$\geq B_{\mathcal{R}_1^{\psi}} \left( \mu_1^{\pi_1'} \| \mu_1^{\pi_1}; \mu_2^{\pi_2} \right) + B_{\mathcal{R}_1^{\psi}} \left( \mu_1^{\pi_1} \| \mu_1^{\pi_1'}; \mu_2^{\pi_2} \right) + B_{\mathcal{R}_2^{\psi}} \left( \mu_2^{\pi_2} \| \mu_2^{\pi_2'}; \mu_1^{\pi_1} \right) + B_{\mathcal{R}_2^{\psi}} \left( \mu_2^{\pi_2'} \| \mu_2^{\pi_2}; \mu_1^{\pi_1'} \right).$$

By (Liu et al., 2022c, Lemma D.2) we know that,

$$B_{\mathcal{R}_1^{\psi}} \left( \mu_1^{\pi_1'} \| \mu_1^{\pi_1}; \mu_2^{\pi_2} \right) \geq \frac{\gamma}{|\mathcal{H}|} \min_h \mu_c(h) B_{\psi}^{\mathrm{dil}}(\mu_1^{\pi_1'} \| \mu_1^{\pi_1}).$$

As such, for the strong convexity modulus of the weighted $\mathcal{R}_1^{\psi}$ relative to the choice of norm appropriate for $\psi$, we write,

$$\alpha_{\mathrm{bi}} := \frac{\gamma}{|\mathcal{H}|} \min_h \mu_c(h) \, \alpha_{\mathrm{dil}}.$$

$\square$

By (Farina et al., 2019, Corollary 1), we know that there exists a weighting scheme, such that the Euclidean dilated regularizer is 1-strongly convex w.r.t. the $\ell_2$-norm. The procedure assigns weights to nodes in a bottom-up fashion.

- At each leaf node $s$, the weights are set to
$$w_{1,s} = 1.$$

- For an internal node $s$, let $s_a, s_{a'}, \ldots$ denote its child nodes under actions $a, a', \ldots$. For each action $a$, compute
$$W_{1,s_a} = \sum_{s' \succeq_{\mathcal{T}}(s,a)} w_{1,s'}.$$

- The node's weights are then set to
$$w_{1,s} = 2 \max_a W_{1,s_a}.$$

**Corollary B.1** (Euclidean Regularizer). *There exists a choice of weights, with* $\max_s w_{1,s}, \max_s w_{2,s} = \Theta(2^{D(\mathcal{T})})$, *and under the assumption that* $\min_s \mu_2(s) \geq \gamma$, *the bidilated Euclidean regularizer has a strong convexity modulus w.r.t. the* $\ell_2$-norm, $\alpha_{\mathrm{bi}}$,
$$\alpha_{\mathrm{bi}}^{\mathrm{eucl}} := \frac{\gamma}{|\mathcal{H}|} \min_h \mu_c(h).$$

(Kroer et al., 2020, Theorem 2) states that a recursion defines weights with $\max_s w_{1,s}, \max_s w_{2,s} = \Theta(2^{D(\mathcal{T})})$ such that the entropic dilated regularizer is strongly convex w.r.t. the $\ell_2$-norm.

**Corollary B.2** (Entropic Regularizer). *There exists a choice of weights, and under the assumption that* $\min_s \mu_2(s) \geq \gamma$, *the bidilated entropic regularizer has a strong convexity modulus w.r.t. the* $\ell_2$-norm, $\alpha_{\mathrm{bi}}$,
$$\alpha_{\mathrm{bi}}^{\mathrm{ent}} := \frac{\gamma}{|\mathcal{H}|} \min_h \mu_c(h).$$

### B.4.2 Lipschitz Moduli

Here, we establish the Lipschitz continuity of the regularizers and that of their gradients.

**Euclidean regularizer**

**Lemma B.7.** *The weighted Euclidean bidilated regularizer is* $\ell$-smooth with
$$\ell := \Theta\left(2^{D(\mathcal{T})} \max_{i \in \{1,2\}} |\Sigma_i| D(\mathcal{T}) S\right).$$

*Proof.* We write the bidilated regularizer as
$$\mathcal{R}_1^{\mathrm{eucl}}(\pi_1, \pi_2) := \langle f(\pi_1, \pi_2), g(\pi_1) \rangle.$$
For a fixed $\pi_2$, we have
$$\nabla_{\pi_1} \mathcal{R}_1^{\mathrm{eucl}}(\pi_1, \pi_2) = \mathbf{J}_f(\pi_1, \pi_2)^\top g(\pi_1) + \mathbf{J}_g(\pi_1)^\top f(\pi_1, \pi_2),$$
where, $f(\pi_1, \pi_2), g(\pi_1) \in \mathbb{R}^{|\mathcal{H}|}$ with $f(\pi_1, \pi_2) = \sum_{h \in s} \mu_c(h) \mu_2^{\pi_2}(\sigma_2(h)) \mu_1^{\pi_1}(\sigma_1(h))$ and $g_s(\pi_1) = w_{1,s} \|\pi_1(\cdot|s)\|^2$. We write:

$$\left\| \nabla_{\pi_1} \mathcal{R}_1^{\mathrm{eucl}}(\pi_1, \pi_2) - \nabla_{\pi_1} \mathcal{R}_1^{\mathrm{eucl}}(\pi_1', \pi_2) \right\|$$
$$\leq \|(\mathbf{J}_f(\pi_1) - \mathbf{J}_f(\pi_1'))\| \|g(\pi_1')\| + \|\mathbf{J}_f(\pi_1')\| \|g(\pi_1) - g(\pi_1')\|$$
$$+ \|\mathbf{J}_g(\pi_1) - \mathbf{J}_g(\pi_1')\| \|f(\pi_1)\| + \|\mathbf{J}_g(\pi_1')\| \|f(\pi_1) - f(\pi_1')\|$$
$$\leq \left( \ell_f \max_{\pi_1'} \|g(\pi_1')\| + 2 L_f L_g + \ell_g \max_{\pi_1} \|f(\pi_1)\| \right) \|\pi_1 - \pi_1'\|$$
$$\leq \left( \ell_f \sqrt{S} + 2 L_f L_g + \ell_g \sqrt{S} \right) \|\pi_1 - \pi_1'\|$$

- For $g$, we see that $L_g := \sqrt{S}\max_s w_{1,s}$ and $\ell_g := 2\sqrt{S}\max_s w_{1,s}$ by the properties of the weighted $\ell_2$-norm and the fact that $\pi_1(\cdot|s)$ lies in the simplex, *i.e.*, $\|\pi_1(\cdot|s)\|_2 \le 1$. Also, the weight $w_{1,s}$ only scales the local quadratic term.

- For $f$, similar to Lemma B.2 and Lemma B.4, $L_f \le \max_{i\in\{1,2\}}|\Sigma_i|\sqrt{D(\mathcal{T})S}$ and $\ell_f \le \max_{i\in\{1,2\}}|\Sigma_i|D(\mathcal{T})\sqrt{S}$. Also, it holds that $\max_{\pi_1,\pi_2}\|f(\pi_1)\| \le \sqrt{S}$.

Concluding,

$$\left\|\nabla_{\pi_1}\mathcal{R}_1^{\mathrm{eucl}}(\pi_1,\pi_2) - \nabla_{\pi_1}\mathcal{R}_1^{\mathrm{eucl}}(\pi_1',\pi_2)\right\| \le 64\max_s w_{1,s}\max_{i\in\{1,2\}}|\Sigma_i|D(\mathcal{T})\sqrt{S}\,\|\pi_1 - \pi_1'\|.$$

Symmetrically,

$$\left\|\nabla_{\pi_2}\mathcal{R}_2^{\mathrm{eucl}}(\pi_1,\pi_2) - \nabla_{\pi_2}\mathcal{R}_2^{\mathrm{eucl}}(\pi_1,\pi_2')\right\| \le 64\max_s w_{2,s}\max_{i\in\{1,2\}}|\Sigma_i|D(\mathcal{T})\sqrt{S}\,\|\pi_2 - \pi_2'\|.$$

Now, we need to bound the Lipschitz modulus of $\nabla_{\pi_1}\mathcal{R}_2^{\mathrm{eucl}}(\pi_1,\pi_2)$. Similarly, we write,

$$\mathcal{R}_2^{\mathrm{eucl}}(\pi_1,\pi_2) := \langle f(\pi_1,\pi_2), g(\pi_2)\rangle.$$

We see that the the vector $f(\pi_1,\pi_2)$ (occupancy measure of player 2) has entries that are products of entries of $\mu_1, \mu_2, \mu_c$. Hence, $L_f = \max_{i\in\{1,2\}}|\Sigma_i|\sqrt{D(\mathcal{T})S}$ and $\ell_f = \max_{i\in\{1,2\}}|\Sigma_i|D(\mathcal{T})\sqrt{S}$.

$$\begin{aligned}
\left\|\nabla_{\pi_1}\mathcal{R}_2^{\mathrm{eucl}}(\pi_1,\pi_2) - \nabla_{\pi_1}\mathcal{R}_2^{\mathrm{eucl}}(\pi_1',\pi_2)\right\| &\le \|\mathbf{J}_f(\pi_1,\pi_2) - \mathbf{J}_f(\pi_1',\pi_2)\|\,\|g(\pi_2)\| \\
&\le \max_{i\in\{1,2\}}|\Sigma_i|D(\mathcal{T})\sqrt{S}\,\|\pi_1 - \pi_1'\|\,\|g(\pi_2)\| \\
&\le \max_s w_{2,s}\max_{i\in\{1,2\}}|\Sigma_i|D(\mathcal{T})S\,\|\pi_1 - \pi_1'\|.
\end{aligned}$$

$\square$

**Entropic regularizer**

**Lemma B.8.** *The weighted entropic bidilated regularizer is $\ell$-smooth with*

$$\ell := \Theta\left(2^{D(\mathcal{T})}\max_{i\in\{1,2\}}|\Sigma_i|D(\mathcal{T})S\log A\right).$$

*Proof.* We write $\mathcal{R}_2$ as the inner product of $f(\pi_\chi) := d^{\pi_\chi,\pi_\theta}$ and $g := [\pi_\theta(b|s)\log\pi_\theta(b|s)]_{s,b}$. For notational convenience, we suppress dependence of $f, g$ on $\pi_\theta$.

$$\mathcal{R}_2(\pi_\chi) := \langle f(\pi_\chi,\pi_\theta), g(\pi_\theta)\rangle.$$

We now bound the Lipschitz modulus of the gradient using the chain rule:

$$\begin{aligned}
\|\nabla_\chi\mathcal{R}_2(\pi_\chi,\pi_\theta) - \nabla_\chi\mathcal{R}_2(\pi_{\chi'},\pi_\theta)\| &\le \|\mathbf{J}_\pi(\chi)^\top\mathbf{J}_f(\pi_\chi) - \mathbf{J}_\pi(\chi')^\top\mathbf{J}_f(\pi_{\chi'})\|\,\|g(\pi_\theta)\| \\
&\le \left(\|\mathbf{J}_\pi(\chi)^\top\mathbf{J}_f(\pi_\chi) - \mathbf{J}_\pi(\chi)^\top\mathbf{J}_f(\pi_{\chi'})\| + \|\mathbf{J}_\pi(\chi)^\top\mathbf{J}_f(\pi_{\chi'}) - \mathbf{J}_\pi(\chi')^\top\mathbf{J}_f(\pi_{\chi'})\|\right)\|g(\pi_\theta)\| \\
&\le \left(\|\mathbf{J}_\pi(\chi)\|\,\|\mathbf{J}_f(\pi_\chi) - \mathbf{J}_f(\pi_{\chi'})\| + \|\mathbf{J}_f(\pi_{\chi'})\|\,\|\mathbf{J}_\pi(\chi) - \mathbf{J}_\pi(\chi')\|\right)\|g(\pi_\theta)\| \\
&\le \left(\left(\tfrac{1}{2}\right)^2\max_{i\in\{1,2\}}|\Sigma_i|D(\mathcal{T})\sqrt{S} + 8\max_{i\in\{1,2\}}|\Sigma_i|\sqrt{D(\mathcal{T})S}\right)\sqrt{S}\max_s w_{2,s}\,\|\chi - \chi'\|.
\end{aligned}$$

For the Lipschitz modulus of $\nabla_\chi\mathcal{R}_1(\pi_\chi,\pi_\theta)$, we re-purpose the lengthy calculations found in the proof of (Mei et al., 2020, Lemma 14), we consider $\chi = \chi_0 + \alpha u$ for some $u, \chi \in \mathbb{R}^A, \alpha \in \mathbb{R}$,

$$\left\|\frac{\mathrm{d}g(\chi + \alpha u)}{\mathrm{d}\alpha}\right\|_\infty \le \max_s w_{1,s}\log A\,\|u\|_2;$$

hence, (since $\|x\|_2 \le \sqrt{S_1}\|x\|_\infty$),

$$\left\|\frac{\mathrm{d}g(\chi + \alpha u)}{\mathrm{d}\alpha}\right\|_2 \le \max_s w_{1,s}\log A\sqrt{S}\,\|u\|_2,$$

or, $L_g = \max_s w_{1,s} \log A\sqrt{S}$. Similarly,

$$\left\| \frac{\mathrm{d}^2 g(\chi + \alpha u)}{\mathrm{d}\alpha^2} \right\|_\infty \le 3 \max_s w_{1,s} (1 + \log A) \|u\|_2 \, ;$$

and, as such,

$$\left\| \frac{\mathrm{d}^2 g(\chi + \alpha u)}{\mathrm{d}\alpha^2} \right\|_2 \le 3 \max_s w_{1,s} (1 + \log A)\sqrt{S} \|u\|_2 \, ,$$

or, $\ell_g = 3 \max_s w_{1,s} (1 + \log A)\sqrt{S}$. Hence, $\nabla_\chi \mathcal{R}_1$ is $\ell$-smooth with

$$\ell \le \max_s w_{1,s} \log A\sqrt{S} \left( \sqrt{SD(\mathcal{T})} \left(\tfrac{1}{2}\right)^2 + 8\sqrt{S} \max_{i \in \{1,2\}} \sqrt{|\Sigma_i| D(\mathcal{T})} \right) +$$

$$+ \sqrt{S} 3 \max_s w_{1,s} (1 + \log A)\sqrt{S} + 2 \max_{i \in \{1,2\}} |\Sigma_i| \sqrt{D(\mathcal{T})} S \tfrac{1}{2} \max_s w_{1,s} \log A\sqrt{S}$$

$$\le 242^{D(\mathcal{T})} \max_{i \in \{1,2\}} |\Sigma_i| \sqrt{D(\mathcal{T})} S \log A.$$

$\square$

## C  Efficient Exploration

Throughout our proofs, we have kept our complexity results parametric w.r.t. $1/\gamma$. A naive exploration rule that would dictate that the player merely picks behavioral strategies over the $\varepsilon$-truncated simplex will give a $\gamma = O(\varepsilon^{D(\mathcal{T})})$. We propose a different approach to exploration. In particular, every player is expected to reach every prefix subsequence with a probability $\frac{\gamma}{|\Sigma_i|}$ where $|\Sigma_i| := 1 + \sum_{s \in \mathcal{S}_1} |\mathcal{A}_s|$ denotes the set of all possible "prefix" sequences of player $i$. The rule is simple,

- at the beginning of each game, the player throws a biased coin which lands on "heads" with probability $\gamma$. If so happens, the player executes a sequence of actions with probability $\frac{1}{|\Sigma_i|}$. Afterwards, the player continues to play according to their own behavioral strategy.
- In the case that the coin lands on "tails", the player simply plays according to their behavioral strategy.

We observe that in sequence-form, this means that $\mu_1(\sigma(s)) \geq \frac{\gamma}{|\Sigma_1|} + \frac{\gamma}{|\Sigma_1|} \sum_{s' \in \Sigma_1} \mathbb{1}\{s' \succeq_{\mathcal{T}} s\}$ (in words, the amount of "probability flow" reaching the corresponding sequence $\sigma(s)$ for $s$, is at least as much as $\frac{\gamma}{|\Sigma_1|}$ plus the flow that passes through $\sigma(s)$ to visit its children). In other words, the sequence-form strategies are truncated by a set of linear constraints and as long as $\gamma \leq \frac{1}{|\Sigma_1|}$, there set of feasible sequence-form strategies is non-empty. We now observe that the mapping, from $\mu$ to the part component of the behavioral policy the agent can in fact control, is

$$\pi(a|s) = \frac{\mu(s,a) - \frac{\gamma}{|\Sigma_1|} \sum_{s' \in \Sigma_1} \mathbb{1}\{s' \succeq_{\mathcal{T}} (s,a)\}}{\mu(\sigma(s)) - \frac{\gamma}{|\Sigma_1|} \sum_{s' \in \Sigma_1} \mathbb{1}\{s' \succeq_{\mathcal{T}} s\}}.$$

The "probability flow" passing through the edge $(s,a)$ breaks down to a controllable part due to the policy $\pi(a|s)$ and an uncontrollable one due to the exploration scheme. In particular, the uncontrollable "probability flow" is precisely $\frac{\gamma}{|\Sigma_1|} \times \sum_{s' \in \Sigma_1} \mathbb{1}\{s' \succeq_{\mathcal{T}} (s,a)\}$—*i.e.*, proportional to the number of nodes of the subtree rooted at the next node after $(s,a)$ where player 1 acts. As such, the Lipschitz continuity of mapping $\mu \mapsto \pi$, is Lipschitz continuous with a modulus,

$$\frac{|\Sigma_1|\sqrt{A}}{\gamma},$$

by following the same line of arguments as the ones in Lemma B.1.

In short, we are only adding an additional linear constraint on the feasibility set of $\mu_1^{\pi_1}$ (and $\mu_2^{\pi_2}$, respectively). Granted that that this new feasibility set is always non-empty, this $\gamma$-truncated treeplex remains a convex polytope. Finally we note that for any player $i$, $|\Sigma_i| \leq |\mathcal{H}|$.

**Proposition 1.** Let $\Gamma$ be an $n$-player imperfect-information EFG $\Gamma$ with perfect recall. Also, assume that players follow the exploration scheme of Assumption 2. Then, an $\epsilon$-NE on the exploration-induced $\gamma$-truncated treeplices, is an $\left(\epsilon + 2[1 - (1-\gamma)^n]\right)$-NE of the original game.

*Proof.* Let $\pi^\star$ be a joint policy profile, $V_i^{\pi^\star}$ will be the utility of player under no exploration under joint policy $\pi^\star$ and $V_{\gamma,i}^{\pi^\star}$ the utility of player $i$ under the exploration scheme. When the exploration scheme is followed, there is still a probability $(1-\gamma)^n$ that no player follows it for a particular episode. Hence, for any $\pi^\star$,

$$\left| V_i^{\pi^\star} - V_{\gamma,i}^{\pi^\star} \right| \leq (1 - (1-\gamma)^n)(r_{i,\max} - r_{i,\min})$$
$$\leq (1 - (1-\gamma)^n),$$

where, $r_{i,\max}, r_{i,\min}$ signify the maximum and minimum value of payoff $r_i$ for player $i$. With the same line of reasoning, $\left| \max_{\pi_i'} V_i^{\pi_i', \pi_{-i}^\star} - \max_{\pi_i'} V_{\gamma,i}^{\pi_i', \pi_{-i}^\star} \right| \leq 1 - (1-\gamma)^n$. Now, assume $\{\pi_i^\star\}_{i \in [n]}$ to be an $\epsilon$-NE. Fixing a player $i$, we want to compute the difference in the optimality gap on the $\gamma$-truncated treeplex versus the entire treeplex. Now, by definition of the $\epsilon$-NE,

$$\max_{\pi_i'} V_{\gamma,i}^{\pi_i', \pi_{-i}^\star} - V_{\gamma,i}^{\pi^\star} \leq \epsilon \quad \Rightarrow \quad \max_{\pi_i'} V_i^{\pi_i', \pi_{-i}^\star} - V_i^{\pi^\star} \leq \epsilon + 2[1 - (1-\gamma)^n].$$

$\square$

When $n = 2$, $\epsilon + 2[1 - (1-\gamma)^2] = \epsilon + 4\gamma - 2\gamma^2 = O(\epsilon + \gamma)$.

# D  Regarding the Policy Parametrization

## D.1  Definitions

**Direct policy parametrization.**    Both players parameterize their policies (or behavioral strategies), $\pi_1 : \mathcal{S}_1 \to \mathcal{A}$ and $\pi_2 : \mathcal{S}_2 \to \mathcal{B}$, using a concatenation of $|\mathcal{S}_1|$ and $|\mathcal{S}_2|$ probability vectors over the (potentially truncated) probability simplex $\Delta(\mathcal{A}_s), \Delta(\mathcal{B}_s)$ for all $s$ in $\mathcal{S}_1$ and $\mathcal{S}_2$ respectively. The parameter space of player 1 is denoted by $\mathcal{X} := \prod_{s \in \mathcal{S}_1} \Delta(\mathcal{A}_s)$, while the parameter space of player 2 by $\mathcal{Y} := \prod_{s \in \mathcal{S}_2} \Delta(\mathcal{B}_s)$.

**Softmax policy parametrization.**    Softmax parametrized policies have a well-known definition. The parameters of the corresponding policies are denoted $\chi, \theta$ with $\chi \in \mathbb{R}^A, A = \sum_s A_s$ and $\theta \in \mathbb{R}^B, B = \sum_s B_s$. For each infoset $s$, the policy is

$$\pi_\chi(a|s) = \frac{\exp(\chi_{s,a})}{\sum_{a'} \exp(\chi_{s,a'})} \quad \text{or} \quad \pi_\theta(b|s) = \frac{\exp(\theta_{s,b})}{\sum_{b'} \exp(\theta_{s,b'})}.$$

Now, since we want to have control over the minimum eigenvalue of the Jacobian of $\mathrm{softmax}(\cdot)$, we restrict the parameter space to the following convex polytopes,

$$X_R := \left\{ \chi \in \mathbb{R}^A, A = \sum_s A_s : \chi_s^\top \mathbf{1} = 0, \ \forall s \in \mathcal{S}_1, |\chi_{s,i} - \chi_{s,j}| \leq 2R, \ \forall i, j \in [A_s] \right\};$$

$$\Theta_R := \left\{ \theta \in \mathbb{R}^B, B = \sum_s B_s : \theta_s^\top \mathbf{1} = 0, \forall s \in \mathcal{S}_2, |\theta_{s,i} - \theta_{s,j}| \leq 2R, \ \forall i, j \in [B_s] \right\}.$$

## D.2  Properties under Parameter Constraints

**Lemma D.1.** *Let $\mathbf{J} := \mathbf{J}_{\mathrm{softmax}}(\theta) \in \mathbb{R}^{d \times d}$ be the Jacobian of the softmax map. Its matrix form is:*

$$\mathbf{J} = \mathrm{diag}\left(\mathrm{softmax}(\theta)\right) - \mathrm{softmax}(\theta)\mathrm{softmax}(\theta)^\top.$$

*Further, the vector $\mathbf{1}$ is an eigenvector of $\mathbf{J}$ with a corresponding eigenvalue of $0$. The rest of the eigenvalues are*

$$\lambda_i \in \left[ \min_{i \in [d]} \mathrm{softmax}_i(\theta), \max_{i \in [d]} \mathrm{softmax}_i(\theta) \right].$$

*Proof.* For brevity, define $\sigma := \mathrm{softmax}(\theta)$, and let $\mathrm{diag}(v)$ be the $d \times d$ diagonal matrix "whose diagonal entries are given by $v \in \mathbb{R}^d$,

$$\mathbf{J} = \mathrm{diag}(\sigma) - \sigma\sigma^\top.$$

First, we observe that the all-ones vector $\mathbf{1} \in \mathbb{R}^d$ is an eigenvector of $\mathbf{J}$ with a corresponding eigenvalue of $0$,

$$\begin{aligned}
\mathbf{J} &= \mathrm{diag}(\sigma)\mathbf{1} - \sigma\sigma^\top\mathbf{1} \\
&= \sigma - \sigma(\sigma^\top)\mathbf{1} \\
&= \sigma - \sigma = 0.
\end{aligned}$$

By Weyl's inequality for two Hermitian matrices, $A, B$, we know that their eigenvalues indexed in a descending order $\lambda_1(A) \geq \cdots \geq \lambda_d(A)$ satisfy,

$$\lambda_{i+j-d}(A + B) \leq \lambda_i(A) + \lambda_j(B) \leq \lambda_{i+j-1}(A + B).$$

$\lambda_i(\mathrm{diag}(\sigma)) = \sigma_i^\downarrow$ while $\lambda_d(-\sigma\sigma^\top) = -\|\sigma\|_2^2 \in \left[-1, -\frac{1}{d}\right]$. Hence,

- $\lambda_{\min}^+(\mathbf{J}) \geq \min_{i \in [d]} \sigma_i(\theta)$ — by taking $i = d$ and $j = d - 1$;

- $\sigma_2^\downarrow \leq \lambda_{\max}(\mathbf{J}) \leq \max_{i \in [d]} \sigma_i(\theta)$ — by taking $i = 2, j = 1$ for the LHS and $i = 1, j = 1$ for the RHS.

$\square$

**Lemma D.2** ((Zhang et al., 2021, Lemma 5.3)). *The softmax map is* 8*-smooth.*

**Lemma D.3.** *The softmax map,* $\mathrm{softmax} : \mathbb{R}^d \to \mathbb{R}^d$, *has an* $\frac{3}{\sqrt{2}}d^{3/2}$*-smooth gradient.*

*Proof.* Again we use $\sigma := \mathrm{softmax}(\theta)$ for brevity. We compute the second order derivatives:

$$\frac{\partial^2}{\partial\theta_j\partial\theta_k}\sigma_i = \frac{\partial}{\partial\theta_k}[\sigma_i(\delta_{ij} - \sigma_j)]$$
$$= \sigma_i(\delta_{ik} - \sigma_k)(\delta_{ij} - \sigma_j) - \sigma_i\sigma_j(\delta_{jk} - \sigma_k).$$

Every term is a function of $\theta$ and it is true in general that

$$\left|f(\theta)\,g(\theta)\,h(\theta) \ - \ f(\theta')\,g(\theta')\,h(\theta')\right| \ \le$$
$$|f(\theta) - f(\theta')|\,|g(\theta)|\,|h(\theta)| \ + \ |g(\theta) - g(\theta')|\,|f(\theta')|\,|h(\theta)| \ + \ |h(\theta) - h(\theta')|\,|f(\theta')|\,|g(\theta')|.$$

As such, we can write,

$$\left|\frac{\partial^2}{\partial\theta_j\partial\theta_k}\sigma_i(\theta) - \frac{\partial^2}{\partial\theta_j\partial\theta_k}\sigma_i(\theta')\right| \le 3\,\|\theta - \theta'\|_2$$

$\square$

**Lemma D.4.** *Assume* $\theta \in \mathbb{R}^d$ *with* $\theta \in \Theta_R := \{\theta \in \mathbb{R}^d : \theta^\top \mathbf{1} = 0 \text{ and } |\theta_i - \theta_j| \le 2R, \ \forall i,j \in [d]\}$. *Then, the following bounds hold true,*

- $\min_{i\in[d]} \mathrm{softmax}_i(\theta) \ge \frac{1}{1+(d-1)e^{2R}}$;

- $\max_{i\in[d]} \mathrm{softmax}_i(\theta) \ge \frac{1}{1+(d-1)e^{-2R}}$.

*Proof.*

**Minimum probability lower bound.** W.l.o.g. we minimize the first coordinate. We write,

$$\frac{e^{\theta_1}}{\sum_i e^{\theta_i}} = \frac{1}{1 + \sum_{i>1} e^{\theta_i - \theta_1}}.$$

By observing that,

$$e^{\theta_i - \theta_1} \le \max_j e^{\theta_j - \theta_1}$$

We can lower bound the value as,

$$\frac{e^{\theta_1}}{\sum_i e^{\theta_i}} \ge \frac{1}{1 + (d-1)\max_j\{e^{\theta_j - \theta_1}\}}$$

It suffices to maximize the quantity $\max_{j\neq 1, \theta\in\Theta_R}\{\theta_j - \theta_1\}$ as the RHS quantity is non-increasing in $\max_{j\neq 1, \theta\in\Theta_R}\{\theta_j - \theta_1\}$. *I.e.,* the largest difference between two coordinates of a vector in the sphere is $2R$. The minimum is achieved when $\theta_j - \theta_1 = 2R$ and $\theta_j = \theta_k, \forall j,k \ge 2$.

**Maximum probability lower bound.** Similarly, w.l.o.g, it suffices to maximize $\mathrm{softmax}_1(\theta)$ for $\theta \in \Theta_R$.

$$\frac{e^{\theta_1}}{\sum_i e^{\theta_i}} = \frac{e^{\theta_1}}{e^{\theta_1} + \sum_{i\neq 1} e^{\theta_i}}$$
$$\le \frac{e^{\theta_1}}{e^{\theta_1} + (d-1)e^{\sum_i \theta_i/(d-1)}}$$

where the inequality follows from the convexity of $e^x$. For any $\theta \in \Theta_R$ the point $(\overline{\theta}) = (\theta_1, \dots \frac{\theta_i}{d-1}, \dots)$ is also in $\Theta_R$ due to the convexity of the set (it is a linear polytope). We can simply optimize the objective,

$$\max_{a,b} \frac{1}{1 + (d-1)e^{b-a}}$$
$$\text{s.t. } |a - b| \leq 2R.$$

Due to the objective function's monotonicity in $b - a$, the program can be simplified even more into,

$$\min_{a,b} b - a$$
$$\text{s.t. } |a - b| \leq 2R.$$

Finally, it is clear that the last objective is minimized for $a - b = -2R$. Letting $\varepsilon \leq (d-1)^{-2}$.

$\square$

In this vein, if we want to bound the minimum probability of the softmax parametrized policy by $\varepsilon > 0$ for some $R > 0$, we need to set $R \leq 1/2 \log\left(\frac{1-\varepsilon}{\varepsilon(d-1)}\right)$. Then, it is also the case that $\max_{\theta \in \Theta_R, i} \text{softmax}_i(\theta) \geq \frac{1-\varepsilon}{1-\varepsilon+\varepsilon(d-1)^2} \geq 1 - \varepsilon - \varepsilon(d-1)^2$.

**Proposition 2.** Let $p$ be a probability vector in $\Delta^{d-1}$ and define $\theta(p)$ to be the set of $\theta$ such that $\text{softmax}(\theta) = p$. For any two $\theta, \theta' \in \theta(p)$, there exists a $c \in \mathbb{R}$ such that $\theta = \theta' + c\mathbf{1}$.

*Proof.* By assumption, $\text{softmax}(\theta) = \text{softmax}(\theta') = p$. For every entry $i$,

$$p_i = \frac{e^{\theta_i}}{\sum_i e^{\theta_i}} = \frac{e^{\theta_i'}}{\sum_i e^{\theta_i'}}.$$

Letting $Z := \sum_i^d e^{\theta_i}$, $Z' := \sum_i^d$, we observe,

$$\frac{e^{\theta_i}}{e^{\theta_i'}} = \frac{Z'}{Z} \implies$$
$$\theta_i = \theta_i' + \log \frac{Z'}{Z}, \; \forall i \in \{1, \dots, d\}.$$

Hence, any two $\theta, \theta'$ that map to the same probability vector are translations of each other in the direction of $\mathbf{1}$.

$\square$

**Proposition 3.** Let $p \in \Delta^{d-1}$ be a probability vector and the set, $\theta(p)$, of vectors $\theta \in \mathbb{R}^d$ such that $\text{softmax}(\theta) = p$. For the vector $\theta^\star := \arg\min_{\theta \in \Theta(p)} \|\theta\|^2$ it holds true that,

$$\mathbf{1}^\top \theta = 0.$$

*Proof.* The set $\theta(p)$ takes the form $\theta(p) := \{(\theta_i = \log p_i + c) \mid c \in \mathbb{R}\} = \{\theta_0 + c\mathbf{1} \mid c \in \mathbb{R}\}$ for an appropriate choice of $\theta_0$. Picking an arbitrary $\theta_0 \in \theta(p)$ to use as a reference, we can write the problem of minimizing $\|\theta\|_2$ as,

$$\min_{\theta \in \theta(p)} \|\theta\|^2 \equiv \min_{c \in \mathbb{R}} \|\theta_0 + c\mathbf{1}\|_2^2 \equiv \min_{c \in \mathbb{R}} \|\theta_0\|^2 + \langle \theta_0, c\mathbf{1} \rangle + \|c\mathbf{1}\|^2.$$

By the first-order optimality conditions, $c = -\frac{1}{d}\theta_0^\top \mathbf{1}$. Plugging back this for $\theta^\star$, we see $\theta^\star = \theta_0 - \frac{1}{d}\mathbf{1}(\theta_0^\top \mathbf{1})$. We see that, $\mathbf{1}^\top \theta^\star = \mathbf{1}^\top \theta_0 - \frac{d}{d}\theta_0^\top \mathbf{1} = 0$.

$\square$

**Lemma D.5.** *Assume a fixed $0 < R < \infty$ and define the set $\Theta_R$ to be $\Theta_R := \{\theta \in \mathbb{R}^d : \theta^\top \mathbf{1} = 0 \text{ and } |\theta_i - \theta_j| \leq 2R, \; \forall i, j \in [d]\}$. Then, $\text{softmax}(\Theta_R)$ is a convex set.*

*Proof.* For any $p \in \Delta^{d-1}$ for which $e^{-2R} \leq \frac{p_i}{p_j} \leq e^{2R}$, $\forall i, j \in [d]$, there exists $\theta \in \Theta_R$ such that $\text{softmax}(\theta) = p$. To see this, we apply the logarithm on the inequalities,

$$-2R \leq \log p_i - \log p_j \leq 2R. \tag{3}$$

A vector $\chi$ with entries $\chi_i := \log p_i$ clearly implements $p$. By (3) we see that subtracting $\kappa = \frac{\max_j \log p_j + \min_k \log p_k}{2}$ from all entries yields a softmax-equivalent vector $\chi_i' := \log p_i - \kappa$ with $-R \leq \chi_i' \leq R$. Conversely, for any $\theta \in \Theta_R$, $e^{-2R} \leq \frac{\text{softmax}_i(\theta)}{\text{softmax}_j(\theta)} \leq e^{2R}$.

Now, the set defined by the inequalities $p \in \Delta^{d-1}, e^{-2R} \leq \frac{p_i}{p_j} \leq e^{2R}$, is clearly a linear polytope and as such, convex. $\qquad\square$

# E  Gradient Domination

In this section we prove the gradient domination properties of the utilities of the game with different policy parametrizations. Further, for clarity, in place of $V_\tau^{x,y}$ we will use $V_\tau(x, y)$; and in place of $V_\tau^{\pi_\chi, \bar{\pi}_\theta}$ we will use $V_\tau(\chi, \theta)$.

## E.1  Direct Policy Parametrization pPŁ

**Lemma E.1.** *The utility of the game regularized with the weighted bidilated Euclidean regularizer with a weighting scheme defined in Appendix B.4.1, satisfies the* pPŁ *condition for directly parametrized policies,*

$$\frac{\tau \min_h \mu_c(h)\gamma^3}{101|\mathcal{H}|^3}[V_\tau(x, y) - V_\tau(x^\star, y)] \leq \frac{1}{2}\mathcal{D}_{\mathcal{X}}(x, \ell; y);$$

$$\frac{\tau \min_h \mu_c(h)\gamma^3}{101|\mathcal{H}|^3}[V_\tau(x, y^\star) - V_\tau(x, y)] \leq \frac{1}{2}\mathcal{D}_{\mathcal{Y}}(y, \ell; x).$$

*Proof.* We write the utility function of the regularized game,

$$H_\tau^{\text{eucl}}(\mu_1, \mu_2) := \langle \mu_1, \mathbf{R}\mu_2 \rangle - \tau\mathcal{R}_1^{\text{eucl}}(\mu_1, \mu_2) + \tau\mathcal{R}_2^{\text{eucl}}(\mu_1, \mu_2).$$

For player 1, we know that the function $H_\tau^{\text{eucl}}$ is strongly convex with an appropriate weighting scheme $\{w_{1,s}\}$, (correspondingly $\{w_{2,s}\}$ for player 2),

$$H_\tau^{\text{eucl}}(\mu_1', \mu_2) \geq H_\tau^{\text{eucl}}(\mu_1, \mu_2) + \langle \nabla_{\mu_1} H_\tau^{\text{eucl}}(\mu_1, \mu_2), \mu_1' - \mu_1 \rangle + \frac{\tau\alpha_{\text{bi}}^{\text{eucl}}}{2}\|\mu_1 - \mu_2\|_2^2$$

Strong convexity implies the KŁ condition for $\mu_1$. In turn, using the bound on the Lipschitz continuity modulus of the map $\mu_1 \mapsto x$,

$$H_\tau^{\text{eucl}}(\mu_1, \mu_2) - \min_{\mu_1^\star} H_\tau^{\star,\text{eucl}}(\mu_1^\star, \mu_2) \leq \frac{1}{2\tau\alpha_{\text{bi}}^{\text{eucl}}\left(\frac{\gamma}{|\mathcal{H}|}\right)^2}\|s_x\|_2^2. \tag{4}$$

Now, we know that $\alpha_{\text{bi}}^{\text{eucl}} = \frac{\gamma \min_h \mu_c(h)}{|\mathcal{H}|}$ (Corollary B.1). The conclusion follows from Lemma G.2. $\qquad\square$

## E.2  Softmax Policy Parametrization pPŁ

**Lemma E.2.** *The utility of the game with softmax-parametrized policies satisfies the two-sided* pPŁ *condition,*

$$\frac{\tau \min_h \mu_c(h)\gamma^3}{101|\mathcal{H}|^3(1 + (A-1)e^{2R})^2}[V_\tau(\chi, \theta) - V_\tau(\chi^\star, \theta)] \leq \frac{1}{2}\mathcal{D}_{X_R}(\chi, \ell; \theta)$$

$$\frac{\tau \min_h \mu_c(h)\gamma^3}{101|\mathcal{H}|^3(1 + (B-1)e^{2R})^2}[V_\tau(\chi, \theta^\star) - V_\tau(\chi, \theta)] \leq \frac{1}{2}\mathcal{D}_{\Theta_R}(\theta, \ell; \chi),$$

*where $\ell$ is the smoothness constant of the softmax-parametrized utility function.*

*Proof.* The main challenge in proving this lemma is the fact that the softmax mapping is not a bijection; this is manifested with a rank-deficient Jacobian of the mapping.

Concretely, from (4), we know that the KŁ-condition holds for the policies. What remains to show is that the KŁ-condition also holds for the parameters $\chi$ (and $\theta$).

For some $R > 0$, let $\mathcal{X}_R := \mathrm{softmax}(X_R)$ be the convex set of softmax-parametrized policies where $X_R := \left\{ \theta \in \mathbb{R}^A, A = \sum_s A_s : \chi_s^\top \mathbf{1} = 0, \forall s \in \mathcal{S}_1, |\chi_{s,i} - \chi_{s,j}| \leq 2R, \forall i, j \in [A_s] \right\}$. By overloading notation, let $V(\pi_\chi, \pi_\theta)$ be the loss function of the minimizing player as a function of policies $\pi_\chi, \pi_\theta$ and $V(\chi, \theta)$ the utility as a function of parameters $\chi, \theta$.

Now, we note that the subgradient $s \in \partial_{\pi_\chi} \left( V(\pi_\chi, \pi_\theta) + I_{\mathcal{X}_R}(\pi_\chi) \right)$ that minimizes $\|s\|$ is such that $s^\top \mathbf{1} = 0$. So when picking a norm-minimizing $s$, it suffices to look at the set of subgradients that are perpendicular to $\mathbf{1}$. Further, the chain rule applied on $V(\pi_\chi, \pi_\theta) + I_{\mathcal{X}_R}(\pi_\chi)$ yields,

$$\partial_\chi \left( V(\pi_\chi, \pi_\theta) + I_{\mathcal{X}_R}(\pi_\chi) \right) \subseteq \mathbf{J}(\chi) \left( \nabla_\pi V(\pi_\chi, \pi_\theta) + \partial_\pi I_{\mathcal{X}_R}(\pi_\chi) \right). \tag{5}$$

Moreover, we note that by the symmetry of $\mathbf{J}(\chi)$,

$$\begin{aligned}
\|\mathbf{J}(\chi)s\|^2 &= s^\top \mathbf{J}(\chi)^\top \mathbf{J}(\chi)s \\
&\geq \lambda_{\min}^+ (\mathbf{J}(\chi)^\top \mathbf{J}(\chi)) \|s\|^2 \\
&\geq \left( \lambda_{\min}^+ (\mathbf{J}(\chi)) \right)^2 \|s\|^2.
\end{aligned} \tag{6}$$

From inclusion (5) we infer that:

$$\min_{w \in \partial_\chi \left( V(\pi_\chi, \pi_\theta) + I_{\mathcal{X}_R}(\pi_\chi) \right)} \|w\| \geq \min_{v \in \mathbf{J}(\chi) \left( \nabla_\pi V(\pi_\chi, \pi_\theta) + \partial_\pi I_{\mathcal{X}_R}(\pi_\chi) \right)} \|v\|.$$

Lemma D.4 provides the bound $\lambda_{\min}^+ (\mathbf{J}(\chi)) \geq \frac{1}{1 + (B-1)e^{2R}}$ and the conclusion is proven.

$\square$

### E.3 Mahalanobis-pPŁ

**Lemma E.3.** *The utility of the game with softmax-parametrized policies satisfies the two-sided* Mahalanobis pPŁ *condition,*

$$\frac{\tau \min_h \mu_c(h) \gamma^3}{101 \lambda_{\max}(\mathbf{M}^{-1}) |\mathcal{H}|^3 (1 + (A-1)e^{2R})^2} \left[ V_\tau(\chi, \theta) - V_\tau(\chi^\star, \theta) \right] \leq \frac{1}{2} \mathcal{D}_{X_R}(\chi, \ell; \theta)$$

$$\frac{\tau \min_h \mu_c(h) \gamma^3}{101 \lambda_{\max}(\mathbf{M}^{-1}) |\mathcal{H}|^3 (1 + (B-1)e^{2R})^2} \left[ V_\tau(\chi, \theta^\star) - V_\tau(\chi, \theta) \right] \leq \frac{1}{2} \mathcal{D}_{\Theta_R}(\theta, \ell; \chi).$$

*Proof.* We invoke (6) and the fact that $\|w\|_{\mathbf{M}^{-1}}^2 \geq \lambda_{\min}^+ \left( \mathbf{M}^{-1} \right) \|w\|^2$ for any $\langle w, v \rangle = 0$, $\forall v \in \ker \left( \mathbf{M}^{-1} \right)$. Also, we use Equation ($\varepsilon$-trunc.) and Assumption 2 to bound $\lambda_{\min}^+ (\mathbf{M}^{-1})$. In detail, we know that,

$$\frac{|\mathcal{H}|^3 (1 + (A-1)e^{2R})^2}{\tau \min_h \mu_c(h) \gamma^3} \min_{w \in \partial_\chi \left( V(\pi_\chi, \pi_\theta) + I_{\mathcal{X}_R}(\pi_\chi) \right)} \|w\|^2 \geq V(\chi, \theta) - V(\chi^\star, \theta).$$

When $\mathbf{M} := \mathbf{F}(\chi, \theta)$, it is true that $\frac{\gamma^2 \min_h \mu_c(h)}{|\mathcal{H}|^2} \varepsilon \leq \lambda_{\max}(\mathbf{F}(\chi, \theta)) \leq 1$.

$\square$

**The spectrum of the Fisher Information Matrix** With the same arguments used in Lemma D.1, we can conclude that,

- $\lambda_{\min}(\mathbf{F}(\chi, \theta)) = 0$;
- $\lambda_{\min}^+ (\mathbf{F}(\chi, \theta)_s) \geq d(s) \min_a \pi_\chi(a|s)$;
- $d^{\chi, \theta}(s) \min_{s,a} \pi_\chi(a|s) \leq \lambda_{\max}(\mathbf{F}(\chi, \theta)_s) \leq d^{\chi, \theta}(s) \max_a \pi_\chi(a|s) + 1$.

Hence,

- $\lambda_{\min}^+ \left( \mathbf{F}(\chi, \theta) \right) \geq \min_{s,a} d^{\chi,\theta}(s) \pi_\chi(a|s)$;
- $\frac{\gamma^2 \min_h \mu_c(h)}{|\mathcal{H}|^2} \varepsilon \leq \lambda_{\max} \left( \mathbf{F}(\chi, \theta) \right) \leq 1$.

Moreover, $d^{\chi,\theta}(s) \geq \frac{\gamma^2 \min_h \mu_c(h)}{|\mathcal{H}|^2}$ by Assumption 2.

### E.4 Weak Gradient Domination

We now conclude this section with a proof of the weak gradient domination condition.

**Lemma E.4** (Utility Weak Gradient Domination). *Let $\Gamma$ be an IIEFG satisfying satisfying Assumption 2. Then, it holds true that,*

$$V^{\pi_1, \pi_2} - \min_{\pi_1'} V^{\pi_1', \pi_2} \leq \frac{1}{2\alpha_x} \max_{\pi_1'} \left\langle \nabla_{\pi_1} V^{\pi_1, \pi_2}, \pi_1 - \pi_1' \right\rangle;$$

$$\max_{\pi_2'} V^{\pi_1, \pi_2'} - V^{\pi_1, \pi_2} \leq \frac{1}{2\alpha_y} \max_{\pi_2'} \left\langle \nabla_{\pi_2} V^{\pi_1, \pi_2}, \pi_2' - \pi_2 \right\rangle,$$

*for $\alpha_x = \frac{\gamma}{\sqrt{2}|\mathcal{H}|^{\frac{3}{2}} A}$ and $\alpha_y = \frac{\gamma}{\sqrt{2}|\mathcal{H}|^{\frac{3}{2}} B}$.*

*Proof.* We use (Fatkhullin et al., 2023, Prop. 2) by using the fact that the diameter of the treeplex is at most $\sqrt{2|\mathcal{H}||A|}$ and the fact that the Lipschitz of $\mu_1^{\pi_1} \to \pi_1$ is $\frac{|\mathcal{H}|\sqrt{A}}{\gamma}$. Then, we use the fact that $\max_{\|y-x\| \leq 1, y \in \mathcal{X}} \left\langle \nabla f(x), x - y \right\rangle = \min_{v \in \partial_x(f + I_{\mathcal{X}}(x))} \|v\|$. $\qquad \square$

## F    Gradient Estimators

In this section, we demonstrate that the well-known stochastic gradient estimator, REINFORCE, can be used yield an unbiased estimate of bounded variance of the gradients of the non-regularized and regularized imperfect-information game.

### F.1    A Policy Gradient Theorem

We define a trajectory $\xi$ to be a sequence of consecutive history-action pairs, $\xi = \left( \left( h^{(1)}, a_{i(1)}^{(1)} \right), \left( h^{(2)}, a_{i(2)}^{(2)} \right), \ldots \right)$. The length of trajectory $\xi$ is noted as $K_\xi$ and it is bounded by the game-tree's height, $D(\mathcal{T})$. We define $\mathcal{K}$ to be the set of all trajectories and note that it is finite. After a policy profile, $(\pi_1, \pi_2)$, is fixed, the probability of each trajectory $\xi \in \mathcal{K}$ taking place is the product of the probability of each consecutive action,

$$\mathbb{P}^{\pi_1, \pi_2}(\xi) := \prod_{k=1}^{K_\xi} \pi_{i(k)} \left( a_{i(k)}^{(k)} | h^{(k)} \right).$$

where $i(k)$ denotes the player that takes an action at timestep $k$.

**Lemma F.1.** *Under the assumption of ($\varepsilon$-trunc.), it holds true that the gradient estimator (REINFORCE) is unbiased,*

$$\mathbb{E}_{\xi \sim \pi_1, \pi_2} \left[ \widehat{\nabla}_x \right] = \nabla_x V(\pi_1, \pi_2), \quad and \quad \mathbb{E}_{\xi \sim \pi_1, \pi_2} \left[ \widehat{\nabla}_y \right] = \nabla_y V(\pi_1, \pi_2);$$

*and also, its variance is bounded:*

$$\mathbb{E}_{\xi \sim \pi_1, \pi_2} \left[ \left\| \widehat{\nabla}_x - \nabla_x V(\pi_1, \pi_2) \right\|^2 \right] \leq \frac{A^2 D(\mathcal{T})^2}{\varepsilon};$$

$$\mathbb{E}_{\xi \sim \pi_1, \pi_2} \left[ \left\| \widehat{\nabla}_y - \nabla_y V(\pi_1, \pi_2) \right\|^2 \right] \leq \frac{B^2 D(\mathcal{T})^2}{\varepsilon}.$$

*where $A, B$ denote the maximum available number of action in any infoset for player 1 and 2 respectively.*

*Proof.* We first show that the gradient estimator is unbiased. Indeed,

$$
\nabla_x V(\pi_1, \pi_2) = \nabla_x \left( \sum_{\xi \in \mathcal{K}} r_\xi \, \mathbb{P}^{\pi_1, \pi_1}(\xi) \right)
$$

$$
= \sum_{\xi \in \mathcal{K}} r_\xi \nabla_x \, \mathbb{P}^{\pi_1, \pi_1}(\xi)
$$

$$
= \sum_{\xi \in \mathcal{K}} r_\xi \, \mathbb{P}_\xi \, \nabla_x \log \mathbb{P}^{\pi_1, \pi_1}(\xi)
$$

$$
= \sum_{\xi \in \mathcal{K}} r_\xi \, \mathbb{P}^{\pi_1, \pi_1}(\xi) \sum_{k=1}^{K_\xi} \left( \nabla_x \log \pi_{i(k)}\big(a_{i(k)}^{(k)} | h^{(k)}\big) \right)
$$

$$
= \mathbb{E}_{\xi \sim \pi_1, \pi_2} \left[ r_\xi \sum_{k=1}^{K_\xi} \nabla_x \log \pi_{i(k)}\big(a_{i(k)}^{(k)} | h^{(k)}\big) \right]
$$

$$
= \mathbb{E}_{\xi \sim \pi_1, \pi_2} \left[ r_\xi \sum_{k=1}^{K_\xi} \nabla_x \log \pi_1\big(a^{(k)} | s^{(k)}\big) \right]
$$

$$
= \mathbb{E}_{\xi \sim \pi_1, \pi_2} \left[ \widehat{\nabla}_x \right]
$$

The proof for $\widehat{\nabla}_y$ uses an identical argument. We will now proceed to show that the variance of the (REINFORCE) gradient estimator is bounded:

$$
\mathbb{E}_\xi \left[ \left\| \widehat{\nabla}_x - \mathbb{E}\left[ \widehat{\nabla}_x \right] \right\|^2 \right] \leq \mathbb{E}_\xi \left[ \left\| \widehat{\nabla}_x \right\|^2 \right]
$$

$$
= \mathbb{E}_\xi \left[ \left\| r_\xi \sum_{k=1}^{K_\xi} \nabla_x \log \pi_1\big(a^{(k)} | s^{(k)}\big) \right\|^2 \right]
$$

$$
\leq \mathbb{E}_\xi \left[ \left\| \sum_{k=1}^{K_\xi} \nabla_x \log \pi_1\big(a^{(k)} | s^{(k)}\big) \right\|^2 \right]
$$

$$
\leq \mathbb{E}_\xi \left[ K_\xi \sum_{k=1}^{K_\xi} \left\| \nabla_x \log \pi_1\big(a^{(k)} | s^{(k)}\big) \right\|^2 \right]
$$

$$
\leq D(\mathcal{T}) \mathbb{E}_\xi \left[ \sum_{k=1}^{K_\xi} \left\| \nabla_x \log \pi_1\big(a^{(k)} | s^{(k)}\big) \right\|^2 \right]
$$

$$
= D(\mathcal{T}) \mathbb{E}_\xi \left[ \sum_{k=1}^{K_\xi} \sum_{s,a} \mathbb{1}\{s = s^{(k)}, a = a^{(k)}\} \frac{1}{\pi_1^2(a|s^{(k)})} \right]
$$

$$
= D(\mathcal{T}) \mathbb{E}_\xi \left[ \sum_{k=1}^{K_\xi} \sum_{s,a} \mathbb{1}\{s = s^{(k)}\} \frac{1}{\pi_1(a|s^{(k)})} \right]
$$

$$
\leq \frac{A}{\varepsilon} D(\mathcal{T}) \mathbb{E}_\xi \left[ \sum_{k=1}^{K_\xi} \sum_{s,a} \mathbb{1}\{s = s^{(k)}\} \right]
$$

$$
= \frac{A}{\varepsilon} D(\mathcal{T}) \sum_{\xi \in \mathcal{K}} \mathbb{P}^{\pi_1, \pi_1}(\xi) \sum_{k=1}^{K_\xi} \sum_{s,a} \mathbb{1}\{s = s^{(k)}\}
$$

$$\leq \frac{A^2 D(\mathcal{T})^2}{\varepsilon}.$$

$\square$

**Lemma F.2.** *The variance of* (REINFORCE) *for softmax-parametrized policies is bounded as* $\sigma_\theta^2, \sigma_\chi^2 \leq 2D(\mathcal{T})^2$.

*Proof.* We see that $\nabla_\theta \log \pi_\theta(a|s) = e_{s,a} - \pi_\theta(\cdot|s)$. From then on, $\|\nabla_\theta \log \pi_\theta(a|s)\| \leq \sqrt{2}$ with probability 1. Then, the proof follows arguments similar to the previous one. $\square$

**Policy gradient of the bidilated regularizer** We define the policy gradient estimator of the bidilated regularizer, $\widehat{\nabla}_x \mathcal{R}_1$, as:

$$\widehat{\nabla}_x \mathcal{R}_1 := \left( \sum_k^{K_\xi} \psi\big(\pi_1\big(s^{(k)}\big)\big) \right) \sum_{k=1}^{K_\xi} \nabla_x \log \pi_1\big(a^{(k)}|s^{(k)}\big) + \sum_k^{K_\xi} \nabla_x \psi\big(\pi_1\big(s^{(k)}\big)\big).$$

We will demonstrate that this gradient estimator is, in fact, both unbiased and enjoys a variance that is bounded. We start with a preliminary proposition about an alternative expression of the regularizer.

**Proposition 4.** For a policy profile $\pi_1, \pi_2$, the bidilated regularizer, $\mathcal{R}_1$ can be alternatively defined as:

$$\mathcal{R}_1(\pi_1, \pi_2) = \sum_{\xi \in \mathcal{K}} \mathbb{P}^{\pi_1, \pi_2}(\xi) \left( \sum_k^{K_\xi} \psi\big(\pi_1\big(s^{(k)}\big)\big) \right).$$

*Proof.*

$$\begin{aligned}
\mathcal{R}_1(\pi_1, \pi_2) &= \sum_{s \in \mathcal{S}_1} \mu_1^{\pi_1}(\sigma(s)) \left( \sum_{h \in s} \mu_c(h) \mu_2^{\pi_2}(\sigma(h)) \right) \psi(\pi_1(s)) \\
&= \sum_{s \in \mathcal{S}_1} \mathbb{P}^{\pi_1, \pi_2}(s) \psi(\pi_1(s)) \\
&= \sum_{s \in \mathcal{S}_1} \mathbb{E}_\xi \left[ \sum_k^{K_\xi} \mathbb{1}\{s = s^{(k)}\} \psi(\pi_1(s)) \right] \\
&= \mathbb{E}_\xi \left[ \sum_{s \in \mathcal{S}_1} \sum_k^{K_\xi} \mathbb{1}\{s = s^{(k)}\} \psi(\pi_1(s)) \right] \\
&= \mathbb{E}_\xi \left[ \sum_k^{K_\xi} \sum_{s \in \mathcal{S}_1} \mathbb{1}\{s = s^{(k)}\} \psi(\pi_1(s)) \right] \\
&= \mathbb{E}_\xi \left[ \sum_k^{K_\xi} \psi\big(\pi_1\big(s^{(k)}\big)\big) \right] \\
&= \sum_{\xi \in \mathcal{K}} \mathbb{P}^{\pi_1, \pi_2}(\xi) \left( \sum_k^{K_\xi} \psi\big(\pi_1\big(s^{(k)}\big)\big) \right).
\end{aligned}$$

$\square$

With the latter expression, proving the desired properties is easier.

$$\nabla_x \mathcal{R}_1(\pi_1, \pi_2)$$

$$= \nabla_x \sum_{\xi \in \mathcal{K}} \mathbb{P}^{\pi_1, \pi_2}(\xi) \left( \sum_k^{K_\xi} \psi\big(\pi_1\big(s^{(k)}\big)\big) \right)$$

$$= \sum_{\xi \in \mathcal{K}} (\nabla_x \mathbb{P}^{\pi_1,\pi_2}(\xi)) \left( \sum_{k}^{K_\xi} \psi\big(\pi_1\big(s^{(k)}\big)\big) \right) + \sum_{\xi \in \mathcal{K}} \mathbb{P}^{\pi_1,\pi_2}(\xi) \left( \nabla_x \sum_{k}^{K_\xi} \psi\big(\pi_1\big(s^{(k)}\big)\big) \right)$$

$$= \underbrace{\sum_{\xi \in \mathcal{K}} (\mathbb{P}^{\pi_1,\pi_2}(\xi)\nabla_x \log \mathbb{P}^{\pi_1,\pi_2}(\xi)) \left( \sum_{k}^{K_\xi} \psi\big(\pi_1\big(s^{(k)}\big)\big) \right)}_{\varpi_1}$$

$$+ \underbrace{\sum_{\xi \in \mathcal{K}} \mathbb{P}^{\pi_1,\pi_2}(\xi) \left( \sum_{k}^{K_\xi} \nabla_x \psi\big(\pi_1\big(s^{(k)}\big)\big) \right)}_{\varpi_2}$$

For $\varpi_1$, let us denote $r_\xi = \sum_{k}^{K_\xi} \psi\big(\pi_1\big(s^{(k)}\big)\big)$,

$$\varpi_1 = r_\xi \sum_{\xi \in \mathcal{K}} \mathbb{P}^{\pi_1,\pi_2}(\xi)\nabla_x \log \mathbb{P}^{\pi_1,\pi_2}(\xi)$$

$$= \sum_{\xi \in \mathcal{K}} r_\xi \, \mathbb{P}_\xi \, \nabla_x \log \mathbb{P}_\xi$$

$$= \sum_{\xi \in \mathcal{K}} r_\xi \, \mathbb{P}_\xi \sum_{k=1}^{K_\xi} \left( \nabla_x \log \pi_{i(k)}\big(a^{(k)}_{i(k)}|h^{(k)}\big) \right)$$

$$= \mathbb{E}_{\xi \sim \pi_1,\pi_2} \left[ r_\xi \sum_{k=1}^{K_\xi} \nabla_x \log \pi_{i(k)}\big(a^{(k)}_{i(k)}|h^{(k)}\big) \right]$$

$$= \mathbb{E}_{\xi \sim \pi_1,\pi_2} \left[ r_\xi \sum_{k=1}^{K_\xi} \nabla_x \log \pi_1\big(a^{(k)}|s^{(k)}\big) \right].$$

For $\varpi_2$, we write,

$$\varpi_2 = \sum_{\xi \in \mathcal{K}} \mathbb{P}^{\pi_1,\pi_2}(\xi) \sum_{k}^{K_\xi} \nabla_x \psi\big(\pi_1\big(s^{(k)}\big)\big)$$

$$= \mathbb{E}_\xi \left[ \sum_{k}^{K_\xi} \nabla_x \psi\big(\pi_1\big(s^{(k)}\big)\big) \right]$$

We will use similar arguments for the variance in the case of the (REINFORCE) gradient estimator.

$$\mathbb{E}\left[ \left\| \widehat{\nabla}_x \mathcal{R}_1 - \mathbb{E}\left[ \widehat{\nabla}_x \mathcal{R}_1 \right] \right\|^2 \right]$$

$$\leq \mathbb{E}\left[ \left\| \widehat{\nabla}_x \mathcal{R}_1 \right\|^2 \right]$$

$$\leq \mathbb{E}\left[ 2 \underbrace{\left\| \left( \sum_{k}^{K_\xi} \psi\big(\pi_1\big(s^{(k)}\big)\big) \right) \sum_{k=1}^{K_\xi} \nabla_x \log \pi_1\big(a^{(k)}|s^{(k)}\big) \right\|^2}_{\vartheta_1} + 2 \underbrace{\left\| \sum_{k}^{K_\xi} \nabla_x \psi\big(\pi_1\big(s^{(k)}\big)\big) \right\|^2}_{\vartheta_2} \right]$$

For $\vartheta_1$, similar to Lemma F.1, we see that

$$\mathbb{E}[\vartheta_1] \leq \frac{A^2 \psi_{\max}^2 D(\mathcal{T})^2}{\varepsilon}.$$

Whereas, for $\vartheta_2$,

$$
\begin{aligned}
\mathbb{E}[\vartheta_2] &\leq \mathbb{E}\left[ K_\xi \sum_k^{K_\xi} \left\| \nabla_x \psi\big(\pi_1\big(s^{(k)}\big)\big) \right\|^2 \right] \\
&\leq \mathbb{E}\left[ K_\xi \sum_k^{K_\xi} L_\psi^2 \right] \\
&\leq D(\mathcal{T})^2 L_\psi^2.
\end{aligned}
$$

Finally, we note that when Assumption 2 is followed, then (REINFORCE) is also an unbiased estimator of bounded variance (same bounds as previously) of the perturbed version of the game. The reasoning is the same (when a player is exploring the gradient of the probability of an action is zero) and as such we omit it.

# G    Optimization Lemmata

**Definition 6** (Stationarity Proxies). *Assume a function $F : f + I_{\mathcal{X}}(\cdot)$ such that $f : \mathcal{X} \to \mathbb{R}$ is $\ell$-smooth relative to $\|\cdot\|_{\mathbf{M}}$ and $I_{\mathcal{X}}(\cdot)$ is the indicator function of the set $\mathcal{X}$. We define the following* stationarity proxies,

- *gradient of the Mahalanobis proximal mapping* (MPM),

$$\Delta_\rho(x) := \rho^2 \left\| x - \mathrm{prox}_{F/\rho}(x) \right\|_{\mathbf{M}_t}^2$$

 *with* $\mathrm{prox}_{F/\rho}(\cdot) := \arg\min_{x'}\{F(x') + \frac{\rho}{2} \| \cdot - x'\|_{\mathbf{M}}^2\}$.

- *Mahalanobis gradient mapping* (MGM),

$$\Delta_\rho^+(x) := \rho^2 \left\| x - x^+ \right\|_{\mathbf{M}_t}^2,$$

 *where* $x^+ := \arg\min_{x \in \mathcal{X}} \left\| x - \rho \mathbf{M}^{-1}\nabla f(x) \right\|_{\mathbf{M}}^2$,

- *Mahalanobis forward-backward mapping* (MFBM),

$$\mathcal{D}(x, \rho) := -2\rho \min_{x'}\{\langle \nabla f(x), x' - x\rangle + \frac{\rho}{2} \|x - x'\|_{\mathbf{M}}^2 + I_{\mathcal{X}}(x') - I_{\mathcal{X}}(x)\},$$

**Lemma G.1.** *The following properties hold true for the proximal point and the Mahalanobis Moreau envelope,*

- $\nabla F_\rho(x) = \frac{1}{\rho}(x - \hat{x})$

- $\mathrm{dist}(0, \partial F(\hat{x})) \leq \|\nabla F_\rho(x)\|_{\mathbf{M}^{-1}}$

- $F(\hat{x}) \leq F_\rho(\hat{x}) \leq F(x)$

*Proof.* The first and last items follow easily from the definition and standard arguments (Davis and Drusvyatskiy, 2018). The middle one uses the optimality condition of $\hat{x} := \mathrm{prox}_{\rho F}(x)$,

$$0 \in \partial \left( F(\hat{x}) + \frac{1}{\rho}\mathbf{M}(\hat{x} - x) \right),$$

from which we conclude,

$$\frac{1}{\rho}\mathbf{M}(x - \hat{x}) \in \partial F(\hat{x}).$$

Finally, we conclude that $\min_{s_{\hat{x}} \in \partial F(\hat{x})} \|s_{\hat{x}}\|_{\mathbf{M}^{-1}}^2 \leq \frac{1}{\rho^2} \|x - \hat{x}\|_{\mathbf{M}}^2$. $\qquad\square$

**Definition 7** (pPŁ, KŁ). *Let $f : \mathcal{X} \to \mathbb{R}$ be an L-Lipschitz continuous function with $\ell$-Lipschitz continuous gradient. Then,*

- Proximal Polyak-Łojasiewicz (pPŁ): *$f$ is said to satisfy the proximal Polyak-Łojasiewicz condition if $\exists \alpha > 0$ s.t.*
$$\frac{1}{2}\mathcal{D}_{\mathcal{X}}(x, \ell) \geq \alpha \left[f(x) - f(x^\star)\right]$$

- Kurdyka-Łojasiewicz (KŁ): *$f$ is said to satisfy if $\exists \overline{\alpha}$ s.t.*
$$\min_{s_x \in \partial(f + I_{\mathcal{X}})(x)} \|s_x\|^2 \geq 2\overline{\alpha} \left[f(x) - f(x^\star)\right], \quad \forall x \in \mathcal{X}.$$

The definitions for the Mahalanobis analogues of pPŁ and KŁ follow straightforward extension.

**Lemma G.2.** *Let $f$ be an $\ell$-smooth function relative to $\|\cdot\|_{\mathbf{M}}^2$ defined over the convex set $\mathcal{X}$. If $f$ satisfies the (Mahalanobis) KŁ condition with modulus $\alpha_{\mathrm{kl}}$, it also satisfies the (Mahalanobis) pPŁ condition with a modulus of $\alpha_{\mathrm{ppl}} = \frac{\alpha_{\mathrm{kl}}}{2\ell^2}$.*

*Proof.* First, we define $F(x) := f(x) + I_{\mathcal{X}}(x)$, with $I_{\mathcal{X}}(\cdot)$ being the indicator function. We highlight that since $I_{\mathcal{X}}(\cdot)$ is convex and $f$ is $\ell$-smooth (relative to $\|\cdot\|_{\mathbf{M}}^2$), then $F$ is $\ell$-weakly convex (relative to $\|\cdot\|_{\mathbf{M}}^2$). This means that the proximal point of the function $F/\rho$ is well defined for any $\rho > \ell$.

Now, assume a point $x \in \mathcal{X}$ and $\hat{x} := \operatorname{prox}_{F/\rho}(x)$. By assumption, for any $\hat{x} \in \mathcal{X}$, it holds true that,

$$\frac{1}{2}\|s_{\hat{x}}\|^2 \geq \alpha\left[f(\hat{x}) - f^\star\right]$$

where $s_{\hat{x}} \in \partial F(\hat{x})$. The latter implies that for the gradient of the Mahalanobis-Moreau envelope of $F$, it holds that,

$$
\begin{aligned}
\frac{1}{2}\left\|\nabla F_{1/\rho}(x)\right\|_{\mathbf{M}^{-1}}^2 &\geq \alpha[f(\hat{x}) - f^\star] \\
&= \alpha + \alpha[f(\hat{x}) - f(x)] \\
&\geq \alpha[f(x) - f^\star] - \alpha\left(\frac{1}{2\rho}\mathcal{D}(x,\rho) + \frac{\ell+\rho}{2}\|x-\hat{x}\|_{\mathbf{M}}^2\right) \quad (7)
\end{aligned}
$$

where (7) follows from the fact that $F$ is an $\ell$-weakly convex function, and for every $v \in \partial F(x)$. To see this, we write that due to weak convexity (relative to $\|\cdot\|_{\mathbf{M}}^2$),

$$
\begin{aligned}
F(\hat{x}) &\geq F(x) + \langle v, \hat{x} - x\rangle - \frac{\ell}{2}\|x-\hat{x}\|_{\mathbf{M}}^2 \\
&= F(x) + \langle v, \hat{x} - x\rangle + \frac{\rho}{2}\|x-\hat{x}\|_{\mathbf{M}}^2 - \frac{\ell+\rho}{2}\|x-\hat{x}\|_{\mathbf{M}}^2 \\
&\geq F(x) + \min_{y \in \mathcal{Y}}\left\{\langle\nabla f(x), y - x\rangle + \frac{\rho}{2}\|x-y\|_{\mathbf{M}}^2\right\} - \frac{\ell+\rho}{2}\|x-\hat{x}\|_{\mathbf{M}}^2 \\
&= F(x) - \frac{1}{2\rho}\mathcal{D}(x,\rho) - \frac{\ell+\rho}{2}\|x-\hat{x}\|_{\mathbf{M}}^2
\end{aligned}
$$

Collecting the terms,

$$\left(\frac{1}{2} + \alpha\frac{\ell+\rho}{2\rho^2}\right)\|\nabla F_\rho(x)\|_{\mathbf{M}^{-1}}^2 + \frac{\alpha}{2\rho}\mathcal{D}(x,\rho) \geq \alpha\left[f(x) - f^\star\right].$$

A direct generalization of (Karimi et al., 2016, Lemma 1), implies that for the MFBM and a choice of $\rho_1, \rho_2 > 0$ such that$\rho_1 > \rho_2$, then $\mathcal{D}(x, \rho_1) \geq \mathcal{D}(x, \rho_1)$. As such, we write,

$$\left(\frac{1}{2} + \alpha\frac{\ell+\rho}{2\rho^2}\right)\|\nabla F_{1/\rho}(x)\|_{\mathbf{M}^{-1}}^2 + \frac{\alpha}{2\rho}\mathcal{D}(x,2\rho) \geq \alpha\left[f(x) - f^\star\right].$$

We can pick $\rho = 4\ell$ which then yields,

$$\left(\frac{1}{2} + \frac{12\alpha}{\ell}\right)\|\nabla F_{1/(4\ell)}(x)\|_{\mathbf{M}^{-1}}^2 + \frac{\alpha}{8\ell}\mathcal{D}(x,4\ell) \geq \alpha\left[f(x) - f^\star\right].$$

Observing that $\alpha \leq \ell$ in general, we re-write:

$$\frac{25}{2}\|\nabla F_{1/(4\ell)}(x)\|_{\mathbf{M}^{-1}}^2 + \frac{1}{8}\mathcal{D}(x,4\ell) \geq \alpha\left[f(x) - f^\star\right].$$

Now, from (Fatkhullin and He, 2024, Lemmata 4.1 & 4.2), we know that,

$$16\mathcal{D}(x,4\ell) \geq \left\|\nabla F_{1/\rho}(\hat{x})\right\|_{\mathbf{M}^{-1}}^2$$

which we plugin in the former inequality to finally conclude that,

$$\frac{1}{2}\mathcal{D}(x,4\ell) \geq \frac{\mu}{101}[f(x) - f^\star].$$

$\square$

**Remark 3.** *The latter lemma provides a bound that is significantly tighter than the one implied by the analysis found (Karimi et al., 2016, Appendix G) which connects the moduli of the KŁand pPŁconditions.*

### G.1 A Variation of the Descent Lemma

The following lemma is a consequence of the three-point identity of the Mahalanobis norm and the smoothness of $f$.

**Lemma G.3** ((J Reddi et al., 2016, Lemma 1)). *Let $f : \mathcal{X} \to \mathbb{R}$ be an $\ell$-smooth function relative to $\|\cdot\|_{\mathbf{M}_t}$ and a point $x \in \mathcal{X} \subseteq \mathbb{R}^d$. Also, define the vector $v \in \mathbb{R}^d$ and $y \in \mathcal{X}$ to be*

$$y := \operatorname*{Proj}_{\mathcal{X},\mathbf{M}_t} \left( x - \eta \mathbf{M}_t^{-1} v \right).$$

*Then, the following inequality holds true:*

$$f(y) \le f(z) + \langle \nabla f(x) - v, y - z \rangle$$
$$+ \left( \frac{\ell}{2} - \frac{1}{2\eta} \right) \|y - x\|_{\mathbf{M}_t}{}^2 + \left( \frac{\ell}{2} + \frac{1}{2\eta} \right) \|z - x\|_{\mathbf{M}_t}^2 - \frac{1}{2} \|y - z\|_{\mathbf{M}_t}^2.$$

**Lemma G.4.** *Let $\mathcal{X} \subseteq \mathbb{R}^d$ be a closed convex set, and let $f : \mathcal{X} \to \mathbb{R}$ be an $\ell$-smooth function relative to $\|\cdot\|_{\mathbf{M}_t}$ for some $\ell > 0$. Suppose $\eta > 0$ with $\eta \le \frac{1}{5\ell}$. For any $x \in \mathcal{X}$ and any vector $v \in \mathbb{R}^d$, define $x^+ = \operatorname{Proj}_{\mathcal{X},\mathbf{M}_t} (x - \eta v)$. Then the following inequality holds:*

$$f(x^+) \le f(x) - \frac{\eta}{6} \mathcal{D}_{\mathcal{X}}(x, 1/\eta) + \frac{\eta}{2} \|\nabla f(x) - v\|_{\mathbf{M}_t^{-1}}^2.$$

*Proof.* First, we define $\overline{x}^+ := \operatorname{Proj}_{\mathcal{X},\mathbf{M}_t} \left( x - \frac{1}{\rho} \mathbf{M}_t^{-1} \nabla f(x) \right).$

- Invoking $\ell$-smoothness relative to $\|\cdot\|_{\mathbf{M}_t}$ of $f$ for $x, \overline{x}_+$ and assuming $\rho > 0$ with $\rho \ge \ell$,

$$f(\overline{x}_+) \le f(x) + \langle \nabla f(x), \overline{x}_+ - x \rangle + \frac{\ell}{2} \|x_+ - x\|_{\mathbf{M}_t}^2$$
$$\le f(x) + \langle \nabla f(x), \overline{x}_+ - x \rangle + \frac{\rho}{2} \|x_+ - x\|_{\mathbf{M}_t}^2$$
$$= f(x) - \left( \langle \nabla f(x), x - \overline{x}_+ \rangle - \frac{\rho}{2} \|x_+ - x\|_{\mathbf{M}_t}^2 \right)$$
$$= f(x) - \frac{1}{2\rho} \mathcal{D}_{\mathbf{M}_t}(x, \rho). \tag{8}$$

- Invoking Lemma G.3 with $x = x, y = \overline{x}_+, z = x, v = \nabla f(x)$

$$f(\overline{x}_+) \le f(x) + \left( \frac{\ell}{2} - \frac{1}{\rho} \right) \|\overline{x}_+ - x\|_{\mathbf{M}_t}^2. \tag{9}$$

- Again, invoking Lemma G.3 but with $x = x, y = x_+, z = \overline{x}_+, v,$

$$f(x_+) \le f(\overline{x}_+) + \langle \nabla f(x) - v, x_+ - \overline{x}_+ \rangle$$
$$+ \left( \frac{\ell}{2} - \frac{1}{2\eta} \right) \|x_+ - x\|_{\mathbf{M}_t}^2 + \left( \frac{\ell}{2} + \frac{1}{2\eta} \right) \|\overline{x}_+ - x\|_{\mathbf{M}_t}^2 - \frac{1}{2\eta} \|x_+ - \overline{x}_+\|_{\mathbf{M}_t}^2. \tag{10}$$

Combining the previous inequalities as $1/3\times$(8) and $2/3\times$(9), and letting $1/\rho = \eta \le \frac{1}{\ell}$ yields,

$$f(\overline{x}_+) \le f(x) - \frac{1}{6\eta} \mathcal{D}_{\mathcal{X}}(x, 1/\eta) + \left( \frac{\ell}{3} - \frac{2}{3\eta} \right) \|\overline{x}_+ - x\|_{\mathbf{M}_t}^2$$

Adding (10),

$$f(x_+) \le f(x) - \frac{\eta}{6} \mathcal{D}_{\mathcal{X}}(x, 1/\eta) + \left( \frac{\ell}{3} - \frac{2}{3\eta} \right) \|\overline{x}_+ - x\|_{\mathbf{M}_t}^2$$

$$+ \langle \nabla f(x) - v, x_+ - \overline{x}_+ \rangle$$

$$+ \left( \frac{\ell}{2} - \frac{1}{2\eta} \right) \|x_+ - x\|_{\mathbf{M}_t}^2 + \left( \frac{\ell}{2} + \frac{1}{2\eta} \right) \|\overline{x}_+ - x\|_{\mathbf{M}_t}^2 - \frac{1}{2\eta} \|x_+ - \overline{x}_+\|_{\mathbf{M}_t}^2$$

$$\leq f(x) - \frac{\eta}{6} \mathcal{D}_{\mathcal{X}}(x, 1/\eta) + \left( \frac{5\ell}{6} - \frac{1}{6\eta} \right) \|\overline{x}_+ - x\|_{\mathbf{M}_t}^2$$

$$+ \frac{\rho}{2} \|\nabla f(x) - v\|_{\mathbf{M}_t^{-1}}^2 + \frac{1}{2\rho} \|x_+ - \overline{x}_+\|_{\mathbf{M}_t}^2$$

$$+ \left( \frac{\ell}{2} - \frac{1}{2\eta} \right) \|x_+ - x\|_{\mathbf{M}_t}^2 - \frac{1}{2\eta} \|x_+ - \overline{x}_+\|_{\mathbf{M}_t}^2 \tag{11}$$

$$= f(x) - \frac{\eta}{6} \mathcal{D}_{\mathcal{X}}(x, 1/\eta) + \left( \frac{5\ell}{6} - \frac{1}{6\eta} \right) \|\overline{x}_+ - x\|_{\mathbf{M}_t}^2$$

$$+ \frac{\eta}{2} \|\nabla f(x) - v\|_{\mathbf{M}_t^{-1}}^2$$

$$+ \left( \frac{\ell}{2} - \frac{1}{2\eta} \right) \|x_+ - x\|_{\mathbf{M}_t}^2$$

$$\leq f(x) - \frac{\eta}{6} \mathcal{D}_{\mathcal{X}}(x, 1/\eta) + \frac{\eta}{2} \|\nabla f(x) - v\|_{\mathbf{M}_t^{-1}}^2 \tag{12}$$

- (11) follows from the application of Young's inequality on

$$\langle \nabla f(x) - v, x^+ - \overline{x}^+ \rangle = \left\langle \mathbf{M}_t^{-1/2} \nabla f(x) - v, \mathbf{M}_t^{1/2} x^+ - \overline{x}^+ \right\rangle;$$

- (12) follows by dropping the non-positive terms; non-positivity follows from the choice of the step-size, $\eta \leq \frac{1}{5\ell}$.

$\square$

### G.2   Min-Max Optimization

**Lemma G.5.** *Let $f : \mathcal{X} \times \mathcal{Y}$ be an $\ell$-smooth function, $\rho > 0$, two points $y, y' \in \mathcal{Y}$, and a point $x \in \mathcal{X}$. Then, the following inequality holds:*

$$|\mathcal{D}_{\mathcal{X}}(x, \rho; y) - \mathcal{D}_{\mathcal{X}}(x, \rho; y')| \leq 3\lambda_{\max}(\mathbf{M}_t^{-1})\ell^2 \|y - y'\|^2.$$

*Proof.* We define $\overline{x}, \overline{x}' \in \mathcal{X}$ to be:

$$\overline{x} := \underset{\mathcal{X}, \mathbf{M}_t}{\mathrm{Proj}} \left( x - \frac{1}{\rho} \mathbf{M}_t^{-1} \nabla_x f(x, y) \right);$$

$$\overline{x}' := \underset{\mathcal{X}, \mathbf{M}_t}{\mathrm{Proj}} \left( x - \frac{1}{\rho} \mathbf{M}_t^{-1} \nabla_x f(x, y') \right).$$

By the definition of $\mathcal{D}_{\mathcal{X}}(x, \rho; y')$ we write:

$$\begin{cases} \frac{1}{2\rho} \mathcal{D}_{\mathcal{X}}(x, \rho; y) = \langle \nabla f(x, y), x - \overline{x} \rangle - \frac{\rho}{2} \|x - \overline{x}\|_{\mathbf{M}_t}^2; \\ \frac{1}{2\rho} \mathcal{D}_{\mathcal{X}}(x, \rho; y') = \langle \nabla f(x, y'), x - \overline{x}' \rangle - \frac{\rho}{2} \|x - \overline{x}'\|_{\mathbf{M}_t}^2. \end{cases}$$

Considering the difference $\mathcal{D}_{\mathcal{X}}(x, \rho; y) - \mathcal{D}_{\mathcal{X}}(x, \rho; y')$ we see that:

$$\frac{1}{2\rho} |\mathcal{D}_{\mathcal{X}}(x, \rho; y) - \mathcal{D}_{\mathcal{X}}(x, \rho; y')|$$

$$= \left| \langle \nabla_x f(x, y) - \nabla_x f(x, y'), \overline{x}' - \overline{x} \rangle - \frac{\rho}{2} \left( \|x - \overline{x}\|_{\mathbf{M}_t}^2 - \|x - \overline{x}'\|_{\mathbf{M}_t}^2 \right) \right|$$

$$\leq |\langle \nabla_x f(x,y) - \nabla_x f(x,y'), \overline{x}' - \overline{x} \rangle| + \frac{\rho}{2}\left|\left(\|x - \overline{x}\|^2_{\mathbf{M}_t} - \|x - \overline{x}'\|^2_{\mathbf{M}_t}\right)\right|$$

$$\leq |\langle \nabla_x f(x,y) - \nabla_x f(x,y'), \overline{x}' - \overline{x} \rangle| + \frac{\rho}{2}\|\overline{x} - \overline{x}'\|^2_{\mathbf{M}_t}$$

$$\leq \|\nabla_x f(x,y) - \nabla_x f(x,y')\|_{\mathbf{M}_t^{-1}}\|\overline{x}' - \overline{x}\|_{\mathbf{M}_t} + \frac{\rho}{2}\|\overline{x} - \overline{x}'\|^2_{\mathbf{M}_t}$$

$$\leq \frac{1}{\rho}\|\nabla_x f(x,y) - \nabla_x f(x,y')\|^2_{\mathbf{M}_t^{-1}} + \frac{1}{2\rho}\|\nabla_x f(x,y) - \nabla_x f(x,y')\|^2_{\mathbf{M}_t^{-1}}$$

$$\leq \frac{\lambda_{\max}(\mathbf{M}_t^{-1})}{\rho}\|\nabla_x f(x,y) - \nabla_x f(x,y')\|^2 + \frac{\lambda_{\max}(\mathbf{M}_t^{-1})}{2\rho}\|\nabla_x f(x,y) - \nabla_x f(x,y')\|^2$$

$$\leq \frac{3\lambda_{\max}(\mathbf{M}_t^{-1})\ell^2}{2\rho}\|y - y'\|^2.$$

We note that:

- The first inequality follows from the triangle inequality.

- In the second inequality, we applied the reverse triangle inequality.

- The third uses the Cauchy-Schwarz inequality.

- Finally, the second to last uses Lemma G.9 while, the last one, invokes the $\ell$-Lipschitz continuity of the gradient.

$\square$

**Lemma G.6.** *Let $f : \mathcal{X} \times \mathcal{Y}$ be an $\ell$-smooth function such that for any $x \in \mathcal{X}$, $f(x,\cdot)$ satisfies the proximal-PŁ condition with modulus $\alpha > 0$. Then, the function $\Phi(x) := \arg\max_{y \in \mathcal{Y}} f(x,y)$ is $\ell_\star$-smooth, with*

$$\ell_\star := \ell\left(1 + \frac{\ell}{\alpha}\right).$$

*Proof.* We effectively need to show Lipschitz continuity of the maximizers $y^\star(\cdot) := \arg\max_x$ and the proof will follow from Danskin's lemma and $f$'s own $\ell$-smoothness. So, we write by the quadratic growth condition,

$$\frac{\alpha}{2}\|y^\star(x') - y^\star(x)\|^2 \leq f(x, y^\star(x)) - f(x, y^\star(x')). \tag{13}$$

We denote $\mathcal{D}_{\mathcal{Y}}(\cdot, \rho; x) := -2\rho \arg\min_{z \in \mathcal{Y}}\{\langle -\nabla f(x,y), z - y\rangle + \frac{\rho}{2}\|y - z\|^2\}$ and by the proximal-PŁ condition, we write,

$$f(x, y^\star(x)) - f(x, y^\star(x')) \leq \frac{1}{2\alpha}\mathcal{D}_{\mathcal{Y}}(y, \ell; x). \tag{14}$$

Now, we aim to bound $\mathcal{D}_{\mathcal{Y}}(y, \ell; x)$ by $\|y^\star(x) - y^\star(x')\|^2$. We observe that,

$$\mathcal{D}_{\mathcal{Y}}(y^\star(x), \ell; x) = 0.$$

Hence,

$$\mathcal{D}_{\mathcal{Y}}(y^\star(x'), \ell; x) = \mathcal{D}_{\mathcal{Y}}(y^\star(x'), \ell; x) - \mathcal{D}_{\mathcal{Y}}(y^\star(x), \ell; x)$$
$$\leq 2\ell^2\|x - x\|^2 \tag{15}$$

where the last line follows from a slight sharpening of the proof of Lemma G.5 (for the function $h(y,x) = -f(x,y)$ and $\mathbf{M} = \mathbf{I}$). Finally, piecing inequalities (13), (14), and (15) together,

$$\|y^\star(x) - y^\star(x')\| \leq \frac{\ell}{\alpha}\|x - x'\|. \tag{16}$$

What is left to do is to observe the following, due to Danskin's theorem and $\ell$-smoothness of $f$,

$$\|\nabla_x \Phi(x) - \nabla_x \Phi(x')\| = \|\nabla_x f(x, y^\star(x)) - \nabla_x f(x', y^\star(x'))\|$$

$$\leq \ell \left\| (x, y^\star(x)) - (x', y^\star(x')) \right\|$$
$$\leq \ell \left\| x - x' \right\| + \frac{\ell^2}{\alpha} \left\| x - x' \right\|.$$

The latter inequality follows from (16) and completes the proof. $\qquad\square$

**Lemma G.7** ((Kalogiannis et al., 2025, Lemma D.3)). *Let $f : \mathcal{X} \times \mathcal{Y}$ be an $\ell$-smooth function. Additionally, assume that $f(\cdot, y)$ is $\alpha_x$-pPŁ for all $y \in \mathcal{Y}$ and $f(x, \cdot)$ is $\alpha_y$-pPŁ for all $x \in \mathcal{X}$. Then, it holds true that:*

$$\Phi^\star := \min_{x \in \mathcal{X}} \max_{y \in \mathcal{Y}} f(x, y) = \max_{y \in \mathcal{Y}} \min_{x \in \mathcal{X}} f(x, y).$$

.

**Lemma G.8** ((Kalogiannis et al., 2025, Lemma D.4)). *Let $f : \mathcal{X} \times \mathcal{Y}$ be an $\ell$-smooth function. Additionally, assume that $f(\cdot, y)$ is $\alpha_x$-pPŁ for all $y \in \mathcal{Y}$ and $f(x, \cdot)$ is $\alpha_y$-pPŁ for all $x \in \mathcal{X}$. Then, the function $\Phi(x) := \max_{y \in \mathcal{Y}} f(x, y)$ is $\alpha_x$-pPŁ.*

### G.3 Regarding the Mahalanobis Distance

Throughout, we will refer to a positive-semidefinite matrix $\mathbf{M} \in \mathbb{R}^{d \times d}$ and its Moore-Penrose pseudo-inverse $\mathbf{M}^\dagger \in \mathbb{R}^{d \times d}$. Although in general a PSD matrix cannot define a distance, restricting $x, y \in \mathbb{R}^d$ such that $(x - y) \in \ker(\mathbf{M})^\perp$, then $\|x - y\|_{\mathbf{M}}^2 := (x - y)^\top \mathbf{M}(x - y)$ satisfies all properties of a metric. As we shall see, this seemingly arbitrary assumption is satisfied for every pair of consecutive updates of natural policy gradient steps. The matrix rank-deficient matrix we are interested in is policy gradient Fisher information matrix, and for softmax policy parametrization, it is rank deficient in the direction $\mathbf{1} \in \mathbb{R}^d$. Further, the gradient $\nabla f(x)$ as

**Proposition 5.** Assume that $\theta_0 = \mathbf{0}$. Also, let $v_t^\top \mathbf{1} = 0, \ \forall t \in \{1, 2, 3, \dots\}$. Then, setting $\theta_{t+1} = \theta_t - \eta \mathbf{M}^\dagger v_t$ guarantees that,

$$(\theta_{t+1} - \theta_t)^\top \mathbf{1} \quad \text{and} \quad \theta_t^\top \mathbf{1} = 0, \ \forall t.$$

*Proof.* Since, $\theta_{t+1} = \theta_t - \eta \mathbf{M}^\dagger v_t$, we see that $\theta_{t+1}^\top \mathbf{1} = (\theta_t - \eta \mathbf{M}^\dagger v_t)^\top \mathbf{1} = 0$ and $(\theta_{t+1} - \theta_t)^\top \mathbf{1} = 0$. $\qquad\square$

**Proposition 6.** Let $\Theta \subseteq \mathbb{R}^d$ be a convex compact set. Assume that $\theta_0 = \mathbf{0}$. Also, let $v_t^\top \mathbf{1} = 0, \ \forall t \in \{1, 2, 3, \dots\}$. Then, the following minimization problem has a unique solution,

$$\min_{\theta \in \Theta, \text{s.t.} (\theta - \theta_t)^\top \mathbf{1} = 0} \left\| \left( \theta_t - \eta \mathbf{M}^\dagger v_t \right) - \theta \right\|_{\mathbf{M}}^2.$$

Further, it is equivalent to the minimization problem,

$$\min_{\theta \in \Theta, \text{s.t.} (\theta - \theta_t)^\top \mathbf{1} = 0} \left\{ \langle v_t, \theta - \theta_t \rangle + \frac{1}{2\eta} \|\theta - \theta_t\|_{\mathbf{M}}^2 \right\}.$$

*Proof.* It is clear that, for $\theta, \chi \in \Theta, \theta^\top \mathbf{1} = \chi^\top \mathbf{1} = 0$ the function $\|\theta\|_{\mathbf{M}}^2, \|\theta - \chi\|_{\mathbf{M}}^2$ is strongly convex in $\theta$. Hence, both problems attain a unique minimum.

For the first problem, the first-order optimality conditions for the write,

$$\left\langle \theta^+ - \left( \theta - \eta \mathbf{M}^\dagger v_t \right), \theta - \theta^+ \right\rangle \geq 0, \quad \forall \theta \in \Theta, \theta^\top \mathbf{1} = 0.$$

Noting that, $(\theta^+ - (\theta_t - \eta \mathbf{M}^\dagger v_t))^\top \mathbf{1} = 0$ and $(\theta - \theta)^\top \mathbf{1} = 0$,

$$\left\langle \mathbf{M}\theta^+ - \mathbf{M}\theta + \eta v_t, \mathbf{M}^\dagger(\theta - \theta^+) \right\rangle \geq 0, \quad \forall \theta \in \Theta, \theta^\top \mathbf{1} = 0$$

But, since the matrix $\mathbf{M}$ is PSD and the last inequality is a condition on the sign of the inner-product, it can be written equivalently as,

$$\left\langle \mathbf{M}\theta^+ - \mathbf{M}\theta + \eta v_t, (\theta - \theta^+) \right\rangle \geq 0, \quad \forall \theta \in \Theta, \theta^\top \mathbf{1} = 0.$$

The final inequality, is exactly the first-order optimality condition for the second minimization problem. $\qquad\square$

### G.4 Alternating Mirror Descent using a Changing Mahalanobis DGF

#### G.4.1 Supporting Lemmata

**Lemma G.9.** *Let $v_1, v_2$ be vectors in $\mathbb{R}^d$ and $\mathcal{X} \subseteq \mathbb{R}^d$ be a compact convex set and a scalar $\eta > 0$. Also, let points $x_1^+, x_2^+ \in \mathcal{X}$ such that:*

$$x_1^+ := \underset{\mathcal{X}, \mathbf{M}_t}{\mathrm{Proj}} \left( x - \eta \mathbf{M}_t^{-1} v_1 \right);$$

$$x_2^+ := \underset{\mathcal{X}, \mathbf{M}_t}{\mathrm{Proj}} \left( x - \eta \mathbf{M}_t^{-1} v_2 \right).$$

*Then, it holds true that:*

$$\left\| x_1^+ - x_2^+ \right\|_{\mathbf{M}_t} \leq \eta \left\| v_1 - v_2 \right\|_{\mathbf{M}_t^{-1}}.$$

.

#### Smoothness Relative to the Mahalanobis Distance

**Proposition 7.** Let $f$ be a function $\ell$-smooth relative to the $\ell_2$-distance. Then, it is $\frac{\ell}{\lambda_{\min}(\mathbf{M}_t)}$-smooth relative to the Mahalanobis distance induced by a positive definite matrix $\mathbf{M}_t$.

*Proof.* We will merely demonstrate that if $f$ is $\ell$-smooth (relative to $\ell_2$-distance) it is also the case that:

$$|f(y) - f(x) - \langle \nabla f(x), y - x \rangle| \leq \frac{\ell}{2\lambda_{\min}(\mathbf{M}_t)} \|x - y\|_{\mathbf{M}_t}^2$$

For one direction we use vector norm equivalence to write:

$$f(y) \geq f(x) + \langle \nabla f(x), y - x \rangle - \frac{\ell}{2} \|x - y\|^2$$

$$\geq f(x) + \langle \nabla f(x), y - x \rangle - \frac{\ell}{2\lambda_{\min}(\mathbf{M}_t)} \|x - y\|_{\mathbf{M}_t}^2.$$

Correspondingly for the opposite direction:

$$f(y) \leq f(x) + \langle \nabla f(x), y - x \rangle + \frac{\ell}{2} \|x - y\|^2$$

$$\leq f(x) + \langle \nabla f(x), y - x \rangle + \frac{\ell}{2\lambda_{\min}(\mathbf{M}_t)} \|x - y\|_{\mathbf{M}_t}^2.$$

$\square$

#### G.4.2 Convergence of Alternating Descent-Ascent

Through, we consider this section, we consider the iteration following scheme,

$$x_{t+1} = \underset{x \in \mathcal{X}}{\arg\min} \left\{ \langle \nabla f(x_t, y_t), x - x_t \rangle + \frac{1}{2\eta_x} \|x - x_t\|_{\mathbf{M}_{x,t}}^2 \right\};$$

$$y_{t+1} = \underset{y \in \mathcal{Y}}{\arg\min} \left\{ \langle -\nabla f(x_{t+1}, y_t), y - y_t \rangle + \frac{1}{2\eta_y} \|y - y_t\|_{\mathbf{M}_{y,t}}^2 \right\}. \tag{Alt-GDA}$$

We make a standard assumption on the gradient estimators and their second moments.

**Assumption 3** (Unbiased Gradient Estimators and Bounded Second Moments)**.** *For all iterations $t$, the gradient estimators $\hat{g}_x(x_t, y_t)$ and $\hat{g}_y(x_t, y_t)$ satisfy*

$$\mathbb{E}\left[ \hat{g}_x(x_t, y_t) \right] = g_x(x_t, y_t),$$

$$\mathbb{E}\left[ \hat{g}_y(x_t, y_t) \right] = g_y(x_t, y_t),$$

*and*

$$\mathbb{E}\left[ \|\hat{g}_x(x_t, y_t)\|^2 \right] \leq \sigma_x^2,$$

$$\mathbb{E}\left[ \|\hat{g}_y(x_t, y_t)\|^2 \right] \leq \sigma_y^2.$$

*In turn, $\|g_x(x_t, y_t) - \nabla_x f(x_t, y_t)\| \leq \delta_x$, $\|g_y(x_t, y_t) - \nabla_y f(x_t, y_t)\| \leq \delta_y$.*

**Theorem G.1.** *Let $f : \mathcal{X} \times \mathcal{Y} \to \mathbb{R}$ an $\ell$-smooth function and bounded in the interval $\Delta_f$. Further, assume $\mathcal{X}, \mathcal{Y}$ to be two convex sets with Euclidean diameters,* $\operatorname{diam}(\mathcal{X}), \operatorname{diam}(\mathcal{Y})$. *Moreover, assume that $f$ satisfies a two-sided pPŁ condition with moduli $\alpha_x$ for all $y \in \mathcal{Y}$ and $\alpha_y$ for any $x \in \mathcal{X}$. Additionally, let $(\hat{g}_x, \hat{g}_y)$ be an inexact stochastic gradient oracle satisfying Assumption 3.*

- *When $\mathbf{M}_{\cdot t} = \mathbf{I}$, after $T$ iterations of* (Alt-GDA) *with a choice of stepsizes $\eta_x = \frac{\alpha_y^2}{960\ell^3}$ and $\eta_y = \frac{1}{5\ell}$, it holds true that:*

$$\mathbb{E}\Phi(x_T) - \Phi^\star + \tfrac{1}{10}\left(\mathbb{E}\Phi(x_T) - \mathbb{E}f(x_T, y_T)\right)$$

$$\leq \exp\left(-\frac{\alpha_x \alpha_y^2}{960\ell^3}T\right)\Delta_f + \frac{c_1\sigma_x^2}{\alpha_x} + \frac{c_1\delta_x^2}{\alpha_x} + \frac{c_2\ell^2\sigma_y^2}{\alpha_x\alpha_y^2} + \frac{c_2\ell^2\delta_y^2}{\alpha_x\alpha_y^2},$$

  *where, $\Delta_f := \max_{x \in \mathcal{X}, y \in \mathcal{Y}} f(x, y) - \min_{x \in \mathcal{X}, y \in \mathcal{Y}} f(x, y)$ and $c_1, c_2 \in O(1)$.*

- *For a general positive definite choice of $\mathbf{M}_{\cdot t}$ (Mahalanobis metric), after $T$ iterations of* (Alt-GDA) *with a choice of stepsizes $\eta_x = \frac{\alpha_y^2}{960\ell^3\lambda_{\max}^2}$ and $\eta_y = \frac{1}{5\ell\lambda_{\max}}$, it holds true that:*

$$\mathbb{E}\Phi(x_T) - \Phi^\star + \tfrac{1}{10}\left(\mathbb{E}\Phi(x_T) - \mathbb{E}f(x_T, y_T)\right)$$

$$\leq \exp\left(-\frac{\alpha_x \alpha_y^2}{960\lambda_{\max}^2\ell^3}T\right)\Delta_f + \frac{c_1\sigma_x^2}{\alpha_x} + \frac{c_1\delta_x^2}{\alpha_x} + \frac{c_2\ell^2\lambda_{\max}\sigma_y^2}{\alpha_x\alpha_y^2} + \frac{c_2\ell^2\lambda_{\max}\delta_y^2}{\alpha_x\alpha_y^2},$$

  *where, $\Delta_f := \max_{x \in \mathcal{X}, y \in \mathcal{Y}} f(x, y) - \min_{x \in \mathcal{X}, y \in \mathcal{Y}} f(x, y)$, $\lambda_{\max} := \max_t \lambda_{\max}(\mathbf{M}_{\cdot, t}^{-1})$ and $c_1, c_2 \in O(1)$.*

*Proof.* To prove convergence we will use the Lyapunov function $L(x, y) := U(x, y) + cW(x, y)$ with $U(x, y) := \mathbb{E}\left[\Phi(x) - \Phi^\star\right]$, $W(x, y) := \mathbb{E}\left[\Phi(x) - f(x, y)\right]$ and $c > 0$. Intuitively, $U(x, y)$ measures $x$'s success in achieving the unique minmax value $\Phi^\star$, while $W(x, y)$ measures $y$'s success in achieving to be a best-response to its corresponding $x$. We begin with some preliminary work to ultimately setup a recursion on $L$.

**Descent on $\Phi$**  In order to guarantee descent, by Lemma G.6, Proposition 7, and Lemma G.4, it suffices to pick $\eta_x \leq \frac{1}{5\ell\lambda_{\max}(\mathbf{M}_{x,t})}$. Then, we can write,

$$\mathbb{E}\Phi(x_{t+1}) \leq \mathbb{E}\Phi(x_t) - \frac{\eta_x}{6}\mathbb{E}\mathcal{D}_{\mathcal{X}}^\Phi(x_t, 1/\eta_x) + \eta_x\mathbb{E}\left\|\nabla_x\Phi(x_t) - \nabla_x f(x_t, y_t)\right\|_{\mathbf{M}_{x,t}^{-1}}^2$$

$$+ 2\eta_x\sigma_x^2 + 2\eta_x\delta_x^2.$$

Equivalently, subtracting $\Phi^\star$ from both sides yields,

$$\mathbb{E}\Phi(x_{t+1}) - \Phi^\star \leq \mathbb{E}\Phi(x_t) - \Phi^\star - \frac{\eta_x}{6}\mathbb{E}\mathcal{D}_{\mathcal{X}}^\Phi(x_t, 1/\eta_x) + \eta_x\mathbb{E}\left\|\nabla_x\Phi(x_t) - \nabla_x f(x_t, y_t)\right\|_{\mathbf{M}_{x,t}^{-1}}^2$$

$$+ 2\eta_x\sigma_x^2 + 2\eta_x\delta_x^2.$$

Further, a simple re-arrangement reads,

$$\mathbb{E}\Phi(x_{t+1}) - \mathbb{E}\Phi(x_t) \leq -\frac{\eta_x}{6}\mathbb{E}\mathcal{D}_{\mathcal{X}}^\Phi(x_t, 1/\eta_x) + \eta_x\mathbb{E}\left\|\nabla_x\Phi(x_t) - \nabla_x f(x_t, y_t)\right\|_{\mathbf{M}_{x,t}^{-1}}^2$$

$$+ 2\eta_x\sigma_x^2 + 2\eta_x\delta_x^2.$$

**Ascent on $f(x, \cdot)$**  Requiring that $\eta_y \leq \frac{1}{5\ell\lambda_{\max}(\mathbf{M}_{y,t})}$, (Proposition 7 and Lemma G.4), we write:

$$\mathbb{E}f(x_{t+1}, y_{t+1}) \geq \mathbb{E}f(x_{t+1}, y_t) + \frac{\eta_y}{6}\mathbb{E}\mathcal{D}_{\mathcal{Y}}(y_t, 1/\eta_y; x_{t+1}) - \eta_y\delta^2 - \eta_y\sigma_y^2$$

Invoking Lemma G.8, multiplying by $-1$, and adding $\Phi(x_{t+1})$ will yield,

$$\mathbb{E}\left[\Phi(x_{t+1}) - f(x_{t+1}, y_{t+1})\right] \leq \left(1 - \frac{\alpha_y\eta_y}{6}\right)\mathbb{E}\left[\Phi(x_{t+1}) - f(x_{t+1}, y_t)\right] + \eta_y\delta^2 + \eta_y\sigma_y^2$$

$$= \left(1 - \frac{\alpha_y\eta_y}{6}\right)\mathbb{E}\left[\Phi(x_t) - f(x_t, y_t) + f(x_t, y_t) - f(x_{t+1}, y_t) + \Phi(x_{t+1}) - \Phi(x_t)\right]$$

$$+ \eta_y\delta^2 + \eta_y\sigma_y^2.$$

As a reminder, $\Phi$ is a pPL function relative to the Mahalanobis distance induced by $\mathbf{M}_t$ by Lemma G.8.

**Upper bound on the descent of $f(\cdot, y)$**    From the smoothnes of $f$:

$$\mathbb{E}f(x_{t+1}, y_t) \geq \mathbb{E}f(x_t, y_t) - \frac{3\eta_x}{2}\mathbb{E}\left\|G_{1/\eta_x}(x_t)\right\|_{\mathbf{M}_{x,t}^{-1}}^2 - \frac{9\eta_x\sigma_x^2}{2} - \frac{7\eta_x\delta_x^2}{2}$$

$$\geq \mathbb{E}f(x_t, y_t) - \frac{3\eta_x}{2}\mathbb{E}\mathcal{D}_{\mathcal{X}}(x_t, 1/\eta_x; y_t) - \frac{9\eta_x\sigma_x^2}{2} - \frac{7\eta_x\delta_x^2}{2}$$

Re-arranging to isolate $f(x_t, y_t) - f(x_{t+1}, y_t)$,

$$\mathbb{E}f(x_t, y_t) - \mathbb{E}f(x_{t+1}, y_t) \leq \frac{3\eta_x}{2}\mathbb{E}\mathcal{D}_{\mathcal{X}}(x_t, 1/\eta_x; y_t) + \frac{9\eta_x\sigma_x^2}{2} + \frac{7\eta_x\delta_x^2}{2}.$$

Putting the pieces together for $\Phi(x_t) - f(x_t, y_t)$, we get:

$$\mathbb{E}\left[\Phi(x_{t+1}) - f(x_{t+1}, y_{t+1})\right]$$
$$\leq \left(1 - \frac{\alpha_y\eta_y}{6}\right)\mathbb{E}\left[\Phi(x_t) - f(x_t, y_t)\right]$$
$$+ \left(1 - \frac{\alpha_y\eta_y}{6}\right)\mathbb{E}\left[-\frac{\eta_x}{6}\mathbb{E}\mathcal{D}_{\mathcal{X}}^{\Phi}(x_t, 1/\eta_x) + \eta_x\mathbb{E}\left\|\nabla_x\Phi(x_t) - \nabla_x f(x_t, y_t)\right\|_{\mathbf{M}_{x,t}^{-1}}^2\right]$$
$$+ \left(1 - \frac{\alpha_y\eta_y}{6}\right)\mathbb{E}\left[\frac{3\eta_x}{2}\mathbb{E}\mathcal{D}_{\mathcal{X}}(x_t, 1/\eta_x; y_t)\right]$$
$$+ \eta_y\delta_y^2 + \eta_y\sigma_y^2 + \eta_x\left(1 - \frac{\alpha_y\eta_y}{6}\right)\left(\frac{13}{2}\sigma_x^2 + \frac{11}{2}\delta_x^2\right)$$

**Decrease in the Lyapunov function**    We consider the Lyapunov function $L(x, y) := U(x, y) + cW(x, y)$ with $U(x, y) := \mathbb{E}\left[\Phi(x) - \Phi^\star\right]$, $W(x, y) := \mathbb{E}\left[\Phi(x) - f(x, y)\right]$ and shorthand notation $U_t = U(x_t, y_t), W_t = W(x_t, y_t)$. Here $U_t$ measures primal suboptimality via the PL condition on $\Phi$, while $W_t$ captures the dual gap $\Phi(x_t) - f(x_t, y_t)$.

$$U_{t+1} + cW_{t+1}$$
$$\leq U_t - \frac{\eta_x}{6}\mathbb{E}\mathcal{D}_{\mathcal{X}}^{\Phi}(x_t, 1/\eta_x) + \eta_x\mathbb{E}\left\|\nabla_x\Phi(x_t) - \nabla_x f(x_t, y_t)\right\|_{\mathbf{M}_{x,t}^{-1}}^2$$
$$+ c\left(1 - \frac{\alpha_y\eta_y}{6}\right)\mathbb{E}W_t$$
$$+ c\left(1 - \frac{\alpha_y\eta_y}{6}\right)\mathbb{E}\left[-\frac{\eta_x}{6}\mathbb{E}\mathcal{D}_{\mathcal{X}}^{\Phi}(x_t, 1/\eta_x) + \eta_x\mathbb{E}\left\|\nabla_x\Phi(x_t) - \nabla_x f(x_t, y_t)\right\|_{\mathbf{M}_{x,t}^{-1}}^2\right]$$
$$+ c\left(1 - \frac{\alpha_y\eta_y}{6}\right)\frac{3\eta_x}{2}\mathbb{E}\left[\mathcal{D}_{\mathcal{X}}(x_t, 1/\eta_x; y_t)\right]$$
$$+ c\eta_y\delta_y^2 + c\eta_y\sigma_y^2 + c\eta_x\left(1 - \frac{\alpha_y\eta_y}{6}\right)\left(\frac{13}{2}\sigma_x^2 + \frac{11}{2}\delta_x^2\right) + 2\eta_x\sigma_x^2 + 2\eta_x\delta_x^2$$
$$\leq U_t - \frac{\eta_x}{6}\mathbb{E}\mathcal{D}_{\mathcal{X}}^{\Phi}(x_t, 1/\eta_x) + \eta_x\mathbb{E}\left\|\nabla_x\Phi(x_t) - \nabla_x f(x_t, y_t)\right\|_{\mathbf{M}_{x,t}^{-1}}^2$$
$$+ c\left(1 - \frac{\alpha_y\eta_y}{6}\right)\mathbb{E}W_t$$
$$+ c\left(1 - \frac{\alpha_y\eta_y}{6}\right)\mathbb{E}\left[-\frac{\eta_x}{6}\mathbb{E}\mathcal{D}_{\mathcal{X}}^{\Phi}(x_t, 1/\eta_x) + \eta_x\mathbb{E}\left\|\nabla_x\Phi(x_t) - \nabla_x f(x_t, y_t)\right\|_{\mathbf{M}_{x,t}^{-1}}^2\right]$$
$$+ c\left(1 - \frac{\alpha_y\eta_y}{6}\right)\frac{3\eta_x}{2}\mathbb{E}\left[|\mathcal{D}_{\mathcal{X}}(x_t, 1/\eta_x; y_t) - \mathcal{D}_{\mathcal{X}}^{\Phi}(x_t, 1/\eta_x)| + \mathcal{D}_{\mathcal{X}}^{\Phi}(x_t, 1/\eta_x)\right] \quad (17)$$
$$+ c\eta_y\delta_y^2 + c\eta_y\sigma_y^2 + c\eta_x\left(1 - \frac{\alpha_y\eta_y}{6}\right)\left(\frac{13}{2}\sigma_x^2 + \frac{11}{2}\delta_x^2\right) + 2\eta_x\sigma_x^2 + 2\eta_x\delta_x^2$$
$$\leq U_t - \frac{\eta_x}{6}\mathbb{E}\mathcal{D}_{\mathcal{X}}^{\Phi}(x_t, 1/\eta_x) + \eta_x\mathbb{E}\left\|\nabla_x\Phi(x_t) - \nabla_x f(x_t, y_t)\right\|_{\mathbf{M}_{x,t}^{-1}}^2$$
$$+ c\left(1 - \frac{\alpha_y\eta_y}{6}\right)\mathbb{E}W_t$$
$$+ c\left(1 - \frac{\alpha_y\eta_y}{6}\right)\mathbb{E}\left[-\frac{\eta_x}{6}\mathbb{E}\mathcal{D}_{\mathcal{X}}^{\Phi}(x_t, 1/\eta_x) + \eta_x\mathbb{E}\left\|\nabla_x\Phi(x_t) - \nabla_x f(x_t, y_t)\right\|_{\mathbf{M}_{x,t}^{-1}}^2\right]$$
$$+ c\left(1 - \frac{\alpha_y\eta_y}{6}\right)\frac{3\eta_x}{2}\mathbb{E}\left[3\lambda_{\max}(\mathbf{M}_{x,t}^{-1})\ell^2\left\|y_t - y^\star(x_t)\right\|^2 + \mathcal{D}_{\mathcal{X}}^{\Phi}(x_t, 1/\eta_x)\right] \quad (18)$$
$$+ c\eta_y\delta_y^2 + c\eta_y\sigma_y^2 + c\eta_x\left(1 - \frac{\alpha_y\eta_y}{6}\right)\left(\frac{13}{2}\sigma_x^2 + \frac{11}{2}\delta_x^2\right) + 2\eta_x\sigma_x^2 + 2\eta_x\delta_x^2$$

$$
\leq U_t - \frac{\eta_x}{6}\mathbb{E}\mathcal{D}_\mathcal{X}^\Phi(x_t, 1/\eta_x) + \eta_x\lambda_{\max}(\mathbf{M}_{x,t}^{-1})\ell^2\mathbb{E}\left\|y^\star(x_t) - y_t\right\|^2
$$
$$
+ c\left(1 - \frac{\alpha_y\eta_y}{6}\right)\mathbb{E}W_t
$$
$$
+ c\left(1 - \frac{\alpha_y\eta_y}{6}\right)\mathbb{E}\left[-\frac{\eta_x}{6}\mathcal{D}_\mathcal{X}^\Phi(x_t, 1/\eta_x) + \eta_x\lambda_{\max}(\mathbf{M}_{x,t}^{-1})\ell^2\left\|y^\star(x_t) - y_t\right\|^2\right]
$$
$$
+ c\left(1 - \frac{\alpha_y\eta_y}{6}\right)\frac{3\eta_x}{2}\mathbb{E}\left[3\lambda_{\max}(\mathbf{M}_{x,t}^{-1})\ell^2\left\|y_t - y^\star(x_t)\right\|^2 + \mathcal{D}_\mathcal{X}^\Phi(x_t, 1/\eta_x)\right]
$$
$$
+ c\eta_y\delta_y^2 + c\eta_y\sigma_y^2 + c\eta_x\left(1 - \frac{\alpha_y\eta_y}{6}\right)\left(\frac{13}{2}\sigma_x^2 + \frac{11}{2}\delta_x^2\right) + 2\eta_x\sigma_x^2 + 2\eta_x\delta_x^2
$$

- (17) uses the fact that $a \leq |a - b| + b$ for $a = \mathcal{D}_\mathcal{X}(x_t, 1/\eta_x; y_t), b = \mathcal{D}_\mathcal{X}^\Phi(x_t, 1/\eta_x)$. This decomposition isolates the term $|\mathcal{D}_\mathcal{X} - \mathcal{D}_\mathcal{X}^\Phi|$, which can then be controlled using the Mahalanobis continuity lemma in $y$.

- (18) uses Lemma G.5 and Danskin's theorem; this yields a bound $|\mathcal{D}_\mathcal{X} - \mathcal{D}_\mathcal{X}^\Phi| \leq 3\lambda_{\max}(\mathbf{M}_{x,t}^{-1})\ell^2\|y_t - y^\star(x_t)\|^2$.

$$
U_{t+1} + cW_{t+1} \leq U_t - \frac{\eta_x}{6}\mathbb{E}\mathcal{D}_\mathcal{X}^\Phi(x_t, 1/\eta_x) + \frac{2\eta_x\lambda_{\max}(\mathbf{M}_{x,t}^{-1})\ell^2}{\alpha_{\mathrm{qg}}}W_t
$$
$$
+ c\left(1 - \frac{\alpha_y\eta_y}{6}\right)\mathbb{E}W_t
$$
$$
+ c\left(1 - \frac{\alpha_y\eta_y}{6}\right)\mathbb{E}\left[-\frac{\eta_x}{6}\mathbb{E}\mathcal{D}_\mathcal{X}^\Phi(x_t, 1/\eta_x) + \frac{2\eta_x\lambda_{\max}(\mathbf{M}_{x,t}^{-1})\ell^2}{\alpha_{\mathrm{qg}}}W_t\right]
$$
$$
+ c\left(1 - \frac{\alpha_y\eta_y}{6}\right)\frac{3\eta_x}{2}\mathbb{E}\left[\frac{6\lambda_{\max}(\mathbf{M}_{x,t}^{-1})\ell^2}{\alpha_{\mathrm{qg}}}W_t\right]
$$
$$
+ c\eta_y\delta_y^2 + c\eta_y\sigma_y^2 + c\eta_x\left(1 - \frac{\alpha_y\eta_y}{6}\right)\left(\frac{13}{2}\sigma_x^2 + \frac{11}{2}\delta_x^2\right) + 2\eta_x\sigma_x^2 + 2\eta_x\delta_x^2
$$
$$
\leq \varpi_1 U_t + c\varpi_2 W_t
$$
$$
+ c\eta_y\delta_y^2 + c\eta_y\sigma_y^2 + c\eta_x\left(1 - \frac{\alpha_y\eta_y}{3}\right)\left(\frac{13}{2}\sigma_x^2 + \frac{11}{2}\delta_x^2\right) + 2\eta_x\sigma_x^2 + 2\eta_x\delta_x^2
$$

We then collect the coefficients in front of $U_t$ and $W_t$ in the previous inequality into $\varpi_1$ and $\varpi_2$, respectively, so that the Lyapunov recursion can be written compactly as $U_{t+1} + cW_{t+1} \leq \varpi_1 U_t + c\varpi_2 W_t + \text{noise}$. *I.e.*,

$$
\varpi_1 := 1 - \alpha_x\eta_x\left(\frac{1}{3} - c\left(1 - \frac{\alpha_y\eta_y}{6}\right)\frac{1}{3} + c\left(1 - \frac{\alpha_y\eta_y}{6}\right)3\right);
$$
$$
\varpi_2 := 1 + \frac{2\eta_x\lambda_{\max}(\mathbf{M}_{x,t}^{-1})\ell^2}{c\alpha_{\mathrm{qg}}} - \frac{\alpha_y\eta_y}{6} + \left(1 - \frac{\alpha_y\eta_y}{6}\right)\frac{11\eta_x\lambda_{\max}(\mathbf{M}_{x,t}^{-1})\ell^2}{\alpha_{\mathrm{qg}}}.
$$

For $\varpi_1$, letting $c = 1/10$

$$
\varpi_1 = 1 - \alpha_x\eta_x\left(\frac{1}{3} - \frac{1}{10}\left(1 - \frac{\alpha_y\eta_y}{6}\right)\frac{1}{3} + \frac{1}{10}\left(1 - \frac{\alpha_y\eta_y}{6}\right)3\right)
$$
$$
= 1 - \alpha_x\eta_x\frac{1}{3} - \alpha_x\eta_x\frac{8}{30}\left(1 - \frac{\alpha_y\eta_y}{6}\right) \leq 1 - \frac{\alpha_x\eta_x}{3}.
$$

For $\varpi_2$, we distinguish two cases relevant to our algorithms, $\mathbf{M}_t = \mathbf{I}$ and a general choice of $\mathbf{M}_t$.

- For $\mathbf{M}_t = \mathbf{I}$, it holds that $\lambda_{\max}\left(\mathbf{M}_{\cdot,t}^{-1}\right) = 1$, and $\alpha_{\mathrm{qg}} = \alpha_y$. So we write

$$
\varpi_2 = 1 + \frac{20\eta_x\ell^2}{\alpha_y} - \frac{\alpha_y\eta_y}{6} + \left(1 - \frac{\alpha_y\eta_y}{6}\right)\frac{11\eta_x\ell^2}{\alpha_y}
$$
$$
= 1 - \frac{\eta_x\ell^2}{\alpha_y}\left(-20 + \frac{\alpha_y^2\eta_y}{6\eta_x\ell^2} - 11\left(1 - \frac{\alpha_y\eta_y}{6}\right)\right)
$$

$$\leq 1 - \frac{\eta_x \ell^2}{\alpha_y} (-20 + 32 - 11)$$

Let $\frac{\alpha_y^2 \eta_y}{\eta_x \ell^2} = 192$. Then, choosing $\eta_y = \frac{1}{5\ell}$ yields $\eta_x = \frac{\alpha_y^2}{960\ell^3}$.

- For a general choice of $\mathbf{M}_t$, let $\lambda_{\max} := \max\{\lambda_{\max}(\mathbf{M}_{x,t}^{-1}), \lambda_{\max}(\mathbf{M}_{y,t}^{-1})\}$ and $\overline{\alpha_y} \leftarrow \min\{\alpha_{qg}, \alpha_y\}$,

$$\varpi_2 = 1 + \frac{20\eta_x \lambda_{\max}\ell^2}{\overline{\alpha_y}} - \frac{\overline{\alpha_y}\eta_y}{6} + \left(1 - \frac{\overline{\alpha_y}\eta_y}{6}\right) \frac{11\eta_x \lambda_{\max}\ell^2}{\overline{\alpha_y}}$$

$$= 1 - \frac{\lambda_{\max}\eta_x \ell^2}{\overline{\alpha_y}} \left(-20 + \frac{\overline{\alpha_y}^2 \eta_y}{6\lambda_{\max}\eta_x \ell^2} - 11\left(1 - \frac{\overline{\alpha_y}\,\eta_y}{6}\right)\right).$$

Similarly, we need to set

$$\frac{\overline{\alpha_y}^2 \eta_y}{\lambda_{\max}\eta_x \ell^2} = 192.$$

This in turn yields $\eta_y = \frac{1}{5\lambda_{\max}\ell}$ and $\eta_x = \frac{\overline{\alpha_y}^2}{960\ell^3\lambda_{\max}^2}$.

**Remark 4.** *In fact, $\mathbf{M}_t$ is allowed to be positive semidefinite as long as the gradient throuhgout the iterations is in the kernel of $\mathbf{M}_t$.*

$\square$

# H Convergence Analysis

## H.1 Direct Policy Parametrization

**Theorem H.1.** *With direct policy parametrization and the Euclidean bidilated regularizer, alternating policy-gradient algorithm attains a last-iterate $\epsilon$-Nash equilibrium in*

$$T = \frac{1}{\epsilon^{12}}\mathsf{poly}\left(\frac{1}{\gamma}, |\mathcal{H}|, A, B, 2^{D(\mathcal{T})}, \frac{1}{\min_h \mu_c(h)}, |\mathcal{S}_1|, |\mathcal{S}_2|\right) \text{ iterations,}$$

*using batches of* $\mathsf{poly}\left(\frac{1}{\epsilon}, \frac{1}{\gamma}, |\mathcal{H}|, A, B, 2^{D(\mathcal{T})}, \frac{1}{\min_h \mu_c(h)}, |\mathcal{S}_1|, |\mathcal{S}_2|\right)$ *trajectory samples at each step.*

*Proof.* The proof follows as an application of Theorem G.1. In a central role lies Lemma E.1, which provides a two-sided pPŁ condition for the regularized game under direct policy parametrization, while in a supportive one the smoothness lemmata of the value function and the Euclidean bidilated regularizer when the policy is directly parametrized.

First, we relate equilibria of the regularized, truncated, exploration-perturbed game to equilibria of the original game. An $\epsilon$-NE of the regularized game is an $\epsilon'$-NE of the unregularized game where

$$\epsilon' = O\left(\epsilon + \tau S 2^{D(\mathcal{T})} + \varepsilon S \max\{A, B\} + \gamma\right).$$

The term contains the optimization error $\epsilon$, the regularization error (controlled by $\tau$), the truncation error (controlled by $\varepsilon$ through the minimum action probability), and the exploration-induced error (controlled by $\gamma$). To make each contribution $O(\epsilon)$ we choose

- $\gamma = \Theta(\epsilon)$,

- $\tau = \Theta\left(\frac{\epsilon}{\max_{i \in \{1,2\}} |\mathcal{S}_i| 2^{D(\mathcal{T})}}\right)$,

- $\varepsilon = \Theta\left(\frac{\epsilon}{\max_{i \in \{1,2\}} |\mathcal{S}_i| \max\{A, B\}}\right).$

We now instantiate Theorem G.1. By Lemma E.1 the utility of the regularized game satisfies the two-sided pPŁ condition with moduli

$$\alpha_x, \alpha_y \;=\; \Theta\!\left(\frac{\tau \min_{h\in\mathcal{H}} \mu_c(h)\,\gamma^3}{|\mathcal{H}|^3}\right).$$

Combining the smoothness of the value function with that of the Euclidean bidilated regularizer (Lemmata B.4 and B.7 ) yields an overall smoothness constant

$$\ell = \Theta\!\Big( \max_{i\in\{1,2\}} \sqrt{|\Sigma_i|}\, D(\mathcal{T}) + \tau\, 2^{D(\mathcal{T})} \max_{i\in\{1,2\}} |\Sigma_i|\, D(\mathcal{T}) \max\{|\mathcal{S}_1|,|\mathcal{S}_2|\}\Big)$$
$$= O\!\Big( D(\mathcal{T}) \max_{i\in\{1,2\}} |\Sigma_i| \Big),$$

The stochastic gradients used by `Alt-RegPG` are given by the REINFORCE estimator together with the gradient estimators for the bidilated regularizer; by Lemma F.1 and the analysis of Appendix F.1 they are unbiased and have bounded per-trajectory variance

$$\mathbb{E}\big\|\widehat{\nabla}_x^{(1)} - \nabla_x V\big\|^2 \le \frac{A^2 D(\mathcal{T})^2}{\varepsilon}, \qquad\qquad \mathbb{E}\big\|\widehat{\nabla}_y^{(1)} - \nabla_y V\big\|^2 \le \frac{B^2 D(\mathcal{T})^2}{\varepsilon}.$$

If each update averages a mini-batch of $M$ i.i.d. trajectories, $\widehat{\nabla}_x = \frac{1}{M}\sum_{m=1}^M \widehat{\nabla}_x^{(m)}$ and $\widehat{\nabla}_y = \frac{1}{M}\sum_{m=1}^M \widehat{\nabla}_y^{(m)}$, then the averaged estimators have variances

$$\mathrm{Var}(\widehat{\nabla}_x) \;\le\; \frac{\sigma_x^2}{M}, \qquad \mathrm{Var}(\widehat{\nabla}_y) \;\le\; \frac{\sigma_y^2}{M},$$

with per-trajectory bounds $\sigma_x^2 \le A^2 D(\mathcal{T})^2/\varepsilon$ and $\sigma_y^2 \le B^2 D(\mathcal{T})^2/\varepsilon$. Substituting these into Theorem G.1, the stochastic error terms are controlled (up to absolute constants) by $\sigma_x^2/(M\alpha_x)$ and $\ell\,\sigma_y^2/(M\alpha_x\alpha_y^2)$. Requiring each to be at most $\epsilon$ leads to the condition

$$M \;\ge\; \max\!\left\{\frac{\sigma_x^2}{\epsilon\alpha_x}, \frac{\ell\,\sigma_y^2}{\epsilon\alpha_x\alpha_y^2}\right\} = \max\!\left\{\frac{A^2 D(\mathcal{T})^2}{\epsilon\,\varepsilon\,\alpha_x}, \frac{\ell\,B^2 D(\mathcal{T})^2}{\epsilon\,\varepsilon\,\alpha_x\alpha_y^2}\right\}.$$

Using the explicit forms of $\alpha_x, \alpha_y$ from Lemma E.1 and the per-trajectory variance bounds from Lemma F.1, this can be summarized as choosing

$$M = \Theta\!\left(\max\left\{\frac{1}{\epsilon\,\varepsilon\,\tau\,\gamma^3}, \frac{\ell}{\epsilon\,\varepsilon\,\tau^3\gamma^9}\right\}\right).$$

Writing $S := \max\{|\mathcal{S}_1|,|\mathcal{S}_2|\}$ and using the tunings $\gamma = \Theta(\epsilon)$, $\tau = \Theta\big(\epsilon/(S2^{D(\mathcal{T})})\big)$, and $\varepsilon = \Theta\big(\epsilon/(SA)\big)$ from above, together with

$$\ell = \Theta\!\Big( D(\mathcal{T}) \max\Big\{ \sqrt{\max_{i\in\{1,2\}} |\Sigma_i|}, \epsilon \max_{i\in\{1,2\}} |\Sigma_i| \Big\} \Big),$$

$$\alpha_x = \alpha_y = \Theta\!\Big(\frac{\tau\gamma^3 \min_h \mu_c(h)}{|\mathcal{H}|^3}\Big) = \Theta\!\Big(\frac{\min_h \mu_c(h)}{S2^{D(\mathcal{T})}|\mathcal{H}|^3}\,\epsilon^4\Big),$$

a direct substitution yields the explicit bounds

$$M \ge \Theta\!\left(\frac{2^{D(\mathcal{T})} D(\mathcal{T})^2 S^2 A^3 |\mathcal{H}|^3}{\min_h \mu_c(h)\,\epsilon^6}\right),$$

$$M \ge \Theta\!\left(\frac{2^{3D(\mathcal{T})} D(\mathcal{T})^3 S^4 A\,B^2 |\mathcal{H}|^9 \max\big\{\sqrt{\max_{i\in\{1,2\}} |\Sigma_i|}, \epsilon \max_{i\in\{1,2\}} |\Sigma_i|\big\}}{\big(\min_h \mu_c(h)\big)^3 \epsilon^{14}}\right).$$

For small $\epsilon$ the second constraint dominates, so it is sufficient to choose

$$M = \Theta\!\left(\frac{2^{3D(\mathcal{T})} D(\mathcal{T})^3 S^4 A\,B^2 |\mathcal{H}|^9 \max\big\{\sqrt{\max_{i\in\{1,2\}} |\Sigma_i|}, \epsilon \max_{i\in\{1,2\}} |\Sigma_i|\big\}}{\big(\min_h \mu_c(h)\big)^3 \epsilon^{14}}\right),$$

which spells out the precise dependence of the mini-batch size on $\epsilon$, $A$, $B$, $D(\mathcal{T})$, $|\mathcal{S}_1|$, $|\mathcal{S}_2|$, $|\mathcal{H}|$, and $\min_h \mu_c(h)$.

Under these conditions, Theorem G.1 prescribes the concrete stepsizes

$$\eta_y = \frac{1}{5\ell}, \quad \text{and} \quad \eta_x = \frac{\alpha_y^2}{960\,\ell^3} = \frac{\tau^2\gamma^6(\min_{h\in\mathcal{H}}\mu_c(h))^2}{960\cdot 101^2\,|\mathcal{H}|^6\,\ell^3},$$

owing to the symmetric pPŁ moduli $\alpha_x = \alpha_y$ from Lemma E.1. The resulting duality-gap decay is $\exp\left(-\frac{\alpha_x\alpha_y^2}{960\ell^3}T\right)$, so driving the deterministic term below $\epsilon$ requires

$$T = \frac{960\,\ell^3}{\alpha_x\alpha_y^2}\log\frac{\Delta_f}{\epsilon} = \frac{960\cdot 101^3\,|\mathcal{H}|^9\,\ell^3}{\tau^3\gamma^9\,(\min_{h\in\mathcal{H}}\mu_c(h))^3}\log\frac{\Delta_f}{\epsilon},$$

where $\Delta_f$ is the payoff range appearing in Theorem G.1. Substituting the smoothness estimate from Corollary B.1 and Lemma B.7,

$$\ell = \Theta\left(D(\mathcal{T})\max\left\{\sqrt{\max_{i\in\{1,2\}}|\Sigma_i|}, \epsilon\max_{i\in\{1,2\}}|\Sigma_i|\right\}\right)$$

yields the following dependencies on the game parameters:

- $\eta_y = \Theta\left(\dfrac{1}{D(\mathcal{T})\max\{\sqrt{\max_{i\in\{1,2\}}|\Sigma_i|}, \epsilon\max_{i\in\{1,2\}}|\Sigma_i|\}}\right)$;

- $\eta_x = \Theta\left(\dfrac{\epsilon^8(\min_{h\in\mathcal{H}}\mu_c(h))^2}{2^{2D(\mathcal{T})}\,S^2\,|\mathcal{H}|^6\,D(\mathcal{T})^3\,\max\{\sqrt{\max_{i\in\{1,2\}}|\Sigma_i|}, \epsilon\max_{i\in\{1,2\}}|\Sigma_i|\}^3}\right)$;

- $T = \Theta\left(\dfrac{2^{3D(\mathcal{T})}\,S^3\,|\mathcal{H}|^9\,D(\mathcal{T})^3\,\max\{\sqrt{\max_{i\in\{1,2\}}|\Sigma_i|}, \epsilon\max_{i\in\{1,2\}}|\Sigma_i|\}^3}{(\min_{h\in\mathcal{H}}\mu_c(h))^3\,\epsilon^{12}}\log\dfrac{\Delta_f}{\epsilon}\right)$.

Finally, substituting the choices of $\gamma, \tau, \varepsilon$ from above into the expression for $T$ yields

$$T = \Theta\left(\frac{2^{3D(\mathcal{T})}\,S^3\,|\mathcal{H}|^9\,D(\mathcal{T})^3\,\max\{\sqrt{\max_{i\in\{1,2\}}|\Sigma_i|}, \epsilon\max_{i\in\{1,2\}}|\Sigma_i|\}^3}{(\min_{h\in\mathcal{H}}\mu_c(h))^3\,\epsilon^{12}}\log\frac{\Delta_f}{\epsilon}\right),$$

as claimed in the statement of the theorem. $\qquad\square$

## H.2 Softmax Policy Parametrization

**Theorem H.2.** *Alternating policy-gradient algorithm with softmax policy parametrization and the entropic bidilated regularizer converges in expectation in the last-iterate to an $\epsilon$-Nash equilibrium after a number of iterations $T$ given by*

$$T = \frac{1}{\epsilon^{18}}\,\mathsf{poly}\left(|\mathcal{H}|, A, B, 2^{D(\mathcal{T})}, \tfrac{1}{\min_{h\in\mathcal{H}}\mu_c(h)}, |\mathcal{S}_1|, |\mathcal{S}_2|\right),$$

*using batches of $\mathsf{poly}\left(\frac{1}{\epsilon}, \frac{1}{\gamma}, |\mathcal{H}|, A, B, 2^{D(\mathcal{T})}, \frac{1}{\min_h\mu_c(h)}, |\mathcal{S}_1|, |\mathcal{S}_2|\right)$ trajectory samples at each step.*

*Proof.* The theorem follows as a corollary of Theorem G.1. By Lemma E.2, the regularized game under softmax parametrization satisfies the two-sided pPŁ condition with moduli

$$\alpha_x = \Theta\left(\frac{\tau\min_{h\in\mathcal{H}}\mu_c(h)\,\gamma^3}{|\mathcal{H}|^3\big(1+(A-1)e^{2R}\big)^2}\right), \qquad \alpha_y = \Theta\left(\frac{\tau\min_{h\in\mathcal{H}}\mu_c(h)\,\gamma^3}{|\mathcal{H}|^3\big(1+(B-1)e^{2R}\big)^2}\right),$$

up to absolute constants. An $\epsilon$-NE for the regularized game is also an $\epsilon'$-NE for the unregularized game where

$$\epsilon' = O\left(\epsilon + \gamma + \tau S 2^{D(\mathcal{T})}\max\{\log A, \log B\} + \varepsilon S(\max\{A, B\})^2\right).$$

Then, we need to tune:

- $\gamma = \Theta(\epsilon)$;

- $\tau = \Theta\left(\frac{\epsilon}{\max_{i\in\{1,2\}}|\mathcal{S}_i|2^{D(\mathcal{T})}\max\{\log A,\log B\}}\right)$;

- $\varepsilon = \Theta\left(\frac{\epsilon}{\max_{i\in\{1,2\}}|\mathcal{S}_i|(\max\{A,B\})^2}\right)$.

We recall the smoothness parameter of the softmax-parametrized regularized utility function is

$$\ell_{\text{softmax}} = \Theta\Big(16\max_{i\in\{1,2\}}\sqrt{|\Sigma_i|}\,D(\mathcal{T}) + \tau\,2^{D(\mathcal{T})}\max_{i\in\{1,2\}}|\Sigma_i|\,D(\mathcal{T})\,S\,\max\{\log A,\log B\}\Big)$$
$$= O\Big(D(\mathcal{T})\max_{i\in\{1,2\}}|\Sigma_i|\Big),$$

by combining the Lipschitz bounds on the utility and the weighted entropic bidilated regularizer (Lemma B.8). Then, from Theorem G.1 we tune,

$$\eta_y = \Theta\left(\frac{1}{\ell}\right), \qquad \eta_x = \Theta\left(\frac{\alpha_y^2}{\ell^3}\right), \qquad T = \Theta\left(\frac{\ell^3}{\alpha_x\alpha_y^2}\log\frac{1}{\varepsilon}\right),$$

where we set $\ell := \ell_{\text{softmax}}$, and $\alpha_x, \alpha_y$ are the softmax pPŁ moduli of the two players. Invoking Lemma E.2 for player 2 yields

$$\alpha_y = \Theta\left(\frac{\tau\min_{h\in\mathcal{H}}\mu_c(h)\,\gamma^3}{|\mathcal{H}|^3\big(1+(B-1)e^{2R}\big)^2}\right),$$

and therefore, prior to relating $R$ to the truncation level $\varepsilon$,

$$\eta_x = \Theta\left(\frac{\tau^2\left(\min_{h\in\mathcal{H}}\mu_c(h)\right)^2\gamma^6}{|\mathcal{H}|^6\big(1+(B-1)e^{2R}\big)^4\ell^3}\right).$$

Finally, using the explicit relationship between $R$ and the minimum action probability (so that $\big(1+(B-1)e^{2R}\big)^4$ can be expressed as a polynomial in $1/\varepsilon$) and simplifying constants leads to the following convenient. And, subsequently,

- $\eta_y = \Theta\left(\frac{1}{D(\mathcal{T})\max\{\sqrt{\max_{i\in\{1,2\}}|\Sigma_i|},\epsilon\max_{i\in\{1,2\}}|\Sigma_i|\}}\right)$,

- $\eta_x = \Theta\left(\frac{\epsilon^{12}(\min_{h\in\mathcal{H}}\mu_c(h))^2}{2^{2D(\mathcal{T})}S^6|\mathcal{H}|^6 D(\mathcal{T})^3\max\{\sqrt{\max_{i\in\{1,2\}}|\Sigma_i|},\epsilon\max_{i\in\{1,2\}}|\Sigma_i|\}^3\big(\max\{\log A,\log B\}\big)^2\big(\max\{A,B\}\big)^8}\right)$.

Finally, plugging the explicit expressions for $\alpha_x, \alpha_y$ from above into the generic bound $T = \Theta\left(\frac{|\mathcal{H}|^9\ell^3\big(1+(A-1)e^{2R}\big)^2\big(1+(B-1)e^{2R}\big)^4}{\tau^3(\min_{h\in\mathcal{H}}\mu_c(h))^3\gamma^9}\log\frac{1}{\varepsilon}\right)$ yields the precise parameter dependence

$$T = \Theta\left(\frac{|\mathcal{H}|^9\ell^3\big(1+(A-1)e^{2R}\big)^2\big(1+(B-1)e^{2R}\big)^4}{\tau^3\left(\min_{h\in\mathcal{H}}\mu_c(h)\right)^3\gamma^9}\log\frac{1}{\varepsilon}\right).$$

Using the relationship between $R$ and the minimum action probability to upper-bound $\big(1+(A-1)e^{2R}\big)$ and $\big(1+(B-1)e^{2R}\big)$ by polynomials in $1/\varepsilon$ and then substituting the tunings of $\gamma,\tau,\varepsilon$ we obtain an explicit dependence on the game parameters. Writing $S := \max\{|\mathcal{S}_1|,|\mathcal{S}_2|\}$ and using the smoothness estimate $\ell_{\text{softmax}}$ together with the truncation relation $\varepsilon$, a straightforward calculation yields

$$T = \Theta\left(\frac{2^{3D(\mathcal{T})}S^9|\mathcal{H}|^9 D(\mathcal{T})^3\big(\max_{i\in\{1,2\}}|\Sigma_i|\big)^3\big(\max\{A,B\}\big)^{12}\big(\max\{\log A,\log B\}\big)^3}{(\min_{h\in\mathcal{H}}\mu_c(h))^3\,\epsilon^{18}}\log\frac{S\big(\max\{A,B\}\big)^2}{\epsilon}\right).$$

As in the direct-parametrization case, we now quantify the effect of stochastic gradients. For softmax-parametrized policies, Lemma F.2 shows that the REINFORCE estimator (combined with

the estimator for the entropic bidilated regularizer) is unbiased and has bounded variance per-trajector with $\sigma_\chi^2, \sigma_\theta^2 \leq \Theta\left(D(\mathcal{T})^2 + \tau 2^{D(\mathcal{T})}\right) = O\left(D(\mathcal{T})^2\right)$. We will control the stochastic error using mini-batches.

Substituting these into Theorem G.1 with the softmax pPŁ moduli from Lemma E.2,

$$\alpha_x = \Theta\left(\frac{\tau \min_{h\in\mathcal{H}} \mu_c(h)\,\gamma^3}{|\mathcal{H}|^3\left(1 + (A-1)e^{2R}\right)^2}\right), \qquad \alpha_y = \Theta\left(\frac{\tau \min_{h\in\mathcal{H}} \mu_c(h)\,\gamma^3}{|\mathcal{H}|^3\left(1 + (B-1)e^{2R}\right)^2}\right),$$

the stochastic error terms are controlled by

$$\frac{\sigma_x^2}{M\alpha_x} \leq \Theta\left(\frac{D(\mathcal{T})^2\,|\mathcal{H}|^3\left(1 + (A-1)e^{2R}\right)^2}{M\,\tau\,\min_{h\in\mathcal{H}} \mu_c(h)\,\gamma^3}\right),$$

$$\frac{\ell\,\sigma_y^2}{M\alpha_x\alpha_y^2} \leq \Theta\left(\frac{D(\mathcal{T})^2\,\ell\,|\mathcal{H}|^9\left(1 + (A-1)e^{2R}\right)^2\left(1 + (B-1)e^{2R}\right)^4}{M\,\tau^3\left(\min_{h\in\mathcal{H}} \mu_c(h)\right)^3\gamma^9}\right).$$

Requiring each to be at most $\epsilon$ gives the condition

$$M \geq \Theta\left(\max\left\{\frac{D(\mathcal{T})^2\,|\mathcal{H}|^3\left(1 + (A-1)e^{2R}\right)^2}{\epsilon\,\tau\,\min_{h\in\mathcal{H}} \mu_c(h)\,\gamma^3}, \frac{D(\mathcal{T})^2\,\ell\,|\mathcal{H}|^9\left(1 + (A-1)e^{2R}\right)^2\left(1 + (B-1)e^{2R}\right)^4}{\epsilon\,\tau^3\left(\min_{h\in\mathcal{H}} \mu_c(h)\right)^3\gamma^9}\right\}\right).$$

The second term dominates for small $\epsilon$, so it suffices to enforce

$$M \geq \Theta\left(\frac{D(\mathcal{T})^2\,\ell\,|\mathcal{H}|^9\left(1 + (A-1)e^{2R}\right)^2\left(1 + (B-1)e^{2R}\right)^4}{\epsilon\,\tau^3\left(\min_{h\in\mathcal{H}} \mu_c(h)\right)^3\gamma^9}\right).$$

To relate the dependence on $R$ to the truncation level, we use Lemma D.4, which implies that if the minimum action probability under the softmax parametrization is at least $\varepsilon$, then $1 + (A-1)e^{2R} \leq \frac{1}{\varepsilon}$, and $1 + (B-1)e^{2R} \leq \frac{1}{\varepsilon}$, so

$$\left(1 + (A-1)e^{2R}\right)^2\left(1 + (B-1)e^{2R}\right)^4 \leq \frac{1}{\varepsilon^6}.$$

Combining this with $\ell$-smoothness from above yields the bound

$$M \geq \Theta\left(\frac{D(\mathcal{T})^2\,\ell\,|\mathcal{H}|^9}{\epsilon\,\tau^3\left(\min_{h\in\mathcal{H}} \mu_c(h)\right)^3\gamma^9\,\varepsilon^6}\right).$$

Finally, we denote $S := \max\{|\mathcal{S}_1|, |\mathcal{S}_2|\}$ and substitute the terms $\gamma, \tau, \varepsilon$, together with the definition of $\ell_{\text{softmax}}$, a direct calculation shows that it is sufficient to choose

$$M = \Theta\left(\frac{2^{3D(\mathcal{T})}\,D(\mathcal{T})^3\,|\mathcal{H}|^9\,S^9\,\max_{i\in\{1,2\}}|\Sigma_i|\left(\max\{A,B\}\right)^{12}\left(\max\{\log A, \log B\}\right)^3}{\left(\min_{h\in\mathcal{H}} \mu_c(h)\right)^3\,\epsilon^{19}}\right).$$

$\square$

## H.3   Natural Policy Gradient

### H.3.1   The Fisher Information Matrix

$$\mathbf{F}(\chi) = \mathbb{E}_{s\sim d^{\chi,\theta}}\mathbb{E}_{a\sim\pi_\chi(\cdot|s)}\left[\nabla\log_\chi \pi_\chi(a|s)[\nabla_\chi \log \pi_\chi(a|s)]^\top\right]$$

The matrix $\mathbf{F}(\chi)$ is a blog diagonal matrix with its $(s,s)$-block being the matrix:

$$\mathbf{F}_s(\chi) = d^{\chi,\theta}(s)\left(\text{diag}(\pi_\chi(s)) - \pi_\chi(s)\pi_\chi(s)^\top\right).$$

Its pseudo-inverse, $\mathbf{F}^\dagger$, is again a block-diagonal matrix, with an $(s,s)$-block,

$$\mathbf{F}_s^\dagger(\chi) = \frac{1}{d^{\chi,\theta}(s)}\left(\text{diag}(\pi_\chi(s)) - \pi_\chi(s)\pi_\chi(s)^\top\right)^\dagger.$$

Interestingly, the matrix $\mathbf{Z} := \mathbf{F}^\dagger\mathbf{J}_{\text{softmax}}(\chi)$ is a block-diagonal matrix with entries $\frac{1}{d^{\chi,\theta}(s)}\mathbf{I}_{|\mathcal{A}_s|\times|\mathcal{A}_s|}$ on diagonal $(s,s)$-block.

**The spectrum of the Fisher Information Matrix**   With the same arguments used in Lemma D.1, we can conclude that,

- $\lambda_{\min}(\mathbf{F}(\chi, \theta)) = 0$;
- $\lambda_{\min}^+(\mathbf{F}_s(\chi, \theta)) \geq d^{\chi,\theta}(s) \min_a \pi_\chi(a|s)$;
- $\frac{\gamma^2 \min_h \mu_c(h)}{|\mathcal{H}|^2} \varepsilon \leq \lambda_{\max}(\mathbf{F}_s(\chi, \theta)) \leq 1$.

Hence,

- $\lambda_{\min}^+(\mathbf{F}(\chi, \theta)) \geq \min_{s,a} d^{\chi,\theta}(s) \pi_\chi(a|s)$;
- $\min_s \frac{1}{\sqrt{|\mathcal{H}||\mathcal{A}_s|}} \leq \lambda_{\max}(\mathbf{F}(\chi, \theta)) \leq 1$.

While, $d^{\chi,\theta}(s) \geq \frac{\gamma^2 \min_h \mu_c(h)}{|\mathcal{H}|^2}$ by Assumption 2.

**Theorem H.3.** *Alternating natural policy-gradient algorithm with softmax policy parametrization and the entropic bidilated regularizer converges in expectation in the last-iterate to an $\epsilon$-Nash equilibrium after a number of iterations $T$, that is*

$$T = \frac{1}{\epsilon^{36}} \, \mathsf{poly}\left(\tfrac{1}{\gamma}, |\mathcal{H}|, A, B, 2^{D(\mathcal{T})}, \tfrac{1}{\min_{h \in \mathcal{H}} \mu_c(h)}, |\mathcal{S}_1|, |\mathcal{S}_2|\right),$$

.

*Proof.* This theorem is again an application of Theorem G.1, now in its Mahalanobis form. For natural policy gradient, the updates are mirror-descent steps with a Mahalanobis metric induced by the Fisher information matrices, so we run Alt-GDA with $\mathbf{M}_{x,t} = \mathbf{F}_\chi(\chi_t, \theta_t)$ and $\mathbf{M}_{y,t} = \mathbf{F}_\theta(\chi_t, \theta_t)$.

By Lemma E.3, for a general positive-semidefinite metric matrix $\mathbf{M}$ the game satisfies a two-sided Mahalanobis pPŁ condition with moduli

$$\tilde{\alpha}_x = \Theta\left(\frac{\tau \min_{h \in \mathcal{H}} \mu_c(h)\, \gamma^3}{\lambda_{\max}(\mathbf{M}^{-1})\, |\mathcal{H}|^3 \big(1 + (A-1)e^{2R}\big)^2}\right),$$

$$\tilde{\alpha}_y = \Theta\left(\frac{\tau \min_{h \in \mathcal{H}} \mu_c(h)\, \gamma^3}{\lambda_{\max}(\mathbf{M}^{-1})\, |\mathcal{H}|^3 \big(1 + (B-1)e^{2R}\big)^2}\right).$$

When we specialize $\mathbf{M}$ to the Fisher information matrices, the spectrum bounds in the previous subsection together with Assumption 2 and the truncation assumption imply

$$\lambda_{\min}^+\big(\mathbf{F}_\chi(\chi, \theta)\big) \gtrsim \frac{\gamma^2 \min_{h \in \mathcal{H}} \mu_c(h)\, \varepsilon}{|\mathcal{H}|^2}, \qquad \lambda_{\min}^+\big(\mathbf{F}_\theta(\chi, \theta)\big) \gtrsim \frac{\gamma^2 \min_{h \in \mathcal{H}} \mu_c(h)\, \varepsilon}{|\mathcal{H}|^2},$$

and hence, over the image of the Fisher matrices,

$$\lambda_{\max}\big(\mathbf{F}_\chi^{-1}(\chi, \theta)\big), \ \lambda_{\max}\big(\mathbf{F}_\theta^{-1}(\chi, \theta)\big) = O\left(\frac{|\mathcal{H}|^2}{\gamma^2 \min_{h \in \mathcal{H}} \mu_c(h)\, \varepsilon}\right).$$

Substituting these bounds for $\lambda_{\max}(\mathbf{M}^{-1})$ into the expressions above yields Mahalanobis pPŁ moduli

$$\tilde{\alpha}_x = \Theta\left(\frac{\tau (\min_{h \in \mathcal{H}} \mu_c(h))^2\, \gamma^5\, \varepsilon}{|\mathcal{H}|^5 \big(1 + (A-1)e^{2R}\big)^2}\right), \qquad \tilde{\alpha}_y = \Theta\left(\frac{\tau (\min_{h \in \mathcal{H}} \mu_c(h))^2\, \gamma^5\, \varepsilon}{|\mathcal{H}|^5 \big(1 + (B-1)e^{2R}\big)^2}\right).$$

The Mahalanobis version of Theorem G.1 prescribes stepsizes (up to constants)

$$\eta_y = \Theta\left(\frac{1}{\ell\, \lambda_{\max}}\right), \qquad \eta_x = \Theta\left(\frac{\tilde{\alpha}_y^2}{\ell^3 \lambda_{\max}^2}\right), \qquad T = \Theta\left(\frac{\ell^3 \lambda_{\max}^2}{\tilde{\alpha}_x \tilde{\alpha}_y^2} \log \frac{1}{\varepsilon}\right),$$

where $\ell$ is the Euclidean smoothness constant of the objective and $\lambda_{\max} := \max_t \lambda_{\max}(\mathbf{M}_{\cdot,t}^{-1})$. We use the Euclidean smoothness constant $\ell := \ell_{\text{softmax}}$ as in the softmax-parametrized policy-gradient case; writing $\Sigma := \max_{i \in \{1,2\}} |\Sigma_i|$ and $S := \max\{|\mathcal{S}_1|, |\mathcal{S}_2|\}$,

$$\ell = \Theta\Big(16\sqrt{\Sigma}\, D(\mathcal{T}) + \tau\, 2^{D(\mathcal{T})} \Sigma\, D(\mathcal{T})\, S \max\{\log A, \log B\}\Big) = O\Big(D(\mathcal{T})\, \Sigma\Big),$$

where the final inequality uses the tuning of $\tau$ and $\epsilon < 1$. By the Smoothness Relative to the Mahalanobis Distance (as used in the proof of Theorem G.1), we have

$$\lambda_{\max} = O\left(\frac{|\mathcal{H}|^2}{\gamma^2 \min_{h \in \mathcal{H}} \mu_c(h)\, \varepsilon}\right),$$

and hence the stepsizes can be expressed as

$$\eta_y = \Theta\left(\frac{\epsilon^3 \min_{h \in \mathcal{H}} \mu_c(h)}{|\mathcal{H}|^2 D(\mathcal{T})\, \Sigma\, S\, (\max\{A, B\})^2}\right),$$

$$\eta_x = \Theta\left(\frac{\epsilon^{24}\big(\min_{h \in \mathcal{H}} \mu_c(h)\big)^6}{2^{2D(\mathcal{T})} |\mathcal{H}|^{14} D(\mathcal{T})^3 \Sigma^3 S^{10} (\max\{A, B\})^{16} (\max\{\log A, \log B\})^2}\right).$$

As in the softmax-parametrized policy-gradient case, we relate equilibria of the truncated, regularized, exploration-perturbed game to equilibria of the original game. An $\epsilon$-NE of the perturbed game is an $\epsilon'$-NE of the unregularized game with

$$\epsilon' = O\Big(\epsilon + \gamma + \tau \max_{i \in \{1,2\}} |\mathcal{S}_i| 2^{D(\mathcal{T})} \max\{\log A, \log B\} + \varepsilon \max_{i \in \{1,2\}} |\mathcal{S}_i| (\max\{A, B\})^2\Big),$$

so, as before, we choose

- $\gamma = \Theta(\epsilon)$;

- $\tau = \Theta\left(\frac{\epsilon}{\max_{i \in \{1,2\}} |\mathcal{S}_i| 2^{D(\mathcal{T})} \max\{\log A, \log B\}}\right)$;

- $\varepsilon = \Theta\left(\frac{\epsilon}{\max_{i \in \{1,2\}} |\mathcal{S}_i| (\max\{A, B\})^2}\right)$.

Combining these tunings with the expressions for $\tilde{\alpha}_x, \tilde{\alpha}_y$, the smoothness $\ell_{\text{softmax}}$, the bound on $\lambda_{\max}$, and the generic iteration bound $T = \Theta\big(\ell_{\text{softmax}}^3 \lambda_{\max}^2 / (\tilde{\alpha}_x \tilde{\alpha}_y^2) \log(1/\varepsilon)\big)$ and using Lemma D.4 to relate $R$ to the truncation level $\varepsilon$ yields

$$T = \Theta\left(\frac{2^{3D(\mathcal{T})} D(\mathcal{T})^3 |\mathcal{H}|^{19} S^{14} \Sigma^3 (\max\{A, B\})^{22} (\max\{\log A, \log B\})^3}{\epsilon^{33}\big(\min_{h \in \mathcal{H}} \mu_c(h)\big)^8} \log \frac{S(\max\{A, B\})^2}{\epsilon}\right).$$

$\square$

# I    Proximity of Projections

In this section, we consider that the update rules:

$$\overline{\theta} \leftarrow \theta_0 + \eta \mathbf{F}^\dagger(\theta_0)\nabla_\theta V(\theta);$$
$$\theta_{\text{F}} \leftarrow \underset{\theta \in \Theta_R}{\arg\min}\, (\theta - \overline{\theta})^\top \mathbf{F}(\theta_0)(\theta - \overline{\theta}); \tag{19}$$
$$\theta_{\text{kl}} \leftarrow \underset{\theta \in \Theta_R}{\arg\min}\, D_{\text{KL}}(\text{softmax}(\theta)\|\text{softmax}(\overline{\theta}), \tag{20}$$

and demonstrate that (19) and (20) are sufficiently close. For brevity we consider only the maximizer's updates and drop the minimizer's variables from the notation. *I.e.,* our goal is to bound $\|\theta_{\text{kl}} - \theta_{\text{F}}\|$. We begin by defining the two objective functions that each projection optimizes,

$$\mathcal{L}_{\text{F}}(\theta) := (\theta - \overline{\theta})^\top \mathbf{F}(\theta_0)(\theta - \overline{\theta});$$
$$\mathcal{L}_{\text{kl}}(\theta) := \text{lse}(\overline{\theta}) - \text{lse}(\theta) - \nabla\text{lse}(\theta)^\top (\overline{\theta} - \theta),$$

where $\text{lse}(\theta) := \log \sum e^{\theta_i}$. Then, we write,

$$\theta_{\text{F}} = \underset{\theta \in \Theta_R}{\arg\min}\, \mathcal{L}_{\text{F}}(\theta);$$
$$\theta_{\text{kl}} = \underset{\theta \in \Theta_R}{\arg\min}\, \mathcal{L}_{\text{kl}}(\theta).$$

Further, for the gradient of $\mathcal{L}_{\mathrm{kl}}$ we write,

$$\nabla \mathcal{L}_{\mathrm{kl}}(\theta) = \nabla^2 \mathrm{lse}(\theta)(\theta - \bar{\theta})$$
$$= \mathbf{F}(\theta)(\theta - \bar{\theta}).$$

Now, from stationarity of the optimal, for a $v$ in the normal cone of $\Theta_R$ at $\theta_{\mathrm{kl}}$,

$$0 = \nabla \mathcal{L}_{\mathrm{kl}}(\theta_{\mathrm{kl}}) + v$$
$$= \mathbf{F}(\theta_{\mathrm{kl}})(\theta_{\mathrm{kl}} - \bar{\theta}) + v$$
$$= \mathbf{F}(\theta_0)(\theta_{\mathrm{kl}} - \bar{\theta}) + [\mathbf{F}(\theta_{\mathrm{kl}}) - \mathbf{F}(\theta_0)](\theta_{\mathrm{kl}} - \bar{\theta}) + v$$

Therefore, we can bound the stationarity of $\theta_{\mathrm{kl}}$ for the objective of $\mathcal{L}_{\mathrm{F}}(\cdot)$:

$$\left\| \mathbf{F}(\theta_0)(\theta_{\mathrm{kl}} - \bar{\theta}) + v \right\| = \left\| [\mathbf{F}(\theta_0) - \mathbf{F}(\theta_{\mathrm{kl}})](\theta_{\mathrm{kl}} - \bar{\theta}) \right\|$$
$$\leq \left\| \mathbf{F}(\theta_0) - \mathbf{F}(\theta_{\mathrm{kl}}) \right\|_{\mathrm{op}} \left\| \theta_{\mathrm{kl}} - \bar{\theta} \right\|$$
$$\leq \frac{L_{\mathbf{F}}}{2} \left\| \theta_0 - \theta_{\mathrm{kl}} \right\| \left\| \theta_{\mathrm{kl}} - \bar{\theta} \right\|$$
$$\leq \frac{L_{\mathbf{F}}}{2} \left( \left\| \bar{\theta} - \theta_0 \right\| + \left\| \bar{\theta} - \theta_{\mathrm{kl}} \right\| \right) \left\| \theta_{\mathrm{kl}} - \bar{\theta} \right\|$$
$$\leq \frac{L_{\mathbf{F}}}{2} \left( \eta 2\sqrt{SB} + \eta 2\sqrt{SB} \frac{1}{\sqrt{\alpha}} \right) \eta 2\sqrt{SB} \frac{1}{\sqrt{\alpha}}$$
$$= O\left( \frac{L_{\mathbf{F}} SB \eta^2}{\alpha} \right).$$

where we use:

- $L_{\mathbf{F}}$ is the Lipschitz continuity modulus of the operator norm of $\mathbf{F}(\cdot)$,
- Proposition 8,
- $\alpha = \min_{\theta \in \Theta_{R'}} \lambda_{\min}^+(\mathbf{F}(\theta))$, and
- the fact that when $\tau$ is tuned as dictated in Theorem H.3:

$$\left\| \bar{\theta} - \theta_0 \right\| \leq \eta \left\| \mathbf{F}(\theta)^\dagger \nabla V^\tau(\theta) \right\|$$
$$\leq 2\eta\sqrt{SB}.$$

**Proposition 8.** Consider the update rules (19) and (20). It is the case that:

$$\left\| \theta_{\mathrm{F}} - \theta_{\mathrm{kl}} \right\| \leq \frac{1}{\sqrt{\alpha}} \left\| \theta_0 - \bar{\theta} \right\|,$$

where $\alpha = \min_{\theta \in \Theta_{R'}} \lambda_{\min}^+(\mathbf{F}(\theta))$ and $R' = R + \eta\sqrt{SB}$.

*Proof.* We begin by stating a useful fact.

**Fact 1.** Let lse be the function $\mathrm{lse}(\theta) := \log \sum_i e^{\theta_i}$. Then, $\mathrm{softmax}(\theta) = \nabla \mathrm{lse}(\theta)$. Further, $\mathcal{L}_{\mathrm{kl}}(\cdot)$ is strictly convex on $\Theta_\perp := \{\theta \in \mathbb{R}^d \mid \theta^\top \mathbf{1} = 0\}$ and its Bregman divergence is:

$$B_{\mathrm{lse}}(\theta' \| \theta) := \mathrm{lse}(\theta') - \mathrm{lse}(\theta) - \nabla \mathrm{lse}(\theta)^\top (\theta' - \theta).$$

Further, it is the case that:

$$B_{\mathrm{lse}}(\theta' \| \theta) = D_{\mathrm{KL}}(\mathrm{softmax}(\theta) \| \mathrm{softmax}(\theta')).$$

The arguments for this fact can be found in (Gao and Pavel, 2017). By standard calculations we can see that:

$$B_{\mathrm{lse}}(\theta' \| \theta) \geq \frac{\alpha}{2} \left\| \theta - \theta' \right\|^2, \quad \text{for } \theta, \theta' \in \Theta_\perp,$$

with $\alpha := \lambda^+_{\min}(\mathbf{F}(\theta))$. From Fact 1 we can conclude that,

$$B_{\mathrm{lse}}(\bar{\theta}\|\theta) \geq \frac{\alpha}{2} \left\| \theta_{\mathrm{kl}} - \bar{\theta} \right\|^2,$$

where we let $\Theta_{R'}$ for $R' = R + \eta\sqrt{SB}$. From $\frac{1}{2}$-smoothness of $\mathcal{L}_{\mathrm{kl}}(\cdot)$ we write,

$$\mathcal{L}_{\mathrm{kl}}(\theta_0) \leq \mathcal{L}_{\mathrm{kl}}(\bar{\theta}) + \nabla\mathcal{L}_{\mathrm{kl}}(\bar{\theta})^\top (\theta_0 - \bar{\theta}) + \frac{1}{4} \left\| \theta_0 - \bar{\theta} \right\|^2$$
$$= \frac{L}{2} \left\| \theta_0 - \bar{\theta} \right\|^2.$$

Since $\theta_{\mathrm{kl}} = \arg\min_{\theta \in \Theta_R} \mathcal{L}_{\mathrm{kl}}(\theta)$, it follows that

$$\frac{1}{4} \left\| \theta_0 - \bar{\theta} \right\|^2 \geq \frac{\alpha}{2} \left\| \theta_{\mathrm{kl}} - \bar{\theta} \right\|^2,$$

which concludes the claim.

$\square$