# OpenReview forum: "Policy Gradient Methods Converge Globally in Imperfect-Information Extensive-Form Games"
_NeurIPS.cc/2025/Conference — NeurIPS 2025 poster_

### Official Review · Reviewer_Re8E · 2025-06-27

**Clarity:** 3
**Significance:** 2
**Originality:** 2
**Rating:** 4
**Confidence:** 3

**Summary:**

The paper establishes the last‐iterate convergence guarantees for alternating policy‐gradient methods in zero‐sum, imperfect-information extensive‐form games. It observes that although the optimization over behavioral policies is nonconvex, each player’s utility is concave in the sequence-form strategy and satisfies a gradient-dominance property. By adding regularization, the authors show the utility further meets the Polyak–Łojasiewicz condition. They analyze three variants—direct parametrization, softmax parametrization, and softmax with natural policy gradients—and prove each converges in polynomial time to an $\epsilon$-approximate Nash equilibrium. Their results bridge theory between Markov games and extensive-form games, providing a principled foundation for the practical successes of MARL in settings like Poker and Hanabi.

**Questions:**

(Q1) This work is closely related to Zeng et. al. (2022), where they assume that the probability of choosing all actions are lower bounded by some constant. In this work, according to Assumption 1, the authors assume that all the actions need to be sufficiently explored. What are the main differences between this work and Zeng et. al. (2022), regarding key assumptions, analysis techniques, etc. ?

(Q2) What if the game exists multiple $\epsilon$-approximated NE? Is the current framework able to identify which NE it converges to?

(Q3) How would the result be to extend it to N-player? How would the polynomial-time algorithm depend on N?

Zeng et. al. (2022): Regularized Gradient Descent Ascent for Two-Player Zero-Sum Markov Games

**Ethical Concerns:**

["NO or VERY MINOR ethics concerns only"]

**Final Justification:**

I appreciate the author's response to my questions and I'll maintain my original evaluation.

**Limitations:**

Yes

**Quality:**

3

**Strengths And Weaknesses:**

Strength: The paper is in general well-written, recognizing the research gap and provide comparisons of Markov games and imperfect-information extensive-form games.

Weaknesses:
My major concern is the empirical validation of the work. I would appreciate it if the authors could provide further numerical or empirical results demonstrating the effectiveness of their algorithms, e.g. to compare all of the three algorithms proposed.

There are some additional questions listed in the below.

---

> ### Author Rebuttal · Authors · 2025-07-31
>
> Dear Reviewer,
>
> We thank you for your positive evaluation of our work. You raise some important points that we want to address. First, regarding the **numerical validation** of our results:
>
> We ran experiments comparing our last algorithm (alternating regularized natural policy gradient) to MMD (Sokota et al 2022) and CFR on Kuhn Poker, Leduc Poker, 2x2 Dark-Hex and Liar's Dice. In all of the experiments our algorithm's performance is comparable to the rest while in some cases it can overperform them.
>
> We pick our last algorithm as it makes for a more fair comparison and it is closer to TRPO and PPO algorithms.
>
> ---
>
> Now, answering your questions one by one:
> > * **Q1:** This work is closely related to Zeng et al. (2022), where they assume that the probability of choosing all actions is lower bounded by some constant. In this work, according to Assumption 1, the authors assume that all the actions need to be sufficiently explored. What are the main differences between this work and Zeng et al. (2022), regarding key assumptions, analysis techniques, etc.?
>
> The assumption in Zeng et al. is an assumption on the problem. It is rather stronger than what you are mentioning. Zeng et al. assume that after regularization, there exists a problem-dependent constant, independent of the coefficient of regularization, that lower bounds the probability of all actions in a Nash equilibrium. They consider an uncosntrained parameter space as they do not need to lower bound the smallest probability of an action.
>
> Our ``assumption'' is not on the side of the game but rather on the algorithm. We also need to control the lower bound of an all actions' probability so we consider **constrained** softmax parametrization.
>
>
> > * **Q2:** What if the game has multiple \$\epsilon\$-approximated Nash equilibria? Is the current framework able to identify which NE it converges to?
>
> The regularized game always has a unique Nash equilibrium as it is a strongly monotone game [1] (Theorem 2).
>
> Nonetheless, our convergence proof does not require uniqueness.
>
> > * **Q3:** How would the result extend to the N-player case? How would the polynomial-time algorithm depend on N?
>
> This is a very interesting question. In general, computing an approximate Nash equilibrium even for 3-player zero-sum games is $
>
> ---
>
> We hope to have addressed your concerns sufficiently,
>
> Sincerely,
>
> The authors
>
>
> ---
>
> [1] Rosen, J.B. "Existence and Uniqueness of Equilibrium Points for Concave N-Person Games." Econometrica

---

> ### Author Response · Authors · 2025-08-06
>
> Dear Reviewer,
>
> As the end of the discussion period is around the corner, we would like to know whether we have satisfactorily addressed your concerns. Please let us know.
>
>
> Best,
>
> The authors

---

> > ### Comment · Reviewer_Re8E · 2025-08-08
> >
> > I appreciate the authors' response to my questions and I'll maintain my original evaluation.

---

### Official Review · Reviewer_jMiC · 2025-07-02

**Clarity:** 1
**Significance:** 3
**Originality:** 2
**Rating:** 3
**Confidence:** 3

**Summary:**

The paper shows that the utility of an extensive-form game has hidden convexity when min- and max-players' policy are tabular and soft-max policies, sufficient exploration is conducted, and the utility is augmented with bidilated regularizers. Based on this result, the paper provides three self-play policy gradient methods whose last-iterate converges to a Nash equilibrium.

**Questions:**

Would you provide a detailed proof of Lemma E.2? Since $\mu_1^\pi$ and $\mu_2^\pi$ do not seem to correspond to policy parameters one-to-one, I do not really understand why the proof of Lemma E.1 can be used for proving Lemma E.2.

**Ethical Concerns:**

["NO or VERY MINOR ethics concerns only"]

**Final Justification:**

While proving the global last-iterate convergence of the policy gradient methods under different policy parameterization based on hidden convexity is a nice contribution, the paper fails to clearly present what are new results in the paper. This writing issue seems to be critical, so I keep my score.

**Limitations:**

Yes.

**Paper Formatting Concerns:**

No.

**Quality:**

2

**Strengths And Weaknesses:**

# Strength

The paper provides a novel look at algorithms for solving extensive-form games from the perspective of hidden convexity. It also shows that policy gradient methods have the last-iterate convergence property, which I think is new and very important for practical reason.

# Weakness

The main weakness of this paper is writing. It was difficult to understand core technical contributions of the paper and which results are new. To me, many theoretical results seem to be straightforward applications of existing results except Lemma E.1 - E.3, which I think are the core technical contributions of the paper and show the utility of an extensive-form game has hidden strong convexity under various parameterization.

I would remove explanation of Markov games and comparison of it to extensive-form games and focus more on explaining the importance and novelty of E.1 - E.3. Furthermore, it would be very much appreciated what technical difficulties exist for deriving presented results. Currently, too much pages are devoted to the introduction of background knowledge, some of which do not really need to be required in the main paper (e.g., Markov games and hidden convexity).

Also, careful revisions are required. Apart from simple typos, it would be very much appreciated if details of proofs are added. For example, in Lemma C.2, I am confused what $\alpha_{\mathrm{dil}}$ is. Another example is the proof of Lemma E.1. There, it is written that "For player 1, we know that the functions is strongly convex with an appropriate weight scheme...", but it would be nice to explicitly mention which theoretical result you are referring to. (I guess Corollary C.1, right?)

A minor weakness is that obtained convergence rates for proposed algorithms are slower than convergence rates for previous algorithms, as mentioned in Conclusion of the paper.

---

> ### Author Rebuttal · Authors · 2025-07-31
>
> Dear Reviewer,
>
> We are glad that you validate the originality and importance of our key technical contribution (last-iterate convergence of policy gradient methods for imperfect-information EFGs). We thank you for the time you took to highlight your concerns. We will do our best to address them.
>
> ---
> ### Presentation and core contributions
>
> Our goal with this paper is to also bridge the two communities of mutli-agent RL and EFGs. We offered an extensive presentation in order to serve this purpose. Nonetheless, we agree with you that part of the preliminaries can be moved to the appendix and that we should elaborate on our results with commentaries and an exposition of technical challenges. That is, the comparison of the two regularizers can be moved to the appendix, the lengthy exploration note also. Further, the extended comparison of EFGs and MGs can be moved after the main results as part of the concluding remarks.
>
> Further, hidden convexity is crucial in proving the gradient domination properties. As such, we deem it important that it is part of the main text.
>
>
> ---
> ### Regarding Lemma E.1-E.3
> They establish the proximal-PL condition of the value function under different policy parametrizations. Hidden strong convexity is a direct consequence of regularization. I.e., in the sequence-form, the utility is concave since it is linear for each player's sequence-form strategy. Then, adding a strongly concave regularizer makes it strongly concave. What is left to show using Lemmas E.1-E.3 is that this translates to a proximal-PL condition for the parameters of the policies.
>
> ---
> ### Proof sketch of Lemma E.2
> The main argument is the following (a full proof is deferred to the new draft). In Lemma E.1, we establish the KL condition satisfied for the policies (the concatenation of simplices, not the parameters). Now, we are interested in translating policy KL condition to a KL condition for the parameters of the softmax policies. In turn, we can get the pPL condition from Lemma B.2.
>
> Let us denote:
> *  $f(\theta)$ to be the utility as a funciton softmax parameters, and
> *  $v(\pi)$ the as a function of policies
>
> also, let $I_\Theta, I_\Pi$ be the indicator functions of the convex feasibility sets $\Theta, \Pi$ respectively. Last, let $c:\Theta \to \Pi$ denote the transform from parameters to policies.
>
> We now observe by the chain-rule of subgradients $s_\theta \in \partial (f + I_{\Theta})(\theta), ~ s_x \in \partial (v + I_{\Pi})(\pi):$
> $$\nabla_\theta c(\theta) s_\pi = s_\theta.$$
>
>
> Although the transform is not 1-1, we see that its jacobian, $\nabla_\theta c(\theta)$, is rank deficient with a unique zero eigenvalue which corresponds to an eigenvector of all-ones $\mathbf{1}$. This allow us to transform the KL condition for policies to a KL condition of parameters using the pseudo-inverse of $\nabla_\theta c(\theta)$.
>
> ---
>
> ### Weighting Scheme
> We use the scheme in (6) found in [1]. A simple implementation would be the following:
> * start at the leaves and give a weight of $1$,
> * for internal nodes, sum the weights $w_s$ of each action $a$'s children nodes, $W_a$. Pick the maximum $\max_{a} W_a$, and set that node's weight to be 2 $\max_{a} W_a$.
>
> ---
>
> Again, we thank you for your time and we hope to have satisfactorily addressed your concerns.
>
> [1] Kroer, C., Waugh, K., Kılınç-Karzan, F. and Sandholm, T., 2020. Faster algorithms for extensive-form game solving via improved smoothing functions. Mathematical Programming.

---

> > ### Comment · Reviewer_jMiC · 2025-08-06
> >
> > Thank you very much for the response.
> >
> > It answered my questions. However, as I (and other reviewers too) wrote, the paper has a major issue in clarity, and I am not sure if it can be fixed in the camera-ready version. Accordingly, I would like to keep my score.
> >
> > I think the result is interesting and is of great interest to the community. Therefore, I highly encourage the authors to polish the paper and resubmit it to another conference. (Also, it would be highly appreciated if the authors upload the paper to ArXiv)

---

> > > ### Author Response · Authors · 2025-08-06
> > >
> > > Dear Reviewer,
> > >
> > > We are happy to have answered your concerns and that you recognize the significance of our contributions!
> > >
> > > Regarding presentation: We re-assure you that we have been working on the presentation and we are confident that we can turn-in a much improved camera-ready version if we were to do it.
> > >
> > > Best,
> > >
> > > The Authors

---

### Official Review · Reviewer_k6sM · 2025-07-02

**Clarity:** 3
**Significance:** 2
**Originality:** 2
**Rating:** 4
**Confidence:** 4

**Summary:**

This paper proves global last-iterate convergence of three policy gradient variants, namely direct parameterization, softmax parameterization (with Policy Gradient and Natural Policy Gradient),, to an $\epsilon$-approximate Nash equilibrium in imperfect-information, perfect-recall, zero-sum extensive-form games. The convergence occurs in polynomial time in $1/\epsilon$ and game parameters. The key idea is that the nonconvex policy landscape becomes gradient dominated due to hidden concavity in sequence-form strategies. This enables analysis via the Polyak-Łojasiewicz (PŁ) condition, bypassing traditional regret-based arguments.

**Questions:**

See the weaknesses section.

**Ethical Concerns:**

["NO or VERY MINOR ethics concerns only"]

**Final Justification:**

I would like to sincerely thank the authors for their responses during the discussion phase. I decided to maintain my score. In particular, in the camera-ready version, I would like to suggest the authors to be more careful and clear about certain mathematical claims, such as including a more detailed rate without the big O notation hiding a lot of details on the dependence on different parameters.

**Limitations:**

Yes.

**Paper Formatting Concerns:**

No.

**Quality:**

3

**Strengths And Weaknesses:**

Strengths:

The paper does a great job reviewing relevant literature and situating the work in the broader context of MARL, EFGs, and nonconvex optimization theory. Additionally, the intuitive explanation of the hidden convexity and gradient domination properties is helpful for readers to grasp the key idea of the paper. While I have not examined every detail of the proof, the overall reasoning is clear and compelling. Overall, the paper is well-written and easy to follow.

Weaknesses:

While the results are promising, the presentation of the main theorems is overly brief, and many important details are deferred to the appendix. For instance, the bound is given as $T = \text{poly}(1/\varepsilon, \dots)$, but the actual order of the polynomial is never made explicit, which weakens interpretability. In particular, in MARL literature, natural policy gradient (NPG) methods are generally shown to be more efficient than policy gradient (PG) especially for softmax parameterization. However, the way that the authors show the results make it unclear whether this is still true in the EFG settings.

Further, the role and selection of the regularization parameter $\tau$ is not well-explained. How does $\tau$ affect the convergence rate or approximation error? Is there guidance on choosing $\tau$ in practice? Also, there seems to be a notation conflict, as $\tau$ is used both as the regularization coefficient and to denote trajectories (in Definition 3), which may confuse readers.

---

> ### Author Rebuttal · Authors · 2025-07-31
>
> Dear Reviewer,
>
> Thank you for your time and your valuable feedback.
>
> * We are updating our draft to make these rates explicit. Nonetheless, we stress that it was not previously known whether policy gradient methods converge to a Nash equilibrium in a polynomial number of iterations in EFGs.
>
> * The main reason as to why alternating regularized NPG has a slower convergence rate is because (i) we cannot use the arguments of [1] which are tailored to the Markov game structure (ii) we need to explicitly control the minimum node reach probability using a constraint on the parameter space. Consecutively, the minimum action probablity inversely scales the Mahalanobis-smoothness of the utility function and in turn the convergence rate.
>
> * $\tau$ needs to be tuned as $\tau \gets O(\epsilon)$ where $\epsilon$ is the accuracy of the desired $\epsilon$-approximate Nash equilibrium. It trade-offs rate-of-convergence with an exploitability lower bound. I.e., a larger $\tau$ guarantees faster convergence but it will introduce a larger bias on the exploitability of the policy profile reached at convergence.
>
> * Thank you for bringing this up. We already corrected it in our draft by using $\xi$ for trajectories.
>
>
> With this note, we hopefully have addressed your concers,
>
> Thank you,
>
> The authors
>
> ---
>
> [1] Shicong Cen, Yuejie Chi, Simon S Du, and Lin Xiao. Faster last-iterate convergence of policy369
> optimization in zero-sum markov games.

---

### Official Review · Reviewer_JY7P · 2025-07-02

**Clarity:** 1
**Significance:** 2
**Originality:** 2
**Rating:** 3
**Confidence:** 4

**Summary:**

This paper address the issue of establishing convergence to an approximate Nash equilibrium ($\varepsilon$-NE) of policy gradient-based methods for solving imperfect-information, two-player, zero-sum extensive-form games (EFGs). The main idea is to (a) consider the EFG in sequence-form, which is known to be solvable via an equivalent bilinear optimization problem over the players' strategies in sequence-form, (b) restrict to simple classes of policy parameterizations (i.e., tabular, softmax) providing a well-behaved mapping from policy parameter space into sequence-form space, (c) suitably regularize each player's utility to ensure that each agent's utility satisfies a one-sided Polyak-Lojasiewicz (PL) condition in the agent's policy parameters, and (d) leverage the PL property to establish convergence of policy gradient descent-ascent algorithms to an $\varepsilon$-NE under suitable conditions. The primary contributions are the establishment of a certain two-sided PL condition for the regularized payoff and convergence analysis for vanilla policy gradient (REINFORCE) with tabular and softmax parameterizations, as well as natural policy gradient with softmax parameterization.

**Questions:**

The following questions may help improve clarity of the paper:
1. What is the main point of the discussion lines 49-56? How does value iteration fail?
2. What is the main point of the discussion lines 57-74? How is it relevant to the rest of the paper?
3. How does one go from the single-variable PL condition of lines 101-107 to a saddle point of the two-variable objective $f$ around line 101?
4. How are the information sets in Definition 1 defined? Which player is maximizing in Definition 1?
5. How exactly are $\sigma, \sigma_1, \sigma_2, \pi_1, \pi_2$ defined?
6. How exactly is $\mu_1^{\pi_1}$ defined in the mention of sequence-form strategies around lines 184-185?
7. In definition 2, should the LHS be $V^{\pi_1^*, \pi_2} - \epsilon$?
8. How are "summed multiplied" and "total reach probability" defined on line 192?
9. How is "reach probability of each history" defined on line 197?
10. Under what conditions is the REINFORCE estimator "an unbiased estimator of the policy gradient", as claimed on line 221?
11. What is $K_\tau$ in Definition 3?
12. In Lemma 2.1, since there is a set of decision variables corresponding to each player in the utility, in what sense does a given player's utility satisfy the PL condition? Clarify the leap to the "two-sided" PL condition being made here.
13. What is the key technical challenge in the proof of Lemma 2.1? What is the significance of the result?
14. What is the key technical challenge in the proof of Theorem 3.1? What is the significance of the result?
15. What is the key technical challenge in the proof of Theorem 3.2? What is the significance of the result?
16. What is the key technical challenge in the proof of Theorem 3.3? What is the significance of the result?
17. Does the bound in Theorem 3.1 have exponentially growing dependence on the problem horizon? If so, how does this impact the claim of polynomial complexity in the abstract and introduction?
18. What is the "size of the game" in line 286?

**Ethical Concerns:**

["NO or VERY MINOR ethics concerns only"]

**Final Justification:**

While the authors have made a substantial and appreciated effort to address the significant lack of clarity in the submission (see Weaknesses 1 & 2 and the 18 Questions in my initial review), the amount of revisions that will be required to address all these issues goes significantly beyond what can be expected for the camera-ready version. I therefore maintain that the submission is not ready for publication in its current form.

**Limitations:**

See weaknesses and questions above for limitations.

**Quality:**

2

**Strengths And Weaknesses:**

**Strengths:**
1. *Relevance to the community:* Convergence analyses for multi-agent reinforcement learning (MARL) methods applied to games is currently an active area, and the particular imperfect-information EFG considered in this work is a game of fundamental importance. For this reason, this work is likely of interest to the theoretical MARL community. In addition, the substantial body of recent work establishing convergence to global optimality of single-agent policy gradient methods using PL and similar conditions has only just started to be leveraged in the MARL for games setting, making the subject of this work quite timely and increasing its interest to the community.
2. *Well-motivated:* The core idea of establishing "hidden convexity-concavity" of the regularized payoff and leveraging this condition to establish convergence to an $\varepsilon$-NE is both sound and interesting. Leveraging the nicely structured sequence-form of the EFG is a natural idea, and the steps (a)-(d) outlined in the **Summary** make clear sense as a way to achieve this.

**Weaknesses.** Despite the strengths listed above, the paper has significant clarity issues making it difficult to accurately evaluate the technical contribution of the work. Major clarity issues include:
1. *Technical contribution:* It is unclear from the main body of the paper what the core technical innovations of the analysis are and what key challenges were overcome in order to achieve the analysis. Lemma 2.1 and Section 3, which constitute the main contributions of the paper, offer little to no discussion is provided offering insight into the meaning behind and challenges in obtaining the results. In addition, though the appendix contains much content, there is little to no exposition tying the results together and relating them back to the main body.
2. *Key concepts poorly defined:* Important concepts, such as the definition of sequence-form (discussed around lines 184-185) and the explicit statement of the two-sided PL condition (not defined in the main body) are not clearly defined in the main body of the paper. Since these are central concepts for the paper, this significantly weakens the ability of the reader to accurately evaluate the work. Without a clear definition of the sequence form it is difficult to see the critical relationship between the policy parameterizations and bilinearity in sequence-form that make hidden convexity-concavity of the regularized objective possible. Without explicit statement and discussion of the PL condition, it is likewise hard to see why this hidden convexity-concavity holds. As with weakness 1, the appendix does provide some additional content (Sec. C and E for sequence-form and PL condition), but there is little to no exposition tying the results together and relating them back to the main body.

---

> ### Author Rebuttal · Authors · 2025-07-31
>
> Dear Reviewer,
>
> We thank you for your time and the extensive valuable comments. We will answer them one by one.
>
> ---
>
> ### Regarding the highlighted weaknesses
> Our **key technical contributions** were providing the first polynomial time policy gradient convergence results for imperfect-information EFGs. We first establish a gradient domination condition for the highly nonconvex utility function of EFGs -- this step is a cornerstone for our analysis. In general, we cannot prove polynomial-time convergence in min-max functions without some gradient domination property.
>
> Then, we analyze alternating mirror descent ascent with changing mirrors for constrained min-max optimization in the presence of inaccurate gradients.
>
> ---
>
> ### Key concepts: Sequence-form strategies, hidden convexity, and pPL condition
> We agree that a clearer definition of the sequence-form strategy would improve exposition. We omitted a longer discussion on the sequence form as it is considered quite standard in the literature of EFG. The sequence form is a convex polytope defined from linear constraints. Intuitively, a sequence-form strategy is a nonnegative vector indexed by a player’s sequences, where a “sequence” is the ordered list of that player’s own actions from the root (including the empty sequence). Each entry gives the probability that the player realizes exactly that sequence under their behavioral choices.
>
> We clarify now that in terms of sequence-form the utility of each player is concave (linear for each player's sequence-form strategy). Since, there is a transform from behavioral strategies to sequence-form, the utility is hidden-concave automatically. Now, hidden concavity can be translated to gradient domination (weak or proximal PL).
>
>
> ---
> ### Answers to questions
>
> >1. What is the main point of the discussion lines 49–56? How does value iteration fail?
>
> This discussion demonstrates why our proposed solution, policy gradient methods, is needed and how other RL algorithms (used for Markov games) fail for incomplete-information extensive-form games.
>
> Nash value-iteration fails because policies at the root of each sub-tree alters the q-values of their children nodes. We can check this by a simple example. Consider the game of matching pennies transformed into an imperfect-information extensive-form game.
>
> * At the root, picks a side of a coin that is not revealed to their opponent.
>
> * This choice will lead to two different histories $H$ or $T$.
>
> * Since the coin is hidden, player $2$ does not distinguish between the two histories and they lie in the same info-set.
>
> Assume that player $1$ initializes their strategy with the uniform strategy; the q-function cannot be defined otherwise.
>
> This means that player $2$ q-function at their info-set is $(0,0)$. Any strategy is a best-response.
>
> Nash value iteration would call for a bottom-up solution. I.e., it prescribes that the policy of each player is a Nash equilibrium of the joint q-function.
> Since agent do not act simultaneously and the last decision is taken by player $2$. Player $2$ picks a minimizing strategy which is any strategy ($\arg \min_{\pi_2 \in \Delta} \pi_2^\top q = \Delta$ ).
>
> Climbing the tree to the root, player $1$ can alter their strategy by picking an appropriate best-response. This process does not lead to a Nash equilibrium. The same would be true for any initialization of player $1$'s strategy.
>
> > 2. What is the main point of the discussion lines 57–74? How is it relevant to the rest of the paper?
>
> This paragraph highlights how different gradient domination conditions hold across different fields of ML.
>
> In our case, the proximal-PL condition fosters the validity of our convergence arguments and also the empirical success of policy gradient algorithms for incomplete information EFGs.
>
> > 3. How does one go from the single-variable PL condition of lines 101–107 to a saddle point of the two-variable objective \$f\$ around line 101?
>
> Essentially, a two-sided PL condition for $\min_x \max_y f(x,y)$ means that $f(\cdot, y)$ is PL for any $y$, and $f(x,\cdot)$ is PL for any $x$.
>
> 4. How are the information sets in Definition 1 defined? Which player is maximizing in Definition 1?
>
> Information sets are defined as sets/groupings of *histories* of the EFG. I.e., they are the underlying states of the game indistinguishable to one another since the players to not have full information. E.g., in poker, a player knows what their opponent has bet, what cards are on the table, and what cards are in their own hands. But, since they do not know what kind of cards the opponent is holding, all possible states of the opponent's hand are in the same infoset.
>
> Player $1$ gets *payoff* $r$, and so is considered to be maximizing in this definition.
>
> > 5. How exactly are \$\sigma, \sigma\_1, \sigma\_2, \pi\_1, \pi\_2\$ defined?
>
> * $\sigma(\cdot)$ takes is a typo. It should be $\sigma_1$ or $\sigma_2$ depending on the player considered
> * $\sigma_1, \sigma_2$ take as input an info-set and return the last history where player 1 (or player 2 respectively) took an action.
> * $\pi_1, \pi_2$ are each a concatenation of simplices, one for each information-set.
>
> > 6. How exactly is \$\mu\_{1,\pi\_1}\$ defined in the mention of sequence-form strategies around lines 184–185?
>
> It is the sequence-form strategy that corresponds to behavioral strategy (or policy in RL terms) $\pi_1$.
>
> > 7. In Definition 2, should the LHS be \$V\_{\pi\_1^\*,\pi\_2} - \epsilon\$?
>
> Yes, thanks for bringing this up.
>
>
> > 8. How are "summed multiplied" and "total reach probability" defined on line 192?
>
>
> The total reach probability is defined as the probability of reaching a particular infoset for a particular policy profile $\pi_1, \pi_2$ taking into consideration chance.
>
> Also, this is a typo. The bidilated regularizer is defined using a strongly convex function multiplied by the latter *total reach probability*.
>
> > 9. How is "reach probability of each history" defined on line 197?
>
> This is $E_{h' \sim \pi_1, \pi_2}[\mathbf{1}\{h' = h\}]$. I.e., the probability of reaching a particular history conditioned on the policy profile $\pi_1, \pi_2$.
>
>
> > 10. Under what conditions is the REINFORCE estimator "an unbiased estimator of the policy gradient," as claimed on line 221?
>
> REINFORCE is generally an unbiased estimator of the policy gradient. The variance can be bounded under the assumption of $\epsilon$-greedy parametrization for direct-param. and generally bounded for softmax policy param.
>
>
> > 11. What is \$K\_\tau\$ in Definition 3?
>
> $K_\tau$ is the length of trajectory $\tau$.
>
> > 12. In Lemma 2.1, since there is a set of decision variables corresponding to each player in the utility, in what sense does a given player's utility satisfy the PL condition? Clarify the leap to the "two-sided" PL condition being made here.
>
> For the utility function $V(\pi_1, \pi_2)$, two-sided proximal-PL means that $V(\cdot, \pi_2)$ satisfies the proximal-PL for any $\pi_2$ and so does $V(\pi_1, \cdot)$ for any $\pi_1$.
>
> > 13. What is the key technical challenge in the proof of Lemma 2.1? What is the significance of the result?
>
> Lemma 2.1 establishes a gradient domination condition for the utility functions of incomplete-information EFGs. This property is crucial and opens the possibility of provable gradients of policy gradient methods.
>
> > 14. What is the key technical challenge in the proof of Theorem 3.1? What is the significance of the result?
>
> The significance of this result is a first provable convergence guarantee of policy gradient methods for incomplete-information EFGs.
>
> The key challenge of this theorem was handling the gradient inaccuracy error $\delta$ which is controlled using the regularization coefficient $\tau$.
>
> > 15. What is the key technical challenge in the proof of Theorem 3.2? What is the significance of the result?
>
>
> The key challenge was coming up with feasible set of parameters $\chi, \theta$ that will guarantee every action is played using a controllable lower bound on the probability while still retaining convex (both in the corresponding restricted policy space and the parameter space Lemma D.3 & D.4)
>
> The significance comes from the fact that we consider the first algorithm that goes beyond direct policy parametrization and make way for future work where the softmax parametrization uses some neural networks.
>
> > 16. What is the key technical challenge in the proof of Theorem 3.3? What is the significance of the result?
>
> The key technical challenge comes from the fact that we had to interpret natural gradient as mirror descent with changing mirror. We refactor the previous convergence proofs to accommodate for the changing mirrors and control errors introduced by this change.
>
> > 17. Does the bound in Theorem 3.1 have exponentially growing dependence on the problem horizon? If so, how does this impact the claim of polynomial complexity in the abstract and introduction?
>
> As we note, the convergence rate depends does depend on some constant exponentiated by the height of the game tree. But, the height of the game tree is in general a logarithm of the game's total size.
>
> > 18. What is the "size of the game" in line 286?
>
> The size of the game is typically the size of the action space and the number of histories of the game.
>
> ---
>
>
> We hope to have addressed your concerns sufficiently and we would love to answer possibly more questions,
>
> Thank you again,
>
> The authors

---

> > ### Comment · Reviewer_JY7P · 2025-08-07
> >
> > Thanks to the authors for their rebuttal. Some additional remarks follow:
> > * **Regarding response to Weakness 1 (W1):** It seems like my initial concern was misunderstood, so I'll rephrase: what specific technical challenges did you overcome to achieve the analysis, and why was this a significant improvement over existing analysis methods (be explicit)?
> > * **Regarding response to W2:** Much of the lack of clarity in the paper stems from sequence-form and related notation not being explicitly defined. Explicit definitions are needed to improve clarity of the paper.
> > * **Regarding responses to Questions 1, 3, 4, 5, 8, 11, 12, 15, 16, 18:** Including careful, explicit discussions on all ten of these points will help improve clarity.
> > * **Regarding responses to Q10:** REINFORCE-style estimators are not unbiased, in general: in the discounted case, using finite rollout lengths causes bias, while in the average-reward case, not being able to sample from the stationary occupancy measure causes bias. The claim that the REINFORCE-style estimators are unbiased in your case is not automatic and needs to be explicitly proven.
> > * **Regarding responses to Q13 and Q14:** The key technical innovation required to achieve this part of the analysis remains unclear.

---

> ### Author Response · Authors · 2025-08-08
>
> Dear Reviewer,
>
> Thank you for engaging in this conversation in such detail, let us further elaborate,
>
> > Re: Technical challenges
>
> The technical challenges we overcame were:
> * shedding light on a nonconvex optimization landscape of min-max (see also answer in Re:Q13)
> * proving last-iterate convergence of a policy gradient method for **constrained** **nonconvex** min-max optimization (most previous result concern unconstrained optimization and the difference is non-trivial)
> * managing to overcome the difficulties fostered by **imperfect-information** EFGs which make the q-values radically different from that of Markov games. In finite-horizon Markov games, policies of past time-steps do not interfere with the q-values of future states. In imperfect-information EFGs, past time-steps change future q-values. This renders Markov game literature that considers policy optimization/gradient methods largely ineffective in getting provable guarantees. We overcame this challenge by tackling the nonconvex optimization landscape head-on after proving that it fosters certain structure (PL condition).
>
>
> > Re: Q10
>
> In Lemma F.1 we prove that REINFORCE is unbiased for imperfect-information EFGs.
>
> > Re: Q13
>
> Lemma 2.1 is a sufficient condition for policy gradient methods to converge. In general, even for the a single agent, the utility function is nonconvex in the control variables. Therefore, without a guarantee like that of Lemma 2.1, there is no reason as to why would policy gradient would converge to an optimal solution. Nonetheless, it is folklore that gradient descent converges to a stationary point. Lemma 2.1 guarantees that any stationary point is also a maximizer of the utility function.
>
> Further, we note that for min-max optimization, there is no known algorithm that converges to even a stationary point in time that is polynomial in $1/\epsilon$ and it remains an open question [1,2]. But, when a gradient domination condition holds (like that of Lemma 2.1), there can be algorithms with provable convergence to stationary points [3].
>
> To our knowledge, no work proves the gradient domination property for EFG utilities before us. It might seem natural, but it was not studied before.
>
> > Re: Q14
>
> Building upon Lemma 2.1, this result contributes the first guarantee that policy gradient methods can provably converge to a Nash equilibrium in imperfect-information EFGs in time that is polynomial in $1/\epsilon$ and the natural parameters of the game.
>
>
> Please, let us know about further concerns,
>
> Best,
>
> The authors
>
> ---
>
> [1] Daskalakis, C., Skoulakis, S. and Zampetakis, M., 2021, June. The complexity of constrained min-max optimization. In Proceedings of the 53rd Annual ACM SIGACT Symposium on Theory of Computing (pp. 1466-1478).
>
> [2] Anagnostides, I., Panageas, I., Sandholm, T. and Yan, J., 2025. The Complexity of Symmetric Equilibria in Min-Max Optimization and Team Zero-Sum Games. arXiv preprint arXiv:2502.08519.
>
> [3] Yang, J., Orvieto, A., Lucchi, A. and He, N., 2022, May. Faster single-loop algorithms for minimax optimization without strong concavity. In International conference on artificial intelligence and statistics (pp. 5485-5517). PMLR.

---

> > ### Comment · Reviewer_JY7P · 2025-08-08
> >
> > Thanks to the authors for their comments, which have mitigated some of the many clarity issues in the submission. Given that significant revisions are required to adequately address all these issues, however, I believe the submission is not ready for publication in its current form.

---

### Note · Authors · 2025-08-16

We would like to thank the reviewers for their extensive comments and effort.

We are glad that the reviewers have recognized the importance of our contributions unequivocally. Their biggest concern has been presentation and whether we can make the improvement for the camera-ready version. Some suggested that we should submit to the next conference. We share the feeling that part of the draft can be improved; we have been continuously improving the draft since submitting it. We also note that the camera-ready version is to be submitted more or less the same date as the submission deadline for the next major conference.

Hence, we strongly believe that all presentation concerns will be addressed in time in case the AC decides to accept our work.

Thank you,

The authors

---

### Decision · Program_Chairs · 2025-09-17

**Decision:**

Accept (poster)

**Comment:**

This paper establishes last-iterate convergence guarantees for alternating policy gradient methods in imperfect-information EFGs, leveraging hidden convexity in sequence-form strategies and the Polyak–Łojasiewicz condition.

During the discussion and rebuttal, reviewers raised concerns primarily about clarity and exposition, as well as the need for more explicit details in the main text. The authors responded constructively by elaborating on the technical challenges, clarifying key definitions, and providing preliminary empirical validation on benchmark games. While some reviewers remained unconvinced that all clarity issues could be resolved within a camera-ready revision, others emphasized that the core contributions are significant, well-situated in the literature, and relevant to the MARL/EFG community.

Overall, the theoretical contribution is solid and timely, and the additional validation provided during the rebuttal period further strengthens the submission. Given the novelty, technical depth, and potential impact, this work leans toward acceptance.